# Aerosol radiative effects with MACv2

S.Kinne, MPI-Meteorology, Hamburg, Germany (e-mail: stefan.kinne@mpimet.mpg.de)

## Abstract

Monthly global maps for aerosol properties of the MACv2 climatology are applied in an off-line radiative transfer model to determine aerosol radiative effects. This model-setup cannot address rapid adjustments by clouds, but current evidence suggests their contribution to be small when compared to the instantaneous radiative forcing. Global maps are presented to detail the regional and seasonal variability associated with (annual) global averages. Radiative effects caused by the aerosol presence (direct effects) and by aerosol modified clouds (indirect effects) are examined. Direct effects are determined for total aerosol, anthropogenic aerosol and extracted individual aerosol components. Indirect effects cover the impact of reduced cloud drop sizes by anthropogenic aerosol.

Present-day global annual radiative effects for anthropogenic aerosol yield (1) a climate cooling of -1.0 W/m2 at the top of the atmosphere (TOA). (2) a surface net-flux reduction of -2.1 W/m2 and (by difference, 3) an atmospheric effect of a +1.1 W/m2. This atmospheric solar heating is almost entirely a direct effect. On a global basis, indirect effects (-0.65 W/m2) dominate direct effects (-0.35 W/m2) for the present-day climate response at the TOA, whereas the present-day surface radiative budget is stronger reduced by direct effects (-1.45 W/m2) than by indirect effects (-0.65 W/m2).

Natural aerosols are on average less absorbing and larger in size. However, their stronger solar TOA cooling efficiency is offset by a non-negligible infrared (IR) greenhouse warming efficiency. In the sum the global average annual direct forcing efficiencies (per unit AOD) for natural and anthropogenic aerosol are similar: -12 W/m2/AOD at all-sky conditions and -24 W/m2/AOD at clear-sky conditions.

The present-day direct TOA impact by all soot (BC) is globally averaged +0.55 W/m2. Between +0.25 to +0.45 W/m2 of that can be attributed to anthropogenic sources, depending on assumptions for the pre-industrial BC reference state. Similarly, the pre-industrial fine-mode reference uncertainty has a strong influence not just on the direct but even more on the indirect effect. Present-day aerosol TOA forcing is estimated to stay within the -0.7 to -1.6 W/m2 range (with the best estimate at -1.0 W/m2).

Calculations with scaled temporal changes to anthropogenic AOD from global modeling indicate that the global annual aerosol forcing has not changed much over the last decades, despite strong shifts in regional maxima for anthropogenic AOD. These regional shifts explain most solar insolation (brightening or dimming) trends that have been observed by ground-based radiation data.

## 1. Introduction

Atmospheric aerosol modulates the radiative energy budget directly (by the aerosol presence) and indirectly (by modifying the properties of clouds). Such impacts are of interest for climate change predictions, because part of today's atmospheric aerosol is anthropogenic. A quantification of aerosol impacts on global scales, however, is difficult. Tropospheric aerosol is highly variable in space and time and the needed pre-industrial reference for anthropogenic impacts is poorly defined. The determination of aerosol impacts requires two simulations: one with aerosol and one with less or no aerosol. Usually

complex 'bottom-up' simulations with global models are applied, in which emissions of different aerosol
sources are chemically and/or cloud processed, mixed, transported and removed. Further assumptions
to size and water uptake are needed to determine associated aerosol optical properties. These in turn
are needed to estimate aerosol radiative impacts in broadband radiative transfer applications. This path
involves many uncertainties. And many repeated simulations are generally needed to constrain natural
variability (mainly by clouds). These 'bottom-up' simulations are essential to account for feedbacks and
delayed or spatially detached responses in the climate system. Fortunately, for aerosol radiative impacts
feedbacks are secondary in strength (*Fiedler et al. 2019*). Thus, off-line radiative transfer applications to
determine an instantaneous impact via a dual call mode seem sufficient. Hereby, selected properties of
aerosol (for direct radiative impacts) and/or of (aerosol modified) clouds (for total / indirect radiative
impacts) are modified with respect to a reference simulation. And baseline properties for aerosol and its
environment are prescribed by monthly global maps linked to observational data. Aerosol properties are
prescribed by the MACv2 aerosol climatology (*Kinne, 2019*). MACv2 defines global (monthly, 1x1deg
gridded) maps for aerosol optical and radiative properties.  Aerosol component detail is derived in a
(reverse processing) 'top-down' approach to define the spectrally resolved aerosol single scattering
properties (needed for broadband radiative transfer). By prescribing optical properties for aerosol and
clouds (with strong links to observations) simulated aerosol radiative impacts are faster, more precise
and more direct than with 'bottom-up' approaches.  First the assumed MACv2 and environmental
properties are presented. Then the applied radiative transfer scheme is outlined. And finally radiative
impacts are presented for total and anthropogenic aerosol - also as a function of time.

## 2. MACv2 Aerosol Properties

The **M**ax Planck institute **A**erosol **C**limatology (MAC) in its second version (*Kinne, 2019*) defines
spectrally resolved monthly global fields for aerosol properties with global coverage (at a 1x1 degree
lat/lon spatial resolution). Annual averages of defining MACv2 aerosol properties are shown in Figure 1.

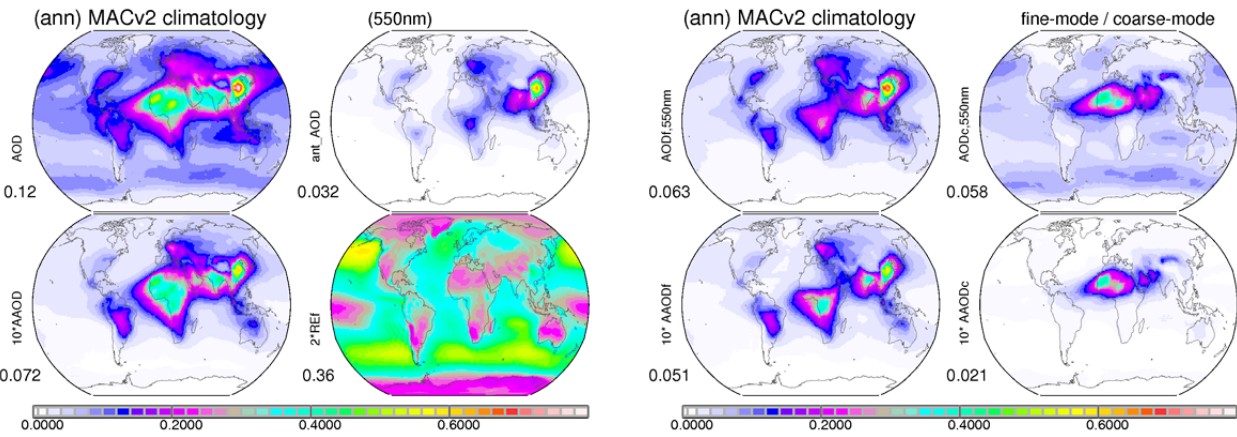

***Figure 1.*** *annual average maps of the MACv2 aerosol climatology. Global distributions are presented for*
*present day column properties of aerosol amount (AOD), absorption (AAOD\*10), anthropogenic AOD and*
*fine-mode effective radius (REf\*2) in um.  Also presented are the mid-visible AOD and AAOD split into*
*fine-mode (radii <0.5um) and coarse mode (radii >0.5um) contributions. Labels indicate global averages.*

1         These global fields are the result of a data merging process for mid-visible (at 550nm) properties

of aerosol optical depth (AOD, column amount) and absorption aerosol optical depth (AAOD, column
absorption). This merging was done separately for fine-mode (radii <0.5um) sizes (AODf, AAODf) and
coarse-mode (radii >0.5um) sizes (AODc, AAODc), as well as for the fine-mode effective radius (REf). In
the merging process regional distributions of monthly background maps are adjusted based on local
matches to trusted monthly statistics by ground based solar photometry observations. Background
maps in MACv2 are median properties of 14 different global models with detailed aerosol modules
(*Kinne et al., 2006*) in the framework of AeroCom phase 1 simulations. Applied photometry monthly
statistics are sun- and sky-samples by CIMEL instruments of the AERONET network (*Holben et al. 2001,*
*Dubovik et al, 2002*) and sun-samples of by-hand-operated MICROTOPS instruments of the Marine
Aerosol Network, MAN (*Smirnov et al., 2009*). The needed spectrally varying single scattering properties
for broadband radiative transfer simulations are set by local mixtures of spectrally predefined (via
refractive index and size) aerosol components. Local component mixture weights are chosen such that
their combined local AODf, AODc, AAODf, AAODc and REf values are consistent with MACv2 data.

15         In MACv2 on a global annual basis about 50% of the total AOD at 550nm (of 0.12) each is

contributed by fine-mode and coarse-mode sizes, whereas the total absorption AAOD at 550nm (of
0.0072) has at stronger fine-mode contributions, at 70%.  Still, the other 30% contributions by coarse-
mode sizes to absorption are significant and are associated with larger mineral dust sizes near sources.

19         Figure 1 also presents MACv2 estimates for the distribution of anthropogenic AOD at present-

day conditions (at the mid-visible wavelength of 550nm). Anthropogenic AOD in MACv2 allows only
contributions by smaller fine-mode (radii <0.5um) aerosol sizes, with major contributions from pollution
and fires. The anthropogenic AOD is determined by applying to the AODf map of MACv2 (local, monthly)
scaling factors [=(AODf,pd -AODf,pi) /AODf,pd)] based on AeroCom phase 2 simulations with CMIP5
present-day (pd) and pre-industrial (pi) emissions (*Lamarque et al, 2010*). Anthropogenic coarse-mode
(radii >0.5um) contributions (e.g. due to land-use change) are ignored, because  (1) its AOD is relatively
small (*see Figure E1 in Appendix E*), (2) its solar and IR forcing effects largely cancel each other (as shown
later) and (3) increases to aerosol concentrations via the coarse-mode  (for indirect effects) are minor.

28         For radiative transfer simulations three wavelength dependent aerosol radiative properties are

needed: aerosol optical depth (**AOD**), single scattering albedo (**SSA**) and asymmetry-factor (**ASY**).
Present-day global annual averages of these MACv2 aerosol radiative properties are summarized in
Table 1 for four selected wavelengths.  Corresponding seasonal maps in the mid-visible spectral region
(at 550nm) for total aerosol and anthropogenic aerosol radiative properties are presented in Figure 2.
*Table 1 global annual averages of radiative properties for present-day (tropospheric) MACv2 aerosol*

| λ(μm) | AOD | | | | SSA | | | ASY | | |
|---|---|---|---|---|---|---|---|---|---|---|
| | *total* | *coarse* | *fine* | *anthr\** | *total* | *coarse* | *fine* | *total* | *coarse* | *Fine* |
| .45 | **0.144** | 0.058 | 0.087 | 0.043 | **0.902** | 0.905 | 0.900 | **0.718** | 0.789 | 0.670 |
| .55 | **0.122** | 0.058 | 0.063 | 0.032 | **0.941** | 0.964 | 0.919 | **0.702** | 0.767 | 0.639 |
| 1.0 | **0.081** | 0.062 | 0.019 | 0.009 | **0.956** | 0.982 | 0.870 | **0.693** | 0.736 | 0.533 |
| 10 | **0.049** | 0.049 | | | **0.580** | 0.560 | | **0.605** | 0.605 | |

*\* anthropogenic SSA and ASY in MACv2 are those of the fine-mode*

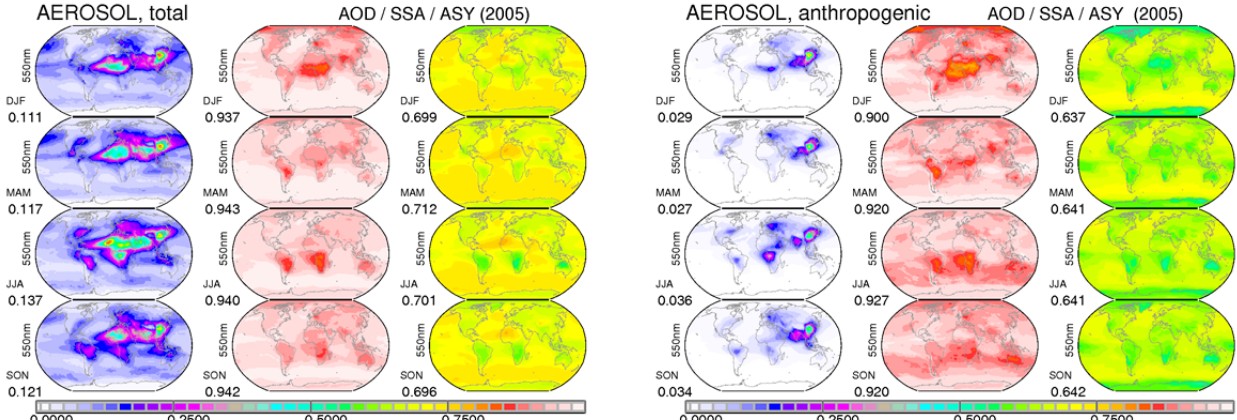

**Figure 2.** *seasonal maps of MACv2 associated present-day mid-visible (550nm) radiative (or single*
*scattering) properties (AOD, SSA, ASY) for total aerosol (left block) and for anthropogenic aerosol (right*
*block). Global averages for each season are indicated below the labels.*

7          In simulations of aerosol radiative effects the relative aerosol altitude distribution with respect
to clouds is important. In MACv2 an AOD fractional scaling is applied based on multi-year simulations
with the ECHAM-HAM global model (*Zhang et al., 2012*). This scaling is done independently for fine-
mode AOD (AODf) and coarse-mode AOD (AODc). With the AOD scaling by size modes, a vertical scaling
for SSA and ASY seems secondary and is ignored. For anthropogenic aerosol only the fine-mode vertical
scaling is relevant, because anthropogenic aerosol in MACv2 has only contributions by fine-mode size
aerosols.   AOD, AODc and AODf distributions with four altitude regions are illustrated in Figure 3.

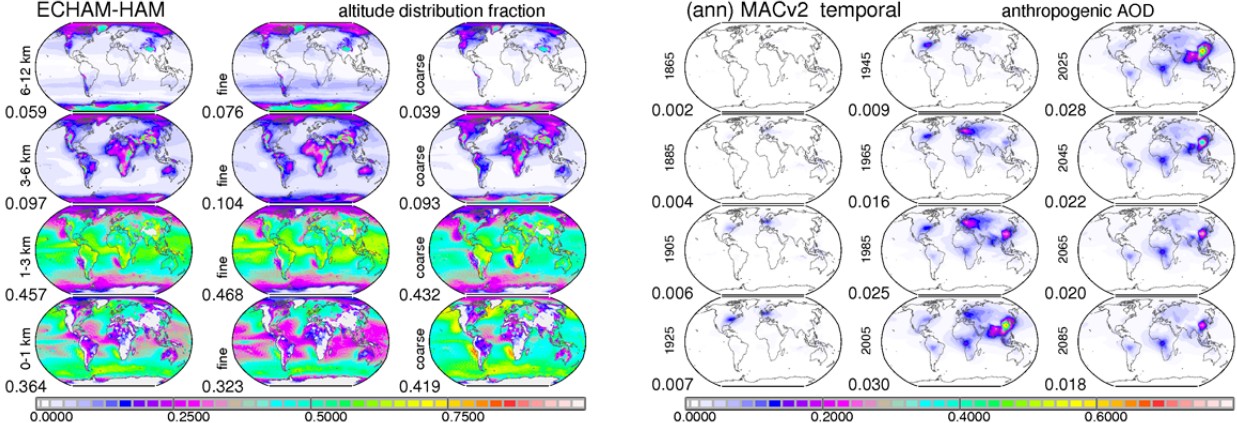

**Figure 3** *annual maps for fractional contributions of total, fine-mode and coarse mode AOD (left block)*
*for four altitude regimes (0-1, 1-3, 3-6, >6km above sea-level) and for anthropogenic AOD (550nm) for*
*different years from 1865 to 2085 (right block). Values below the label indicate global averages.*

22          Figure 3 also presents time-slices (1865, 1885, 1905, 1925, 1945, 1965, 1985, 2005, 2025, 2045,
2065, 2085) for the anthropogenic AOD (antAOD). Temporal changes in anthropogenic AOD are based
on local scaling factors (antAOD,year/antAOD,pd) by ECHAM-HAM (*Stier et al., 2006, Zhang et al. 2012*)
simulations with temporally changing emissions according to historic data and future projections.
3       The resulting anthropogenic AOD maps in Figure 3 display globally highly uneven distributions.
From the late 19[th] century into the 20[th] century anthropogenic AOD steadily increased over Europe and
the eastern US. Only since the middle of the 20[th] century also other continental regions started to
display anthropogenic AOD. Particularly strong were increases over SE Asia. Over the last three decades
(since 1985) the anthropogenic AOD maxima declined over Europe and the eastern US and maxima over
SE Asia and more recently over S Asia began to dominate. With the shift in regional maxima the present-
day global average anthropogenic AOD appears to have reached a plateau, which is probably a
maximum, as future emission scenarios suggest a decline in global anthropogenic AOD.

## 3. Environmental Properties

14       The environmental properties in radiative transfer simulations are represented by monthly
averages to describe seasonal variations in solar insolation, atmospheric state and properties of clouds
and the surface. Monthly mean of the sun's latitudinal position define variations in TOA solar irradiance.
Surface temperatures (*Hansen et al., 2010*) via multi-annual monthly local averages help in choices for
standard atmospheric profiles (*Andersen et al., 1986; Mc Clatchey et al, 1972*). Local combinations of
these standard profiles define local monthly atmospheric state and trace-gas concentrations. For clouds
multi-annual monthly average ISSCP data (*Rossow et al., 1993*) define scene optical depth and the cloud
cover at low (>680hPa), mid and high (<440hPa) altitudes. For surface properties an IR emittance of 0.96
is assumed. Surface solar albedo data over land are based on MODIS sensor data for the visible (UV/VIS)
and near-infrared (n-IR) spectral regions (*Schaaf et al., 2002*). Over oceans, solar albedo data account for
a solar elevation dependence (*Taylor et al., 1996*). Seasonal means of the applied VIS and nIR solar
surface albedo and the ISCCP cloud cover by altitude are presented in Figure 4.

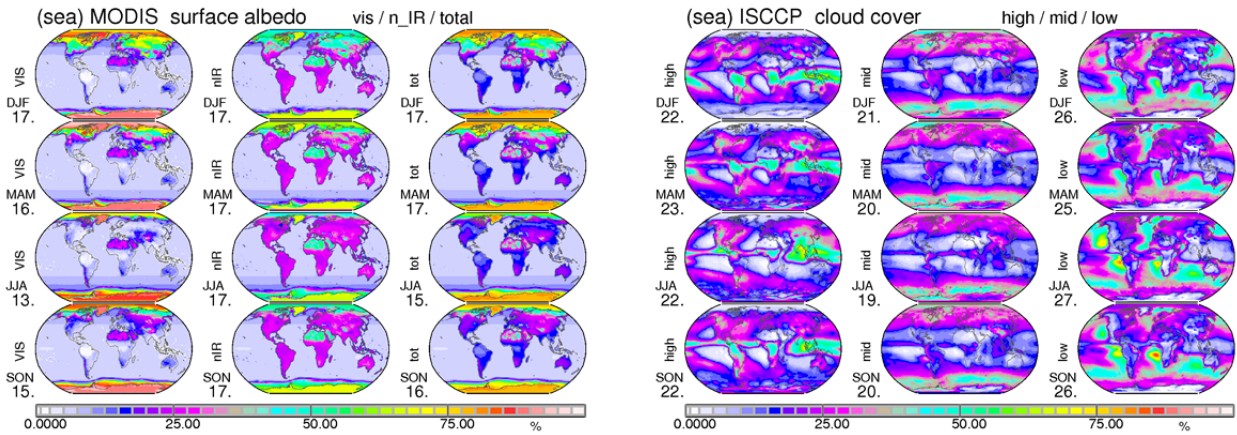

*Figure 4 Seasonal averages for applied environmental properties to (1) solar surface albedo (left block)*
*for the UV/VIS, the near-IR and the total spectral range based on MODIS satellite sensor data and (2)*
*ISCCP based cloud cover (right block) for high (<440hPa), mid (440-680hPa) and low (>680hPa) altitudes.*

# 4. Radiative Transfer Method

The atmospheric radiative transfer simulations apply a two stream radiative transfer scheme (*Meador and Weaver, 1980*). Spectral variability is captured by eight solar and twelve infrared bands with a total of 120 exponential terms to represent atmospheric trace-gas absorption. Vertical variability of the atmosphere is represented by 20 plane parallel homogenous layers and atmospheric state and trace-gas properties are defined via standard atmospheres (*Mc Clatchey et al, 1972*). Independent simulations at each (of the 64800) grid locations are performed for each month with monthly averages. Hereby, daily average solar radiative effects are based on weighted individual simulations at nine different solar zenith angles. Simulations with (ISCCP) clouds always involve simulations at all eight possible permutations for high-, mid- and low- altitude cloud combinations, assuming random overlap and for satellite view corrected local cloud cover data in the three altitude regimes.

Radiative impacts in the atmosphere are defined by differences of two simulations between a modified and a standard configuration. These 'dual-call' radiative transfer applications investigate

- **extra aerosol presence** (*direct effect - with a focus on individual aerosol components*),
- **reduced water droplet radii** *due to extra aerosol (first indirect effect) and*
- *combination of* **both effects**.

The resulting changes to broadband solar and infrared radiative net-fluxes are particularly relevant

- **at the top of the atmosphere** (*TOA) - for the overall climate impact,*
- **at the surface** *- for impacts on surface processes and*
- **in the atmosphere** (*by TOA minus surface impact differences) - for impact on dynamics.*

Dual-call radiative transfer cannot consider climate feedbacks. Long-term Earth System Model (ESM) simulations with a fixed sea-surface temperature (SST), however, indicated that atmospheric feedbacks (*Fiedler et al, 2017, Fiedler et al, 2019 - there referred to as 'rapid adjustments'*) modulate the aerosol radiative impacts at most on the order or 10%. Thus by ignoring radiative forcing feedbacks, no major extra errors are introduced. It also should be pointed out that a dual-call scheme offers more precise answers, as internal (cloud) variability of independent ESM simulations is avoided (*Fiedler et al, 2019*).

# 5. Direct Aerosol Effects

Aerosol direct radiative effects are changes to the atmospheric energy distribution from the aerosol (or the extra aerosol) presence. These effects are quantified by the difference of one simulation with all aerosols and one simulation with no or less aerosol. The anthropogenic aerosol impact, for instance is defined by the difference of radiative effects of a simulation with present day (pd) and pre-industrial (pi) conditions. Only coarse mode aerosol sizes are large enough for infrared radiative effects. As only smaller fine-mode aerosols contribute to the anthropogenic AOD in MACv2, only solar impacts for anthropogenic aerosol need to be investigated. Annual average maps of present day direct aerosol radiative effects are summarized in Figure 5 for cloud-free conditions (clear-sky, 'clr') and with tropospheric clouds (all-sky, 'all') - for both total and anthropogenic aerosol at TOA and surface.

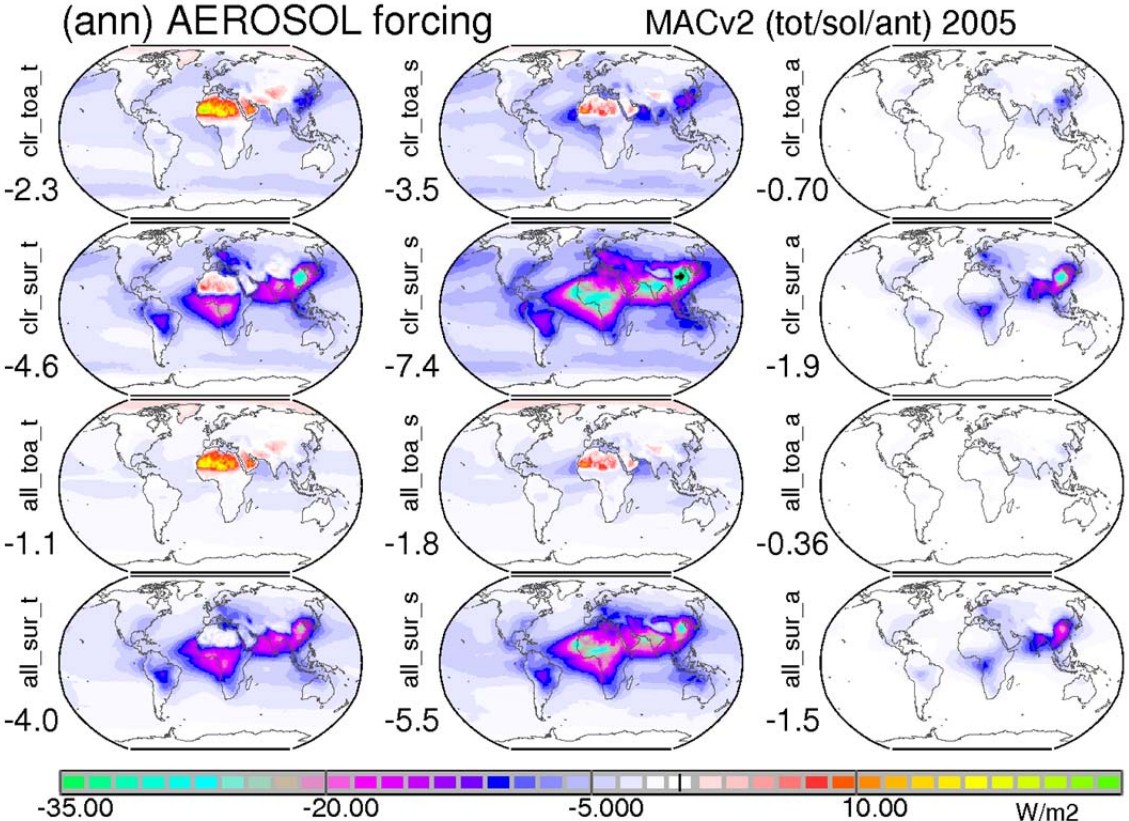

Figure 5 *Annual maps for direct radiative effects of present-day total aerosol (left column), its solar*
*effect (center column) and of present-day anthropogenic aerosol. Clear-sky effects are presented in the*
*upper two rows: at the TOA (row 1) and at the surface (row2). All-sky effects (with ISCCP clouds) are*
*presented in the lower two rows: at the TOA (row 3) and at the surface (row 4). Note, values for the*
*anthropogenic direct forcing (column 3, row3) are relatively small. More detail is given in Figure 6. Blue*
*colors indicate radiative net-flux losses (or a 'cooling'), and red colors indicate radiative net-flux gains (or*
*a 'warming'). Values below the labels indicate global averages.*

11        For the aerosol associated net ('down' minus 'up') –flux changes in Figure 5, negative values
(blue to purple to light-blue) indicate regions with tendency to 'cool', whereas positive values (red to
yellow) display regions that experience a tendency to 'warm'. Global averages for all maps of Figure 5
are negative. Thus present day total (left column in Figure 5) and anthropogenic aerosol (right column in
Figure 5) on average cool.  Hereby global averages of aerosol radiative effects are more negative at the
surface than at the TOA, because some aerosol types also absorb in the atmosphere. Radiative effects
are usually more negative at clear-sky than at all-sky conditions, because reflecting clouds above aerosol
prevent aerosol interactions with solar radiation. This 'clear-sky' minus 'all-sky' difference is much larger
at TOA than at the surface, because elevated absorbing aerosols above clouds dim the reflection of
clouds to space for a relative warming. All, direct radiative effects spatially unevenly distributed.
21        For total aerosol (in the left column of Figure 5) the negative forcing in most regions is partially
offset by regional warming over the N. Africa and Arabia, especially for TOA effects. To understand this
response, it should be recalled that the dominating mineral dust aerosol particles in those regions are
relatively large, elevated (off the ground) and absorbing. For solar only effects (in the center column of
Figure 5) absorbing mineral dust aerosol over the bright desert surface causes a relative TOA warming
by dimming the surface solar reflection to space. And there are also infrared effect, because mineral
dust sizes are relative large. As mineral dust (unlike sea-salt) is often elevated and strongly absorbing,
mineral dust also contributes with a significant greenhouse effect. This greenhouse effect adds to the
solar warming at TOA and offsets with infrared re-radiation the surface the solar cooling at the surface.
On a global average basis ca 35% of the solar energy losses by present-day aerosol are compensated by
IR energy gains. In other regions, where smaller fine-mode sizes dominate, aerosol cools at TOA (unless
surfaces are bright, as with snow or lower clouds) and always cools at the surface. Hereby the local
surface cooling is much more negative than the local TOA response, when fine-mode aerosol is strongly
absorbing, as over biomass burning regions over Africa and S. America and pollution of SE Asia.
12       For anthropogenic aerosol MACv2 only allows sub-micrometer size contributions. Thus only
solar radiative impacts matter. For present-day anthropogenic aerosol (for climate change relevant)
aerosol direct forcing (the direct radiative effects by anthropogenic aerosol at all-sky conditions at TOA)
yields a global cooling at -0.36W/m2. This direct forcing signal is on average more than a magnitude
smaller than detectable solar radiation reductions at the surface or satellite detectable planetary albedo
increases over oceans for total aerosol.  More details on seasonal variability for the present-day direct
forcing for anthropogenic aerosol (at an adjusted finer scale) are presented in Figure 6.

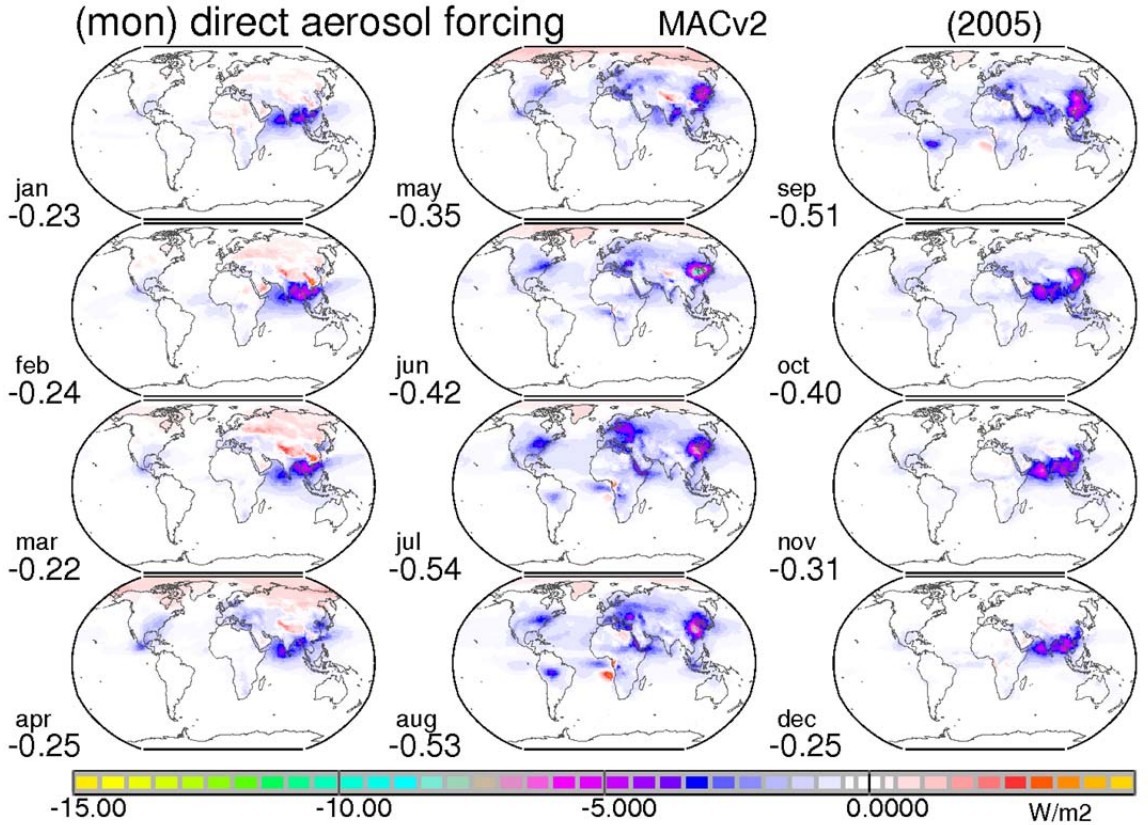

*Figure 6* *Monthly maps for present-day aerosol direct forcing with MACv2 aerosol properties*
On a global average basis the present-day aerosol direct forcing causes a cooling during the
entire year. The TOA cooling is strongest from Jul to Sep mainly due to more sun-hours and higher AOD
values at northern mid-latitudes. Higher AOD are in part caused by increased water uptake. The TOA
cooling is weakest from Dec to April also due to regional warming over snow cover in the northern
hemisphere (e.g. Asia, Arctic). The regional warming over the SE Atlantic in Aug-Sep is due to lower
cloud reflection dimming by elevated biomass burning aerosol. The spatial distribution also illustrates
the location of forcing maxima: May-Oct over E. Asia, Jun-Aug over EU and E. US, Sep to Apr over S. Asia.
More details on present-day direct radiative effects are provided by investigating contributions by the
(sub-micrometer) fine-mode sizes in Figure 7 and by (super-micrometer) coarse-mode size in Figure 8.

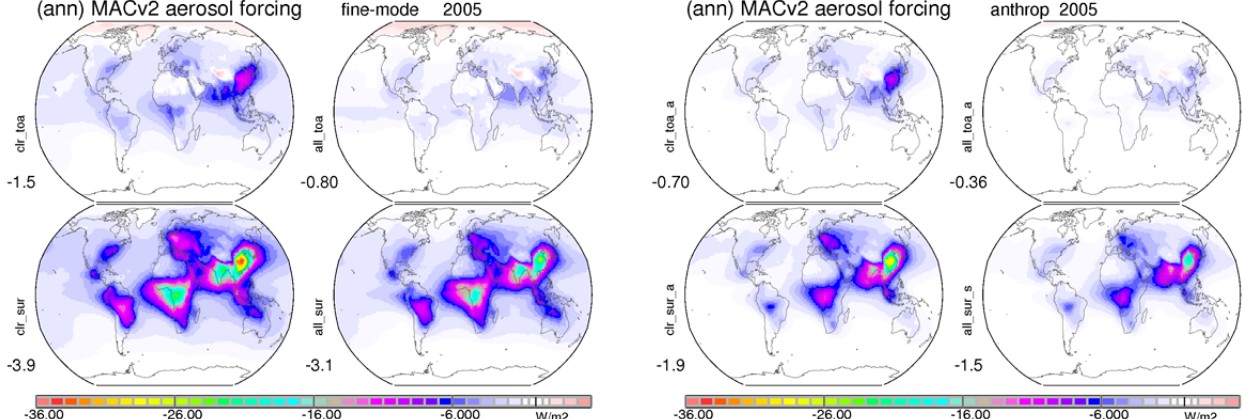

***Figure 7*** *Annual maps for present-day fine-mode (left block) and anthropogenic effects (right block) in*
*MACv2. Aerosol radiative effects at TOA (top row) and at surface (bottom row) are presented for clear-*
*sky conditions (left column in each block) and all-sky conditions (right column in each block). Blue to red*
*colors indicate a cooling and pink colors weak warming. Values below the labels show global averages.*

Fine-mode and coarse mode radiative effects differ is their regional contributions. On average
fine-mode contributions to TOA cooling are stronger than coarse-mode contributions, despite a stronger
solar fine-mode absorption (in the atmosphere). This is explained by regional greenhouse contributions
of coarse-mode aerosols (over northern Africa, Arabia and Asia). The partially offsetting local solar and
infrared contributions to coarse mode direct radiative effects at TOA and surface are shown in Figure 8.
Also compared in Figure 7 and 8 are size-mode contributions by anthropogenic aerosol to
present-day direct radiative effects. For the fine-mode on average 50% of present-day fine-mode
radiative effects can be considered anthropogenic (with the other 50% already contributing at pre-
industrial times). Anthropogenic coarse-mode mineral dust (due to land use change) contribute at about
10% to the dust AOD and to radiative effects at the surface. At the TOA the coarse-mode radiative effect
contributions are only at 5% at TOA due to partially offsetting solar and infrared effects.
Based on a comparison of present-day direct aerosol forcing contributions of -0.36 W/m2 for
the fine-mode and +0.02 W/m2 for the coarse mode, the more than a magnitude smaller coarse mode
contributions to the anthropogenic AOD are ignored in MACv2.

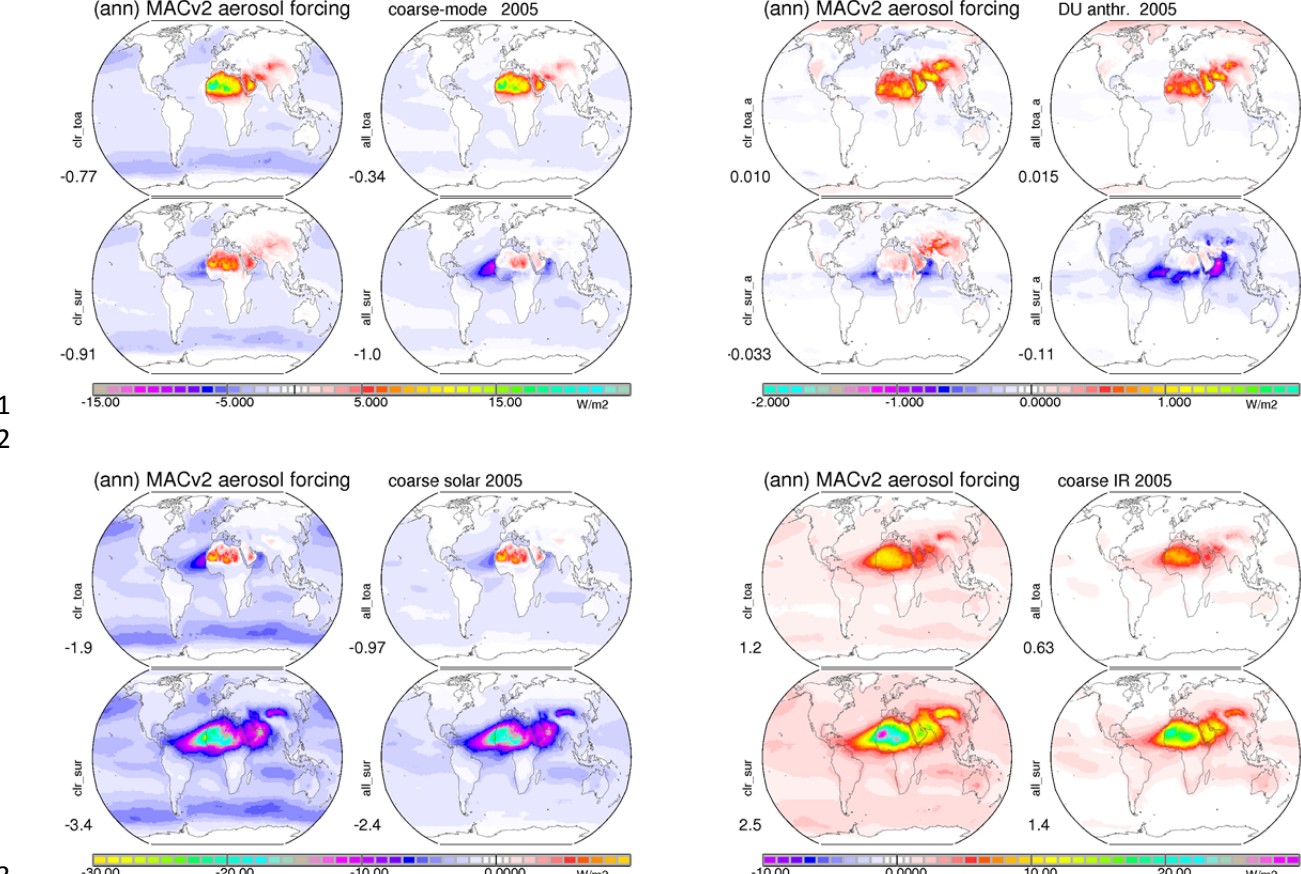

**Figure 8** *Annual maps for the coarse-mode aerosol direct radiative effects at TOA (top) and surface (bottom) for clear-sky (left) and all-sky (right) conditions. Coarse-mode effects (top, left block) are compared to potential anthropogenic contributions (top, right block, note its 10 times smaller color scale range). Blue to purple colors indicate a cooling and red to yellow colors a warming. In addition, solar (bottom, left block) and IR contributions (bottom, right block) to the coarse-mode aerosol direct radiative effects are presented. Note the different scales. Values below the labels indicate global averages.*

With the AOD attributions in MACv2 into radiatively defined components via their mid-visible absorption properties (*Kinne, 2019*) and via the anthropogenic definition in MACv2, present-day direct aerosol radiative effects of components and of their anthropogenic contributions could be determined. For smaller fine-mode aerosol, direct radiative effects are contributed by non-absorbing aerosol represented by sulfate (SU), by weakly absorbing organic matter (OC) and by strongly absorbing soot (BC). Also the combined carbon (CA = OC+BC) effect is examined to illustrate the BC impact, when co-emitters of soot are included. For coarse-mode aerosol, direct radiative effects are contributed by non-absorbing sea-salt (SS) and weakly absorbing mineral dust (DU). Hereby stronger coarse-mode absorption translates into larger mineral dust sizes. Annual maps for present-day component TOA radiative effects at clear-sky and all-sky conditions (forcing) are shown in Figure 9. More details on component direct radiative effects are presented in Appendix E.

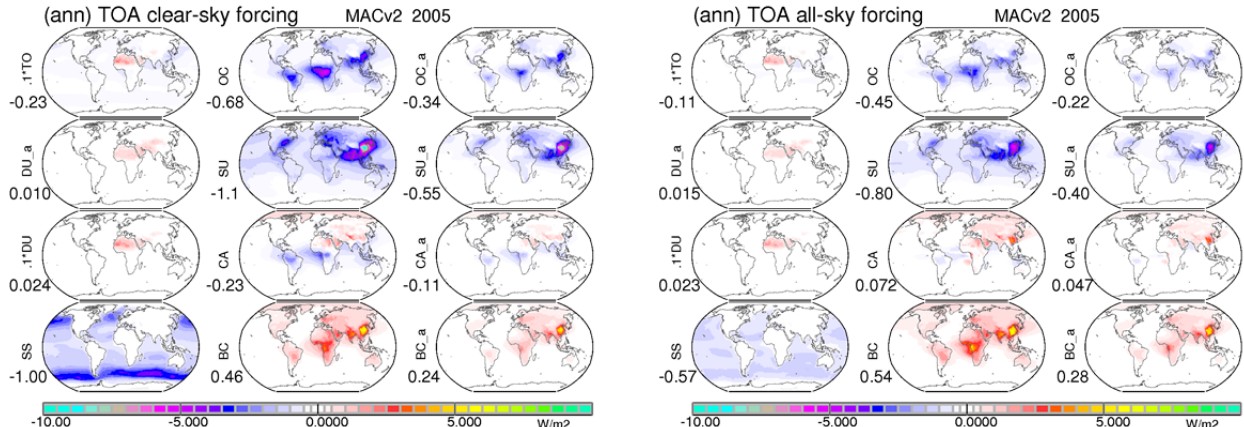

**Figure 9** *Annual maps for today's direct forcing at TOA at clear-sky (left block) and at all-sky conditions*
*(right block) by total aerosol (TO, divided by 10 to fit common scale) and by coarse-mode components of*
*anthropogenic dust (DU_a), dust (DU, divided by 10 to fit common scale) and sea-salt (SS) in the left*
*column, by fine-mode components of non-absorbing sulfate (SU), of weakly absorbing organic matter*
*(OC), of strong absorbing soot (BC) and of the combined carbon (CA=OC+BC) component in the center*
*column and by the anthropogenic contributions of all fine-mode components in the right column. Global*
*averages for the annual direct aerosol component TOA effects are presented below the labels.*

The radiative TOA direct radiative effects of individual components are quite diverse. Climate
cooling by sulfate (SU), sea-salt (SS), organic matter (OC) and dust (DU-o) over oceans is on average
stronger than climate warming by soot (BC) and dust (DU-c) over continents. As MACv2 only considers
anthropogenic contributions to the fine-mode, investigations of anthropogenic impacts can be reduced
to cooling by non-absorbing SU and weakly absorbing OC and warming by strongly absorbing BC. The
combined carbon (CA) approximates the impacts of co-emitted gases that quickly condensate on
existing particles (including soot) to increase scattering on existing particles (also in the context of an
upper limit near 2 for a soot absorption increase via a scattering shell). The combined CA forcing is near
neutral (+0.05 W/m2). This suggests that for approximate estimates for the global average direct forcing
only changes in fine-mode non-absorbing AOD since pre-industrial times should be considered.

The CA near neutral response, however, only applies, if the BC anthropogenic fraction is that of
the fine-mode AOD fraction. With a higher soot (BC) anthropogenic fraction (especially near pollution
regions), as assumed in the BC assessment (*Bond et al., 2013*), the present-day BC forcing increases to
+0.44 W/m2 as shown in Appendix E. This added BC warming (with an alternate present-day CA
warming near +0.20 W/m2) would reduce the present-day total direct aerosol forcing to -0.20 W/m2
climate cooling. However, as this higher alternate fine-mode anthropogenic fraction for BC is also linked
to a year 1750 reference (*Dentener et al., 2006*), such a large BC anthropogenic fraction seems unlikely
for the year 1850. Still, given the uncertainly to the BC pre-industrial state an overall present-day
aerosol direct forcing of -0.25 W/m2 (down from -0.36 W/m2) cannot be easily ruled out.

The consideration of co-emitted gases soot (BC) removal processes may not have the often
attributed potential for short term climate warming mitigations. Thus, the singled out present-day
warming +0.55 W/m2 for all soot (BC) with estimated anthropogenic contributions between +0.25 to
+0.40 W/m2 is deceiving as co-emitters also have to be considered in removal processes.

1        Annual averages of MACv2 aerosol associated direct radiative effects for present-day
atmospheric conditions are summarized in Tables 2 and 3. Table 2 compares radiative effects at TOA,
atmosphere and surface for all aerosol, for fine-mode aerosol, for coarse-mode aerosol and for
individual aerosol components. Table 3 compares solar and infrared contributions of components with
non-negligible infrared impacts. Global maps for component radiative effects of Table 2 are presented in
Appendix D.
***Table 2*** *annual average MACv2 climatology associated aerosol radiative effects for today's tropospheric*
*aerosol at the top of the atmosphere (TOA), at the surface and (by difference) for the atmosphere. Aside*
*for total aerosol (in row 1) also effects of components and if applicable their anthropogenic contributions*
*are indicated. Considered fine-mode components are sulfate (SU), organic matter (OC), soot (BC) and the*
*combined carbon (OC+BC). Considered coarse-mode components are sea-salt (SS) and dust (DU).*

| direct effect (W/m2) | TOA | | | | ATMOSPHERE | | | | SURFACE | | | |
|---|---|---|---|---|---|---|---|---|---|---|---|---|
| | *total* | | *anthr* | | *total* | | *anthr* | | *total* | | *anthr* | |
| | **all** | **clear** | **all** | **clear** | **all** | **clear** | **all** | **clear** | **all** | **clear** | **all** | **clear** |
| total | **-1.1** | **-2.3** | | | **+2.9** | **+2.3** | | | **-4.0** | **-4.6** | | |
| fine | -0.80 | -1.5 | **-0.36** | **-0.70** | +2.3 | +2.4 | **+1.1** | **+1.2** | -3.1 | -3.9 | **-1.5** | **-1.9** |
| - SU | -0.83 | -1.2 | -0.41 | -0.58 | +0.01 | +0.02 | +0.00 | +0.01 | -0.84 | -1.2 | -0.41 | -0.59 |
| - CA | +0.08 | -0.23 | +0.05 | -0.10 | +2.0 | +2.2 | +1.0 | +1.2 | -2.1 | -2.5 | -1.0 | -1.3 |
| - OC | -0.45 | -0.68 | -0.22 | -0.34 | +0.49 | +0.52 | +0.23 | +0.24 | -0.94 | -1.2 | -0.45 | -0.58 |
| - BC | +0.55 | +0.46 | +0.28 | +0.24 | +1.7 | +1.8 | +0.89 | +0.94 | -1.2 | -1.4 | -0.61 | -0.70 |
| - BC* | | | +0.44 | +0.37 | | | +1.4 | +1.5 | | | -0.97 | -1.1 |
| coarse | -0.34 | -0.77 | | | +0.66 | +0.14 | | | -1.00 | -0.91 | | |
| - SS | -0.57 | -1.00 | | | +0.01 | -0.10 | | | -0.58 | -0.90 | | |
| - DU | +0.23 | +0.24 | +.015 | +.010 | +0.68 | +0.25 | +0.12 | +0.13 | -0.45 | -0.01 | -0.11 | -.033 |

*\*based on AeroCom 1 - ref year 1750 (and not AeroCom 2 - ref year 1850) anthropogenic BC fine-mode fractions*
***Table 3*** *annual average MACv2 climatology associated aerosol radiative effects for today's tropospheric*
*aerosol at the top of the atmosphere (TOA) and at the surface, by separating solar and IR contributions.*
*Aside for the total aerosol (in row1) also effects are presented for components with an IR impact, such as*
*seasalt (SS), dust (DU) and anthropogenic dust (aDU).*

| direct effect (W/m2) | TOA | | | | | | SURFACE | | | | | |
|---|---|---|---|---|---|---|---|---|---|---|---|---|
| | *all-sky* | | | *clear-sky* | | | *all-sky* | | | *clear-sky* | | |
| | | solar | IR | | solar | IR | | solar | IR | | solar | IR |
| total | -1.1 | -1.8 | +0.66 | -2.3 | -3.5 | +1.2 | -4.0 | -5.5 | +1.5 | -4.6 | -7.4 | +2.8 |
| coarse | -0.34 | -0.97 | +0.63 | -0.77 | -1.9 | +1.2 | -1.00 | -2.4 | +1.4 | -0.91 | -3.4 | +2.5 |
| - SS | -0.57 | -0.72 | +0.16 | -1.00 | -1.4 | +0.39 | -0.58 | -0.78 | +0.20 | -0.90 | -1.4 | +0.55 |
| - DU | +0.23 | -0.24 | +0.47 | +0.24 | -0.53 | +0.77 | -0.45 | -1.6 | +1.2 | -0.01 | -1.9 | +1.9 |
| - aDU | +.015 | -0.07 | +0.08 | +0.01 | -0.12 | +0.13 | -0.11 | -0.29 | +0.18 | -0.03 | -0.37 | +0.34 |

# 6. Direct Forcing Efficiencies

Forcing efficiencies offer a shortcut to radiative effects without actually performing radiative transfer simulations. Radiative forcing modulations by local monthly environmental properties (such as surface albedo, solar insolation and even clouds) are already included. For instance, a satellite retrieved AOD value or an anthropogenic AOD enhancement is then quickly associated with a radiative effect simply by multiplying the AOD value with the appropriate forcing efficiency. In such applications even information on the likely aerosol composition is included. Applying environmental assumptions and MACv2 aerosol properties in (radiative transfer) off-line simulations, aerosol direct forcing efficiencies are presented with respect to the mid-visible AOD at 550nm (as this particular AOD value in frequently used in conjunction with satellite retrievals and global modeling). Annual maps for forcing efficiencies per unit AOD for total and anthropogenic aerosol (corresponding to the radiative effects of Figure 5) are presented in Figure 10. Seasonal variations of global averages are small, as shown in Appendix D.

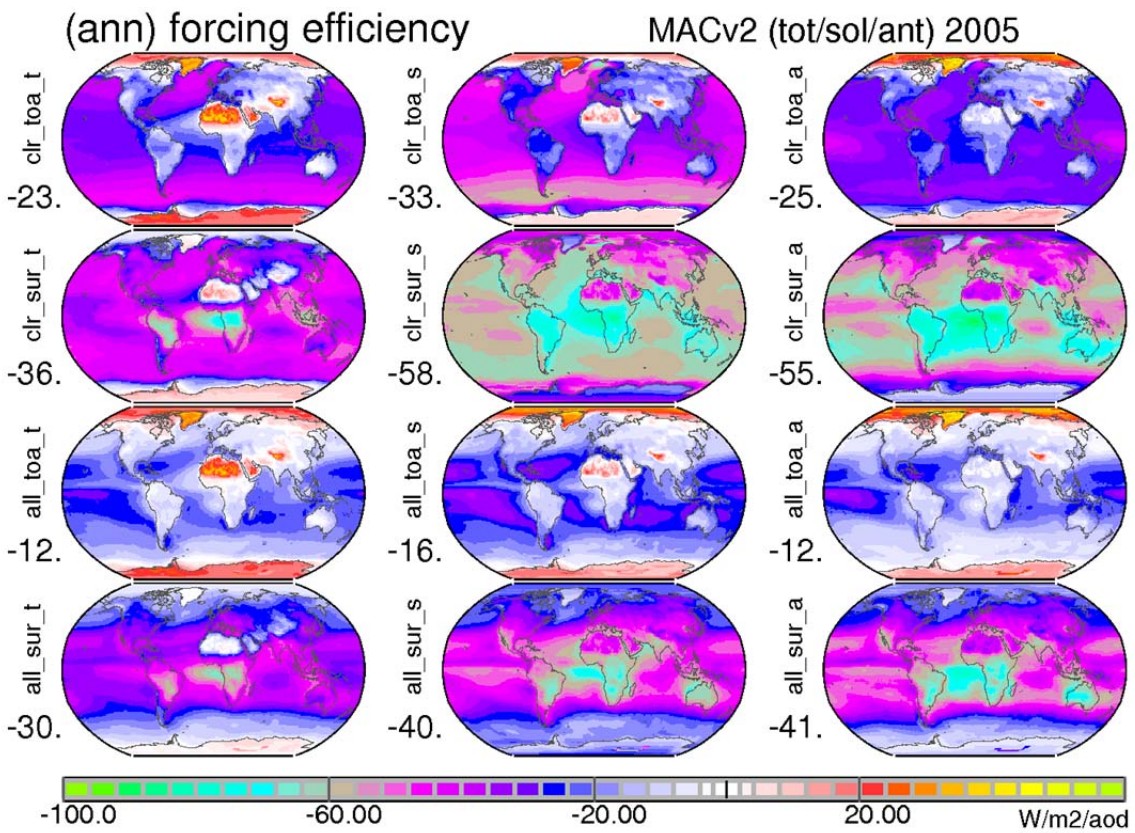

***Figure 10*** *Annual maps for aerosol direct radiative effect efficiencies (per unit AOD) of present-day all aerosol (left column), its solar effect only (center column) and of present day anthropogenic aerosol (right column). All maps correspond to direct radiative effects of Figure 5: Clear-sky efficiencies are presented in the upper rows: at TOA (row 1) and at surface (row2). All-sky efficiencies (with ISCCP clouds) are presented in the lower rows: at TOA (row 3) and at surface (row 4). Values indicate global averages.*

At TOA, global average direct aerosol radiative efficiencies (per unit AOD) are at -24 W/m2/AOD
for cloud-free ('clear-sky') conditions and at -12 W/m2/AOD for more realistic conditions with clouds
('all-sky'). These global forcing efficiencies are almost identical for total and anthropogenic aerosol. This
is a coincidence. The on average 30% stronger solar TOA cooling efficiencies for total aerosol are offset
by a coarse-mode associated IR warming efficiencies - mainly by elevated mineral dust. Spatially,
however, there are large differences associated with aerosol properties (e.g. absorption and size) and
background reflectance data (e.g. surface albedo, lower altitude cloud cover).
At the surface global average only solar direct aerosol radiative efficiencies (per unit AOD) agree
for total and anthropogenic aerosol at -56 W/m2/AOD for clear-sky and at -41 W/m2/AOD for all-sky
conditions. For combined (solar and IR) direct forcing efficiencies those for total aerosol are ca 30% less
negative, due to positive infrared re-radiation ('greenhouse effect') contributions.
With the AOD attributions in MACv2 into radiatively pre-defined aerosol types (*Kinne, 2019*) also
TOA efficiencies for individual aerosol components are determined. TOA forcing efficiencies for the SU,
OC, BC, DU and SS components are presented for clear-sky and all-sky conditions in Figure 11.

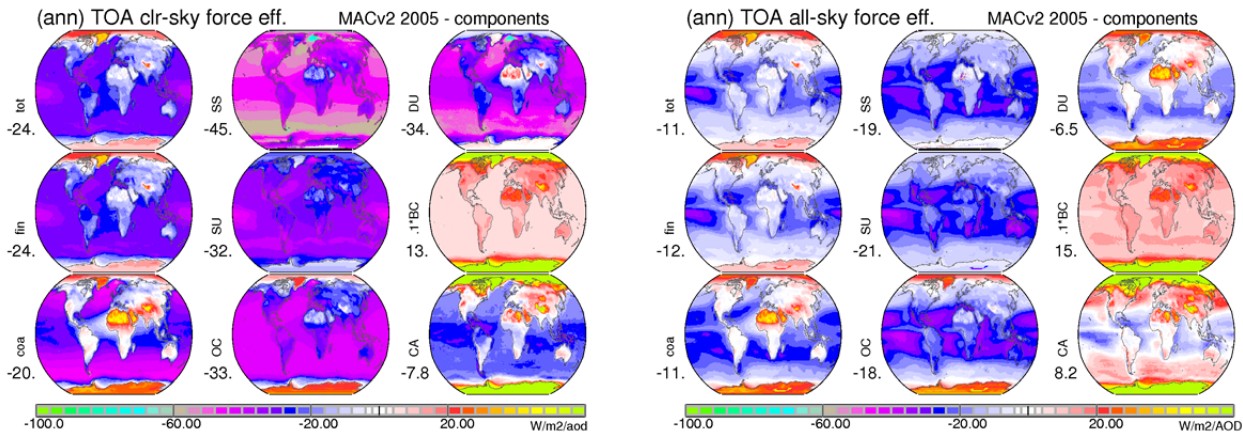

***Figure 11*** *Annual maps for present-day direct component forcing efficiencies (per unit AOD) at TOA at*
*clear-sky (left block) and at all-sky conditions (right block) for total, coarse mode and fine-mode aerosols*
*(left column), for mainly scattering components (center column) of sea-salt (SS), fine-mode (SU), organic*
*matter (OC) (center column) and components with absorption (right column) of dust (DU), soot (BC) and*
*for the combined carbon (CA=OC+BC). The large positive forcing efficiencies for BC (130 W/m2/AOD for*
*clear-sky and 150 W/m2/AOD for all-sky) are divided by 10 to fit the common color scale. Global*
*averages are presented below the labels.*
Direct radiative TOA forcing efficiencies for scattering fine-mode (SU) and coarse-mode (SS)
components are strongly negative and usually more negative over the darker background of oceans.
This also applies for weakly absorbing organic matter (OC) - except over (brighter) polar regions. The
efficiencies are at least -30 W/m2/AOD for clear-sky and near -20 W/m2/AOD for all-sky conditions.
Direct radiative TOA forcing efficiencies for coarse-mode absorbing dust (DU) are also much
more negative at clear-sky conditions compared to all-sky conditions. The clear-sky forcing efficiencies

over oceanic regions are almost as negative as for seasalt, but less negative over continents and continental outflow regions and even slightly positive near source regions. The all-sky forcing efficiency for dust displays only weak negative forcing efficiencies over oceans and usually positive forcing efficiencies over continents. Over major dust source regions (e.g. N. Africa and Arabia) all-sky forcing efficiencies for mineral dust (DU) are particular strong at +40 W/m2/AOD.

Direct radiative TOA forcing efficiencies for strongly absorbing soot (BC) are even more positive (than the all-sky forcing for dust near sources) and high everywhere over the globe, with maxima over (bright) polar surfaces. Unlike for other components BC forcing efficiencies near +150 W/m2/AOD are larger at all-sky conditions than at clear-sky conditions (with +120 W/m2/AOD). This is explained by BC aerosol dimming of solar reflection from lower altitude clouds.

Direct radiative TOA forcing efficiencies for combined carbon (BC+OC) represents the presence of mainly scattering BC co-emitters. Now, the high BC efficiencies are reduced to -10W/m2/AOD at clear-sky and to +10W/m2/AOD at all-sky conditions. Thus, by ignoring scattering co-emitters, climate warming potential of BC is overrated.

# 7. Indirect Aerosol Effects

Extra atmospheric aerosol loads (as from anthropogenic sources) modulate the atmospheric energy distribution not only directly and but also indirectly. Indirect effects are contributed through aerosol imposed changes to other atmospheric properties, most importantly to properties of clouds.  An important aspect is that added aerosols (relatively numerous from anthropogenic fine-mode sources) increase the concentrations of those aerosols that can serve as cloud condensation nuclei (CCN). With more available CCN at a condensation event, the available supersaturated water vapor is distributed onto more aerosol nuclei, so that the resulting water cloud droplets are more numerous and smaller in size - assuming that no changes to the cloud liquid water content (LWC) occur. With smaller drop sizes the solar reflection of a water cloud and along with it the planetary albedo increases (*Twomey, 1974*) for an added climate cooling. Examples are so-called 'ship tracks', when satellite sensors detect increases in planetary albedo of low altitude clouds above the path of polluting ships. But there are further impacts associated with smaller droplets affecting both cloud cover and cloud water content. The mixing with dryer air at cloud boundaries reduces the cloud lifetime (especially if the cloud cover is low), whereas the delay in the onset of precipitation extends the cloud lifetime (especially if cloud cover is high). In addition, there are potential but less investigated aerosol impacts involving mixed-phase and ice clouds.

The strongest observational evidences for aerosol indirect effects involve low altitude water clouds. In contrast, likely aerosol impacts are small for mixed phase clouds (*Christensen et al. 2016*) and for ice-clouds model simulation cannot even agree on the sign of overall impacts (*Penner et al., 2018*). With a focus on aerosol modifications to lower altitude water clouds, there remains the question, if changes to cloud lifetime and/or cloud cover matter in comparison reductions in cloud droplet sizes (the Twomey-effect). A recent satellite retrieval analysis involving a large sulfate aerosol anomaly over the northern Atlantic (*Mallavelle et al., 2017*) confirms the dominance of the Twomey-effect. It was found that sharply increased sulfate aerosol concentrations strongly reduced the cloud droplet sizes. However no significant changes to the cloud liquid water content were retrieved.

It seems straightforward to convert MACv2 suggested CCN increases due to anthropogenic AOD
in cloud droplet number concentration (CDNC) increases (and subsequently in associated cloud droplet
size reductions, assuming that the cloud water content did not change). Unfortunately, MACv2 based
CCN estimates (derived from fine-mode composition and fine-mode extinction, *Kinne 2019*) depend
strongly on assumptions for the supersaturation (or vertical winds at cloud base), which are neither
known nor can be appropriately represented by averages. To make matters worse, uncertainties affect
not only current CCN but also background CCN estimates, as both are needed to derive CDNC changes -
due to CCN saturation effects.
Thus, a simpler approach based on satellite retrievals was selected which directly links aerosol
number with cloud droplet number, as explained in more detail in Appendix A. It involves retrievals of
the same satellite sensor for fine-mode AOD (AODf) as a proxy for aerosol number and for CDNC at the
cloud-tops of low altitude clouds as proxy for cloud droplet number. Based on multi-spectral sensor data
both properties can be retrieved but not simultaneously at the same time and location. Here it is now
stipulated, that regional associations of monthly averages between AODf and CDNC offer meaningful
statistical constraints on aerosol-cloud interactions. Combining all monthly local matches over oceans,
where both, quality (no side-viewing retrieval, no broken cloud scenes) CDNC retrievals and sufficiently
large AODf retrievals (greater 0.05) were available, the AOD-size bin median associations are best
captured by a logarithmic relationship (more details are provided in Appendix A).
**CDNC, factor = ln (1000\*AODf [natural +anthropogenic] +3) / ln (1000\*AODf [natural] +3)**
The relationship predicts a CDNC increase factor due to extra AODf. Hereby the CDNC increase
depends not only on extra AODf from anthropogenic sources but also on natural AODf background. The
factor 3 in the formula accounts for CDNC associated with coarse-mode aerosols, which are important to
be included at very clean background conditions.
The CDNC-AODf relationship associated with observational statistics did not change significantly
with the use of different satellite sensor data (as illustrated in Appendix A). In contrast, the CDNC factor
increases in AeroCom global modeling were found more variable and on average much stronger (as
shown in Appendix A).  In other words, if observational associations can be trusted, then in most
AeroCom models (with detailed aerosol schemes) the Twomey effect (associated with extra
anthropogenic aerosol) is over-parameterized. Note, that for better comparison the modeling data were
subsampled at locations of contributing CDNC vs AODf retrieval pairs.
For estimates of the aerosol first indirect effect in MACv2 the satellite retrieval based CDNC
increase factors were applied for every month and grid point, thus also over continents where no and
not sufficiently accurate satellite retrievals were available. After CDNC increases are converted into
cloud droplet radius reductions (dR = 1/d(CDNC)\*\*[1/3]), assuming no changes to the cloud liquid water
content) two scenarios were simulated in an off-line radiative transfer code: The one scenario applied
reduced cloud droplet sizes according to the CDNC increases and the other scenario used the base-line
droplet size. The differences of these two simulations define aerosol indirect effects in MACv2. Present-
day annual average indirect effects for the solar and the infrared spectral region at TOA and surface are
presented for local anthropogenic AOD in the context of pre-industrial fine mode AOD in Figure 12.

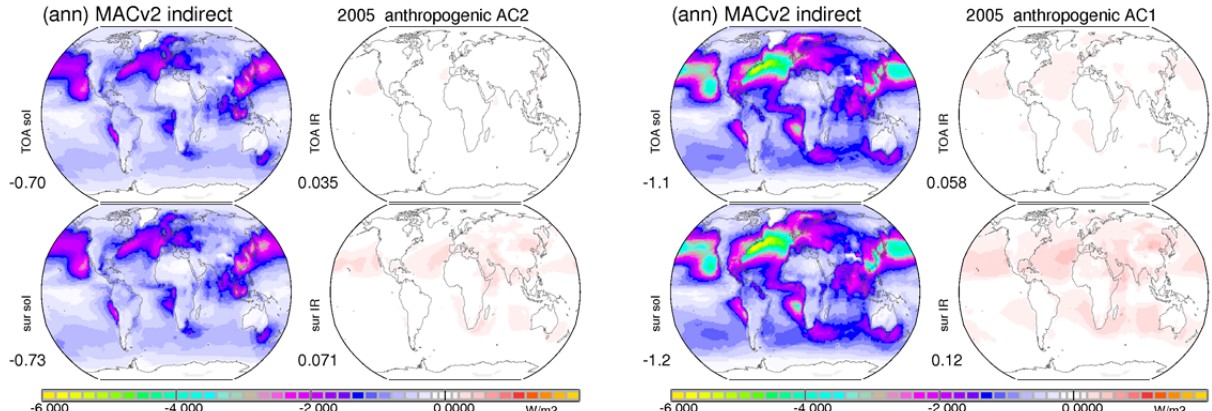

**Figure 12** *Annual maps of indirect (Twomey) effects for present-day anthropogenic aerosol based on year 1850 anthropogenic fine-mode fractions of AeroCom 2 (left block) and on 1750 anthropogenic fine-mode fractions of AeroCom 1 (right block). Compared are in each block solar (left column) and infrared (right column) radiative effects at TOA (top row) and at the surface (bottom row). Blue colors indicate a 'cooling' and red colors a 'warming'. Values below the labels are global averages.*

Figure 12 compares the impacts for two different estimates for the present-day anthropogenic AOD.  One definition is based on the year 1850 and CMIP5 emissions (*Lamarque et al., 2010*), the MACv2 standard. The other definition applies 1750 as reference year and AeroCom phase 1 emissions (*Dentener et al., 2006*). Global annual associated anthropogenic AOD maps are given in Appendix E in Figure E1.

The aerosol first indirect radiative effect is mainly a solar response with increases to the planetary albedo and complementary decreases to the solar surface net-fluxes. There is a lot of spatial variability with the largest contributions over (dark) oceans and the Northern Hemisphere. On a global annual average basis (with the pre-industrial year 1850 reference and CMIP 5 emissions) the low cloud solar cooling at -0.70 W/m2 dominates its infrared (greenhouse) warming at +0.04 W/m2. With the alternate (ca 30% larger) anthropogenic AOD the low cloud solar cooling increases by 50% to -1.1 W/m2. The strong response is explained for two reasons: Aside from the increase to the anthropogenic AOD also the (pre-industrial) background is reduced. And with the reduced background (via the logarithmic relationship) the CDNC response is stronger with respect to the same AODf perturbation. Thus, the indirect aerosol forcing is very sensitive to anthropogenic assumptions. Atmospheric radiative effects (e.g. solar heating) are small, as cloud impacts at TOA and surface are almost identical.

Spatial variations on a monthly basis for simulated present-day aerosol TOA indirect forcing of MACv2 (with the year 1850 reference and CMIP5 data) are presented in Figure 13. Figure 13 illustrates that the aerosol indirect forcing (via the Twomey-effect) is not just influenced by anthropogenic aerosol and background aerosol conditions but also by environmental properties. Important elements are solar energy, sun elevation, solar surface albedo, likelihood of single layer low altitude clouds and a moderate low cloud optical depth for highest susceptibility. The indirect forcing via smaller cloud drops is strongest over mid-latitude oceans (with relative dark surfaces), during spring and summer (with longer sun-shine hours) and mainly the Northern Hemisphere (where most anthropogenic aerosol is found). Indirect effects over stratocumulus regions are relative moderate.

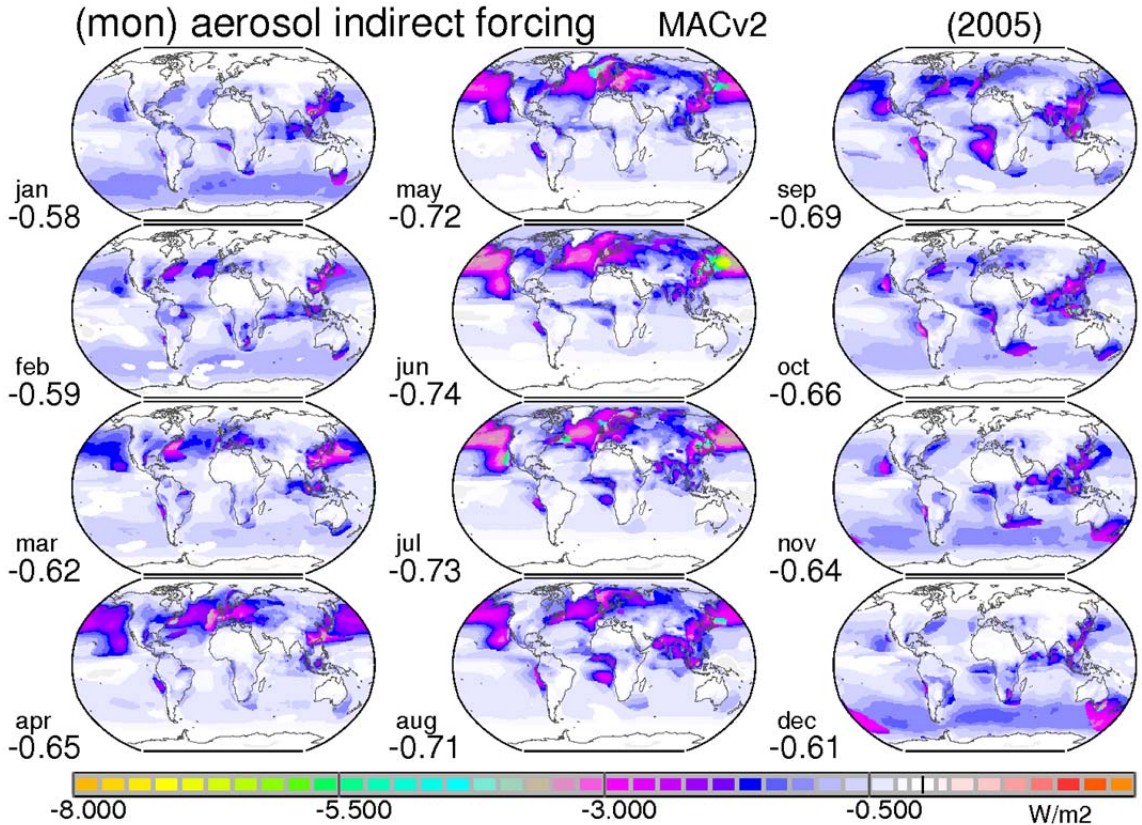

**Figure 13** *Monthly maps for today's indirect (Twomey) forcing by present-day anthropogenic aerosol*

Extra analysis of local environmental potential for a Twomey effect and maps for indirect forcing efficiencies is provided in Appendix F. Due to very low aerosol natural background conditions, indirect forcing efficiencies over Southern Oceans can be very high. Overall, however, contributions from those regions are minor. Nonetheless, possible indirect effect overestimates in these regions cannot be ruled out, also because the AODf vc CDNC relationship is not observationally constrained at very low AODf.

## 8. Direct vs Indirect

MACv2 associated aerosol present-day direct and indirect radiative effects are now compared. Annual maps for TOA direct radiative effects at clear-sky conditions, for direct forcing (with clouds), indirect forcing and the combined forcing (direct and indirect) are presented in Figure 14.

At the TOA climate cooling by the present-day indirect effect (globally averaged at -0.66 W/m2) is about twice as large as the present-day direct effect (globally averaged at -0.35 W/m2). However, the spatial variability of the direct forcing is much more diverse, including regions with local warming (via dimming over snow, over lower clouds or during polar nights). Direct effect dominates near continental sources while the indirect effect has stronger impacts over oceanic regions off sources. The combined climate impact is a cooling (globally averaged at -1.0 W/m2) with cooling everywhere except over Greenland. The present-day clear-sky cooling is about 60% of the combined (direct and indirect) cooling.

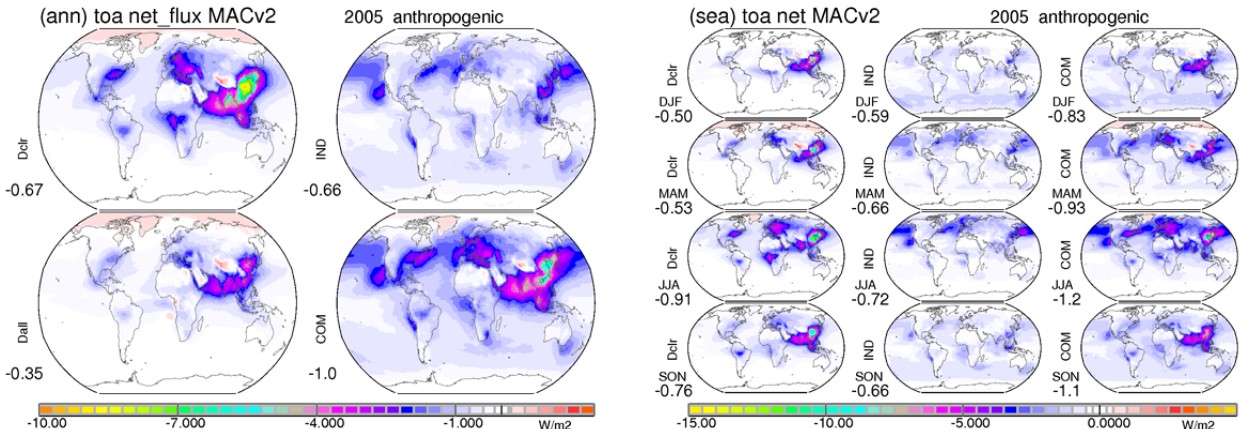

**Figure 14** *TOA radiative effects by present-day anthropogenic aerosol. Annual (left block) and seasonal maps (right block) compare direct radiative effects at clear-sky conditions (Dclr) and all-sky conditions (Dall), aerosol indirect (Twomey) effects through modified clouds (IND) and the combined (direct and indirect) effect (COM). Blue colors indicate 'cooling' net-flux losses and (rare) red colors indicate 'warming' net-flux gains. Values below the labels indicate global averages.*

Figure 14 also compares seasonal variations. With larger AOD, less snow cover and more sunshine both indirect and direct effects have maximum impacts during the northern hemispheric summer season.

Annual maps for radiative effects at the surface and for the atmosphere are shown in Figure 15. Compared are (as in Figure 14) direct radiative effects without and with clouds, indirect radiative effects and the combined effects (direct with clouds and indirect).

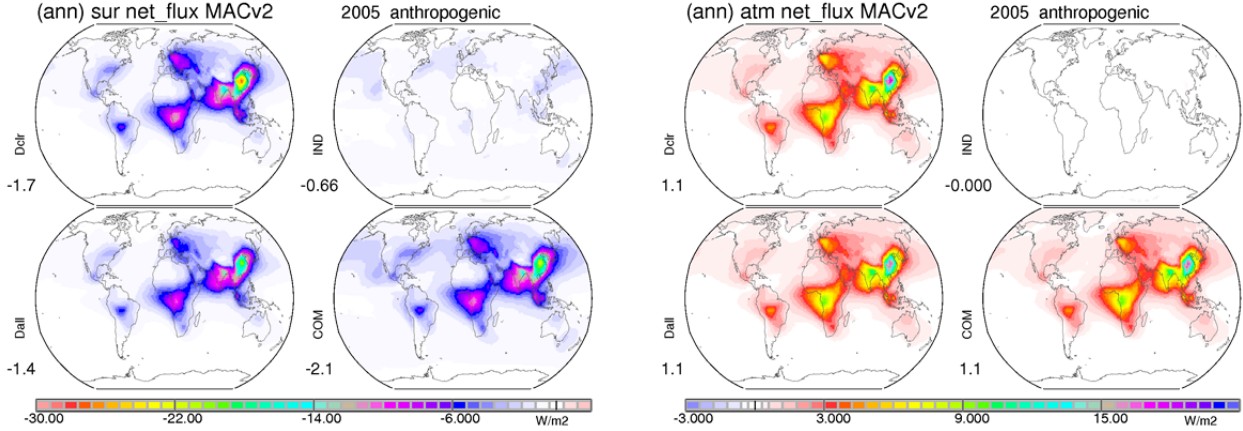

**Figure 15** *Annual radiative effects by present-day anthropogenic aerosol at the surface (left block) and for the atmosphere (right block). Maps compare direct effects at clear-sky conditions (Dclr) and all-sky conditions (Dall), indirect (Twomey) effects through modified clouds (IND) and the combined (direct and indirect) effect (COM). Blue colors indicate 'cooling' net-flux losses and red colors indicate 'warming' net-flux gains. Values below the labels indicate global averages.*

1       In the atmosphere, present-day anthropogenic aerosol warms the atmosphere – on average at
+1.1 W/m2. The atmospheric warming is highly uneven with warming in excess of 10W/m2 near sources
of pollution (e.g. S.Asia, E.Asia) and wildfire regions (e.g. central Africa). The associated solar heating is
almost entirely a the direct effect. Thus, aerosol direct effects control the atmospheric response.
5       At the surface the present-day direct effect yields, due to atmospheric losses, a more negative
radiative effect than at TOA. The present-day direct effect reduces the (solar) flux is on average by about
-1.45 W/m2 . This is stronger than the flux reductions of about -0.66 W/m2 for the indirect effect. Thus
on a global annual average basis, at surface net-flux losses by aerosol are dominated by direct effects.
9       Monthly maps for the present-day combined (direct and indirect) TOA forcing are shown in
Figure 16.  More maps on TOA forcing including fractional contributions of clear-sky regions and cloudy-
sky regions are presented in Appendix C.

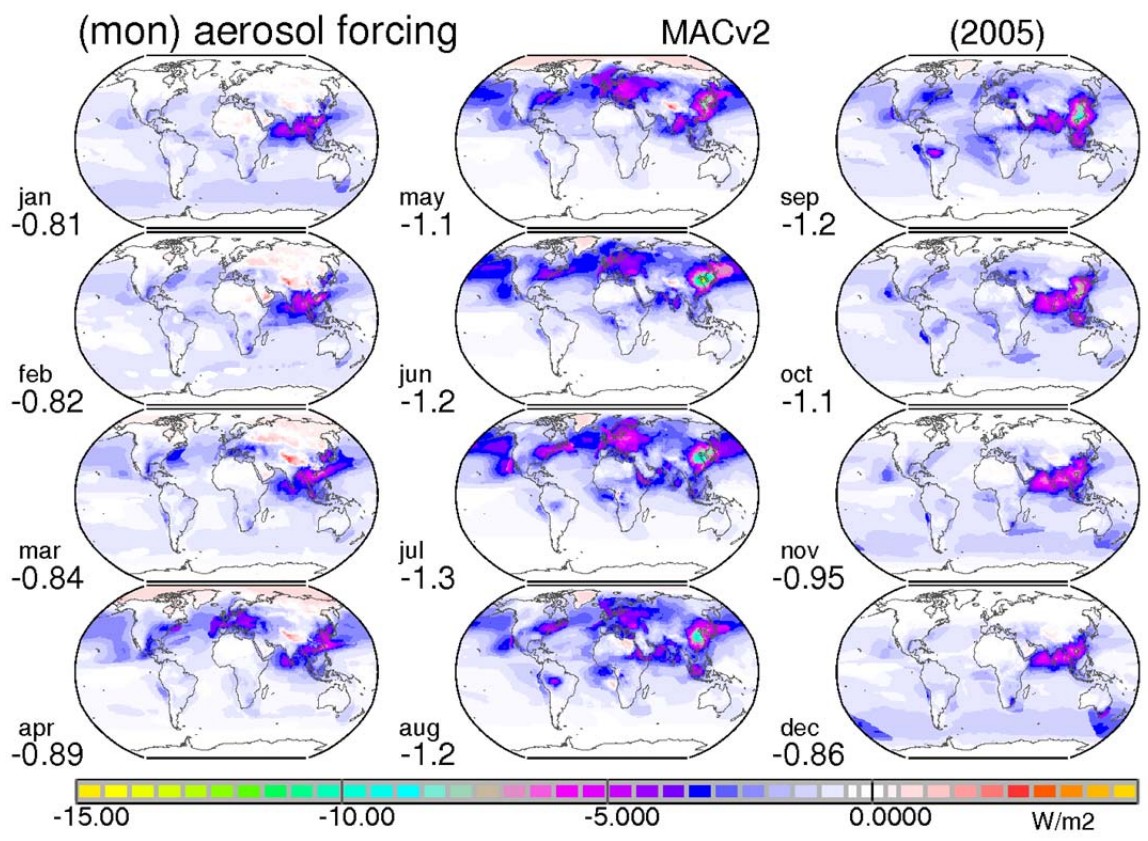

*Figure 16* *Monthly maps for today's total forcing by present-day anthropogenic aerosol*

Interestingly global averages of the clear-sky direct impact at TOA (in Figure 14) and at the
surface (in Figure 15) resembles in magnitude and regional distribution that of the combined (direct and
indirect) impact. Thus, for rough estimates on the aerosol impacts, already clear-sky radiative
simulations could provide rough estimates for global averages for aerosol climate impacts.

# 9. Forcing over Time

The MACv2 aerosol climatology offers global maps for changing anthropogenic AOD over time (*Kinne, 2019, Figure 8*). Hereby the historic scaling back to year 1850 is based on transient 'bottom-up' ECHAM simulation *(Stier et al., 2005)* with NIES emissions, while future scaling is based on regionally simulated responses to changing sulfate emissions of the IPCC 5 RCP 8.5 future scenario. Results of radiative transfer simulations that apply these changing anthropogenic AOD data over time are presented in Figure 17. Maps for selected years (from 1865 to 2065) compare in 40-year steps the annual forcing of the direct, the indirect (Twomey) and the combined (direct and indirect) effect.

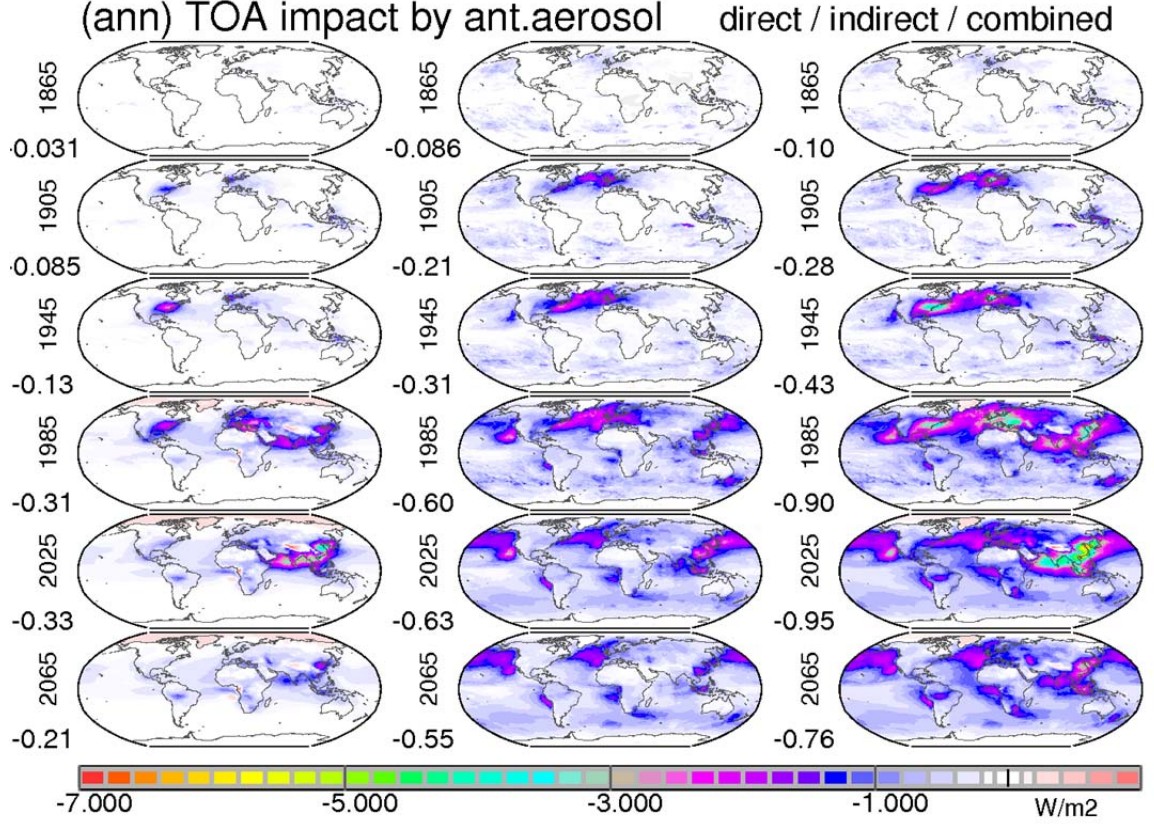

*Figure 17. temporal evolution of the climate forcing by anthropogenic aerosol. For selected years (1865, 1905, 1945, 1985, 2025 and 2065) annual maps or the direct (column 1), the first indirect (column 2) and the combined effect (column 3) are compared. Values below the labels indicate global averages.*

The forcing time-slices show that in the early years of the industrial revolution the fractional forcing contributions by indirect effect were relatively high. This is a consequence of a stronger indirect response at a lower background conditions, which is predicted by the applied logarithmic relationship.
The temporal time-slice maps demonstrate that early into the industrial period mainly the US and Europe where affected by aerosol cooling. By 1985 with SE Asia had emerged as a third affected

region. Since then the aerosol cooling over SE Asia kept on growing while aerosol cooling by over Europe
and the US declined, also due to successful mitigations efforts. With these opposing regional shifts
during the last decades the global average aerosol cooling stayed relatively stable at just below -1.0
W/m2. As in the meantime (by 2015) anthropogenic aerosol loads had reached their regional maximum
over E Asia (while not yet over S Asia) no further increases for future aerosol cooling are being expected,
even if future emission scenarios project the development of a new maximum over W Africa.
7        A different way to illustrate changes and regional shifts associated with extra anthropogenic
aerosols are forcing difference over selected time-periods. Total TOA forcing differences and changes to
the downward solar flux at the surface presented by season for three 40-year intervals in Figure 18.

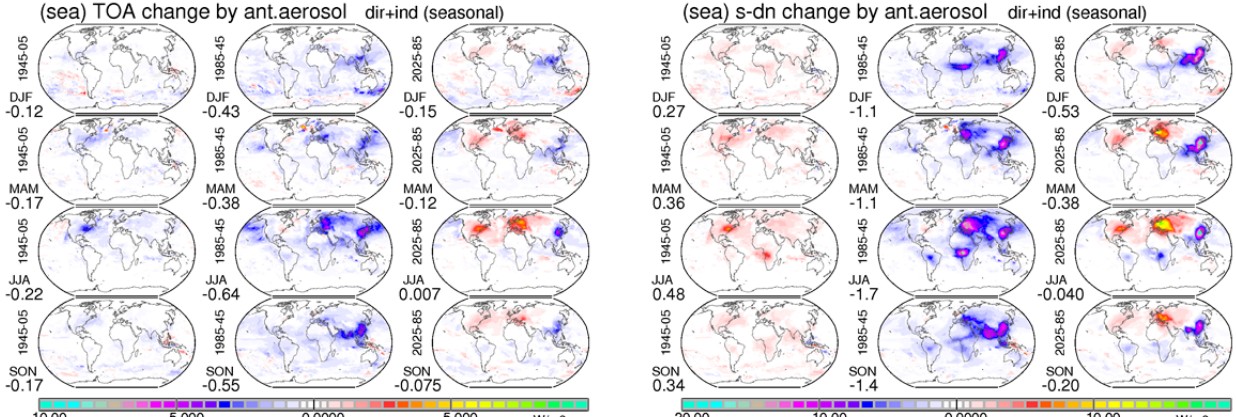

***Figure 18****. Seasonal changes over 20 year periods (1945-1905, 1985-1945, 2025-1985) in aerosol forcing*
*(left block) and to solar downward fluxes by anthropogenic aerosol (right block). Blue colors indicate*
*time-periods of increased cooling aerosol or a dimming (decreases to the solar surface fluxes), while red*
*colors indicate time-periods of a reduced cooling by aerosol or a brightening (increases the solar surface*
*fluxes).  Values below the labels indicate global average changes for the selected time periods.*
20        Between 1945 and 1905 TOA cooling occurred mainly over the eastern US during summers.
Between 1985 and 1945 TOA cooling strongly increased during summer over near Europe and E. Asia
and during the (dry) fall season over S. Asia. Between now and 1985, the (mainly summer) TOA cooling
strongly decreased over Europe and the eastern US but kept increasing over SE Asia.
24        Also shown in Figure 18 are MACv2 aerosol simulated changes to the downward solar fluxes.
These temporal regional and seasonal changes are consistent with observations by ground-based
radiation (*Wild, 2015*). These results strongly suggest that the observed long-term average dimming
(until 1985) following a long-term average brightening (since 1985) can be mainly attributed to
anthropogenic aerosol.

# 10. Uncertainty

The calculations of the aerosol radiative effects and the aerosol radiative forcing include many uncertainties. The focus here is on uncertainties to the aerosol fields of the MACv2 (although also approximations in radiative transfer model, simplifications to environmental data and the application of monthly averages will contribute). An initial test in Figure 19 compares the impact of a different background maps for AODf and AODc in developing the MACv2 aerosol climatology.

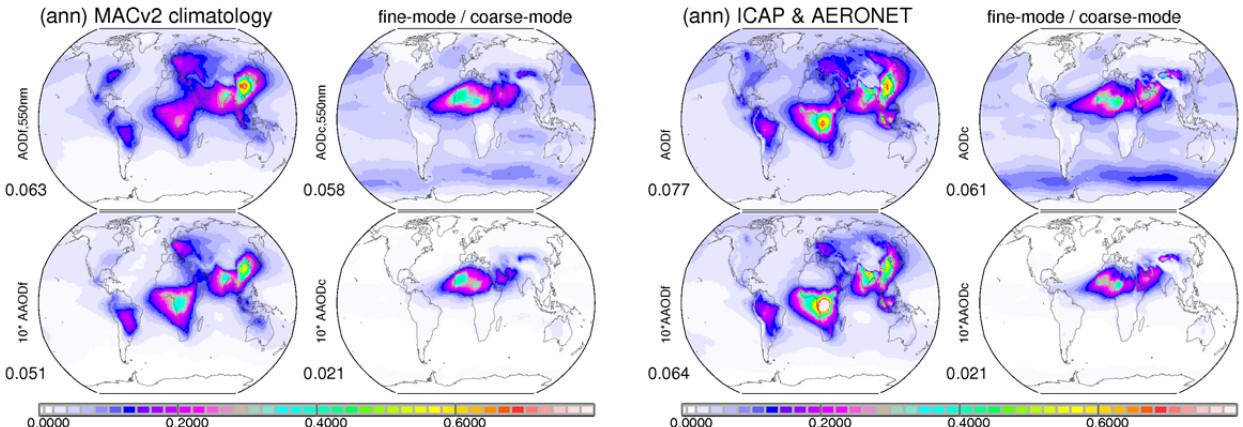

***Figure 19*** *estimates of today's annual maps for tropospheric AODf, AODc, AAODf and AAODc by combining AERONET data with an AeroCom1 median (left block) and with satellite AOD data assimilating ICAP averages (right block). AAOD data are multiplied by 10. Values below labels show global averages.*

As an alternate modeling background in the data merging of MACv2, ICAP ensemble data (*Peng et al., 2018*) from the satellite AOD assimilation community are used. Three hourly data of years 2015 and 2016 for averages of up to seven assimilations are combined into local monthly mean AODf and AODc. Global maps of these aerosol properties replace AODf and AODc maps of the AeroCom phase 1 background data in the merging with the sun-photometer statistics. All other properties (e.g. the absorption strength of each size mode) remain identical.

The climatology with the ICAP background yields on average ca 15% larger values for the mid-visible AOD. Patterns are slightly shifted but major differences are much stronger AODf (and AAODf) maxima over central Africa. It is difficult to judge if these differences are the result of (in assimilations applied MODIS) satellite AOD retrieval overestimates in that region or due to overlooked emissions in modeling. Unfortunately, there are no AERONET reference sites in that region. Globally the AODf is ca 25% larger. Fortunately, most of this extra AODf has a strong natural component so that anthropogenic AOD is increased by just 10% as shown in Figure 20.

The assumed pre-industrial reference states to define 'anthropogenic' contributions introduce probably the largest uncertainty. To illustrate this point, the standard anthropogenic AODf fraction based on a reference year 1850 and CMIP5 emissions (*Lamarque et al., 2010*) was replaced by an alternate approach based on a reference year 1750 and AeroCom emissions (*Dentener et al., 2006*).

Figure 20 shows that with the alternate approach the anthropogenic AOD in MACv2 increases (globally
averaged) by 30% from 0.031 to 0.040. Considering the uncertainty in AOD but limiting the reference to
the year 1850 then the range for anthropogenic AOD (at 550nm) is between 0.030 and 0.038. Another
big uncertainty is the anthropogenic fraction for soot (BC) component, which dominates the absorption.
The present-day anthropogenic BC component warming is uncertain between +0.25 and 0.45 W/m2. If
all of the present-day BC is a considered anthropogenic then there is an +0.55 W/m2 upper ceiling. Thus,
there is no support for larger present-day BC direct warming contributions (as in *Bond et al, 2013*).
The neglect of larger coarse-mode aerosol in the anthropogenic definition in MACv2 seems less
important. Potential coarse-mode anthropogenic contributions via mineral dust (e.g. due to land use
change) seem secondary, especially as anthropogenic dust forcing is near neutral, with solar and
infrared radiative effects largely canceling each other.

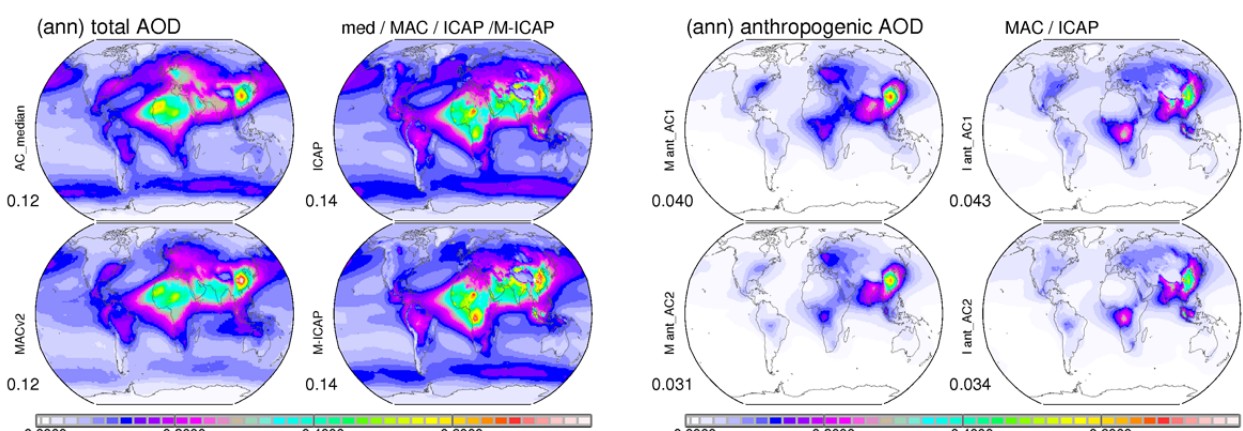

***Figure 20*** *uncertainty estimates for present-day total and anthropogenic AOD from different realizations.*
*Annual maps for tropospheric total AOD (left block) and for anthropogenic AOD (right block) are shown.*
*Total AOD maps compare model-ensemble data of AeroCom1 experiments (left column) and ICAP*
*satellite data assimilations (right column), before (top) and after (bottom) the data merging with sun-*
*photometer data. Anthropogenic AOD maps are results of applying the two different anthropogenic*
*fractions (based on emissions by Dentener - AC1 and Lamarque - AC2) to the fine-mode AOD of the*
*AERONET adjusted maps. Values below the label show global averages.*

Sensitivity studies in Appendix B demonstrate that a 25% larger anthropogenic AOD means a
25% stronger direct forcing and a 40% larger indirect forcing. An alternate ICAP background in the data
merging of MACv2 increases the total AOD for an up to a 15% stronger aerosol direct and aerosol
indirect forcing. And a likely BC anthropogenic fraction that is larger than the fine-mode anthropogenic
fraction will contribute to a less negative direct effect. Also the treatment of the indirect effect makes
many assumptions and is highly simplified. Finally all presented results refer to an instantaneous impact.
Short-term cloud adjustments are not included, but they are much smaller (on the order of 10%) in
comparison (Fiedler et al., 2019) with a tendency to reduce the aerosol radiative effects.
Considering all contributing uncertainties, a -0.7 to -1.6 W/m2 range is estimated for present-
day global annual average aerosol forcing (assuming a 1850 pre-industrial reference year) and a most
likely value near -1.0 W/m2. Note, that the less negative lower bound (-0.7 W/m2) is more certain than
the more negative (-1.6 W/m2) upper bound.
3         The main argument towards the upper bound (-1.6 W/m2) is that the anthropogenic fine-mode
fraction could be larger. The upper bound of the direct effect at -0.45 W/m2 is much better constrained
than the upper bound for the indirect effect. An alternate anthropogenic scaling using AeroCom phase 1
emissions raised indirect effects (from -0.65) to -1.1 W/m2 and direct effects (from -.35) to -0.50 W/m2
for a combined -1.6W/m2 cooling. However, this alternate larger anthropogenic fine-mode fraction
applies an earlier pre-industrial reference year (1750). Thus, such a negative aerosol forcing is unlikely
for a year 1850 reference. Also the simplicity in describing the aerosol indirect effects in MACv2 is very
simple. There are calls for a more processed based modeling treatment or observational approaches
which better distinguish specific aerosol and cloud properties (*Grandey et al., 2010*). However, since the
eventual global patterns for the indirect radiative forcing are already well matched (as environmental
properties play a big role), a more detailed indirect effect treatment may come at the expense of added
uncertainty. Another argument for an upper bound of -1.6 W/m2 is that such strong cooling potential by
anthropogenic AOD at the middle of the 20[th] century would have caused global climate cooling which
was not observed *(Kretzschmar et al, 2016)*.
17        The main argument for the lower bound (-0.7W/m2) is that the lower bound of the direct effect
(which is better constrained than the indirect effect) is estimated at -0.2 W/m2. This lower value can be
explained with a possibly higher anthropogenic fraction for absorbing BC AOD (than simply applying the
anthropogenic fine-mode AOD for anthropogenic BC). For the present-day indirect effect there remains
uncertainty, however, at least a cooling of -0.5 W/m2 should be expected for the reduced drop size
effect.  (In that context it should be noted that the less negative indirect effects of MACv2-SP in
Appendix B do not qualify due to their incomplete global coverage for anthropogenic AOD).
24        To reduce the aerosol forcing range (-0.7 to -1.6 W/m2) progress in quantifying the indirect
effect(s) and pre-industrial references for all aerosol components are needed.
**11. Summary**
29        The MACv2 global monthly aerosol climatology, tied to observational monthly statistics on
aerosol amount, absorption and size from sun-/sky-photometry, was applied in dual-call off-line
radiative transfer to determine aerosol effects on atmospheric radiation. The direct (added presence)
effects of present-day aerosol and the climate change relevant effects of anthropogenic contributions
(since pre-industrial times) are determined. Hereby results are usually presented via global annual and
even monthly maps to visualize regional and temporal detail, as not only aerosol properties but also
influential environmental properties have strong regional and seasonal signatures. Still, for simplicity,
most discussions below apply resulting global annual averages. Hereby radiative net-flux changes are
examined at the climate relevant TOA location, at the surface for exchange processes near the ground
and for the atmospheric dynamics (via the TOA minus surface impact). Radiative net-flux losses refer to
a cooling and radiative net-flux gains refer to a warming.
40        Present-day total aerosol by its presence (direct effect) reduces net-fluxes by -1.1 W/m2 at the
TOA and by -4.0 W/m2 at the surface, so that the energy in the atmosphere is increased by +2.9 W/m2.
Hereby, the net-flux losses are composed of larger solar losses and a partly offsetting smaller infrared
gains, at -1.8 and +0.7 W/m2 at the TOA and -5.5 and +1.5 W/m2 at the surface, while atmospheric solar
warming of +3.7 W/m2 is reduced by -0.8 W/m2 in atmospheric IR cooling.

4         Anthropogenic aerosol is only considered to contribute to sub-micrometer aerosol sizes so that
only solar direct radiative effects matter (and potential IR radiative effects can be neglected). Present-
day anthropogenic aerosols by their added atmospheric presence (direct effect) are estimated to reduce
solar TOA net-fluxes by -0.36 W/m2 and surface net-fluxes by -1.5 W/m2. By difference the atmosphere
is warmed by +1.1 W/m2. In addition, the major aerosol induced impacts to water clouds is considered
(the first indirect effect). More numerous aerosols reduce the cloud droplet sizes. This Twomey effect is
implemented via satellite retrieval based associations between aerosol and drop concentrations. The
present-day first indirect effect reduces mainly solar radiative net-fluxes at the TOA and at the surface
by -0.65 W/m2. Thus, the combined direct and indirect yields a cooling of -1.0 W/m2 at the TOA and of -
2.1 W/m2 at the surface. On average, the indirect effect dominates the (TOA) climate cooling, the direct
effect dominates the cooling at the surface and the direct effect determines the atmospheric heating.
Spatially both direct and indirect effects are strongest during the NH summer. Direct effects are
strongest near continental sources. Indirect effects are stronger away from sources over (dark) oceans.

17         Uncertainties with respect to the aerosol properties should consider that the total AOD could be
15% larger than in MACv2 and that a 25% larger fine-mode anthropogenic fraction than MACv2 is
possible. Simulations with a 30% larger anthropogenic fraction, yield a 30% larger direct effect and an
even 50% larger indirect effect. Thus, the uncertainty for present day forcing (the anthropogenic impact
at the TOA) is very likely between -0.7 and -1.6 W/m2, with -1.0 W/m2 as best estimate. Hereby, the
direct effect is much better constrained (-0.20 to -0.45 W/m2) than the indirect effect (-0.5 to -1.1
W/m2). A better estimate for pre-industrial references (for fine-mode, the BC component and even for
dust) would help in reducing these uncertainties. And also the rather simple representation of the
indirect effect needs to be validated.

26         As MACv2 optical properties also allow to attribute direct radiative effects to components. The
'top-down' component estimates of this presentation for present-day forcing agree with those from
'bottom-up' modeling, as summarized in Appendix G. Present-day total soot (BC) warms at the TOA with
+0.55 W/m2. At least 50% and at most 85% could be attributed to BC, depending on the definition for
the anthropogenic BC fraction but certainly not more than 100%. With the possibility that the
anthropogenic fraction for soot (BC) that is larger than for the fine-mode fraction (in other words that
there was much less BC in the fine-mode aerosol in pre-industrial aerosol) then the assigned  BC
warming (from now +0.28 W/m2) would increase (to about +0.38 at 70% fraction or even +0.45W/m2 at
an 85% fraction). This is turn would reduce the present-day direct aerosol forcing to a -0.26W/m2 (at
70% anthropogenic BC) or just -0.19W/m2 (at 85% anthropogenic BC).  Thus, not only accurate
information on the pre-industrial state of the fine-mode AOD but also on its components, mainly that of
the BC component, at that time (e.g. soot properties) is needed.

38         The combined carbon (BC and OC) component, which approximates the effect of BC co-emitters,
has a near neutral radiative forcing behavior. Thus, short term climate warming mitigation concepts via
a soot removal may not be very effective as also mainly scattering aerosol resulting from co-emitted
trace-gases would be removed. The climate impact for mineral dust (while showing strong warming over
continents, yet strong cooling over oceans) behaves globally almost climate neutral. Thus, potential

anthropogenic impacts from dust, as a result of land-use change, as uncertain as they are, seem less importance for climate change considerations. Thus in the end it are the (in the mid-visible) non-absorbing sub-micrometer size aerosol (mainly sulfate and nitrate) that regulate the anthropogenic AOD.

Calculations with model predicted temporal changes to the anthropogenic AOD indicate that qualitatively the anthropogenic aerosol forcing has not changed much over the last decades and is not likely to increase over the next decades, despite strong regional shifts. These regional shifts explain most solar insolation (brightening or dimming) trends that have been locally observed by decadal time-series ground-based radiation, especially over Europe and the US.

## Resource

MACv2 properties are accessible at [ftp://ftp-projects.zmaw.de/aerocom/climatology/MACv2_2018/](ftp://ftp-projects.zmaw.de/aerocom/climatology/MACv2_2018/) . The data are placed in several subdirectories and a README file describes data content of file-names

| | |
|---|---|
| /550nm | (mid-visible) aerosol properties at 550nm wavelength |
| /CCN | lower cloud-base condensation nuclei and critical radii at diff. supersaturation |
| /detail | ancillary data for radiative transfer simulations |
| /documents | some documentation and figures |
| /forcing | MACv2 associated radiative effects |
| /program | fortran code and ancillary data to create MACv2 aerosol properties |
| /program_force | fortran code and ancillary data to determine MAC aerosol radiative effects |
| /retrieval | MACv2 fields for under-determined solar reflection based AOD retrievals |
| /spectral | 2005 optical data at 3 different spectral resolutions: 20, 30 (RRTM),31 bands |
| /time | same as in /spectral … but data for different years (from 1850 to 2100) |

## Acknowledgments

This study relied on observational data when possible. Central to the effort are data provided by the ground-based sunphotometer network of AERONET lead by B. Holben and the MAN network lead by A. Smirnov. Also satellite data of the MODIS and AATSR sensors were applied to quantify aerosol indirect effects. Hereby in particukar CDNC retrievals contributed by D.Grosvenor, J.Rausch and M. Christensen and analysis work by J.Müsse, who created all figures in Appendix A are acknowledged. Another essential element to this study is global model output from simulations with bottom-up processing in aerosol modules as part of the AeroCom initiative lead by M. Schulz and M. Chin. An ensemble median provides data on spatial context, estimates on aerosol anthropogenic fractions (also as a function of time) and aerosol vertical distribution. Thus, all modeling groups contributing to AeroCom experiments are acknowledged. Finally this work was support by EU-projects, in particular the FP7 EU-Bacchus project (603445) lead by U.Lohmann and by ESA's climate initiative, in particular the aerosol-CCI effort lead by T. Popp and G. de Leeuw and coordinated by S. Pinnock.

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

# Appendix A    *the satellite based AODf vs CDNC relationship*

Observational relationships, which employ regional associations between satellite retrieved properties for fine-mode AOD (AODf) and cloud droplet number concentration (CDNC) allow to associate increases in AODf from anthropogenic sources with increases in CDNC. For selected AODf ranges the median CDNC value is determined and then all median CDNC values are combined to result in logarithmic curve fit. By applying the curve twice locally, once for the present-day AODf and once for the pre-industrial AODf an associated CDNC increase in determined. Hereby the CDNC increase depends on the AODf difference and on the AODf (pre-industrial) background. Further assuming that the cloud water of the affected cloud remains constant, CDNC increases are easily converted into drop size reductions, the needed input to simulate associated increases to the planetary albedo, the Twomey effect (*Twomey, 1974*), in off-line radiative transfer simulations.

AODf and CDNC cannot be retrieved by the same sensor at the same time. However spatial associations within larger regions are expected to offer insights on potential relationships. Retrieval averages are matched for relatively large 1x1 degree in latitude/longitude regions. This is in part done to avoid exaggerations in associations, as local indirect effects (e.g. ship pollution impacts) are usually weaker in the context of larger spatial scales (*Coakley and Walsh, 2002*). For better statistics also only monthly averages are matched. Their use seems justified, because at these large spatial scales monthly associations are almost identical to those using daily data instead (*Christensen, private communication*).

Associations between AODf and CDNC were only considered over oceanic regions (where satellite retrievals for both properties are more reliable due to the dark background) and only when both sufficient signal and quality could be assured. Thus, only CDNC retrievals for overcast conditions under near nadir views are considered. The investigated matches include two different MODIS sensor based CDNC retrievals for an entire year (*collection 5.1, for year 2007 provided by Dan Grosvenor, Leeds and collection 6.0 for year 2008 provided by John Rausch, Vanderbilt*) with matching AODf data from NASA's LAADS site and an ATSR sensor based CDNC retrieval (*for year 2008 provided by Matt Christensen*) for a single season with matching AODf data by RAL's ORAC retrieval. 1x1 degree regions with valid matches between AODf and CDNC data are illustrated in Figure A1.

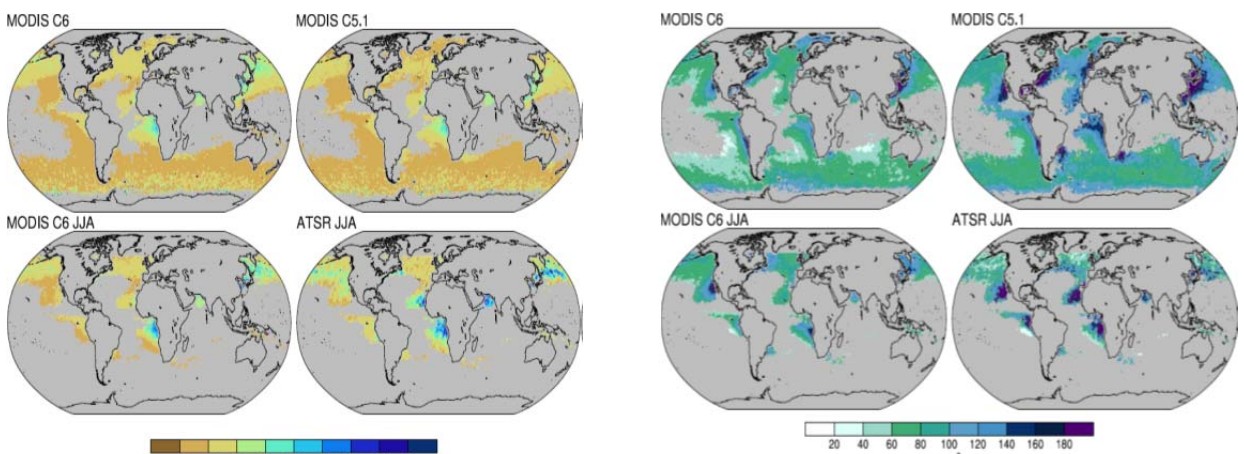

***Figure A1*** *Annual maps (top) and a JJA seasonal maps (bottom) for AODf (left block) and CDNC (right block) for different retrievals with MODIS and ATSR sensor data.*

Between the three different retrievals for CDNC and AODf there are often large absolute
differences. The more important relative differences (as expressed by the logarithmic shapes,
constructed from connection median CDNC points in AODf sub-bins), however, are smaller. The
resulting logarithmic fits along with the data scatter are presented in Figure A2. In that figure the AODf
to CDNC associations of all available months and locations are combined in a single plot.

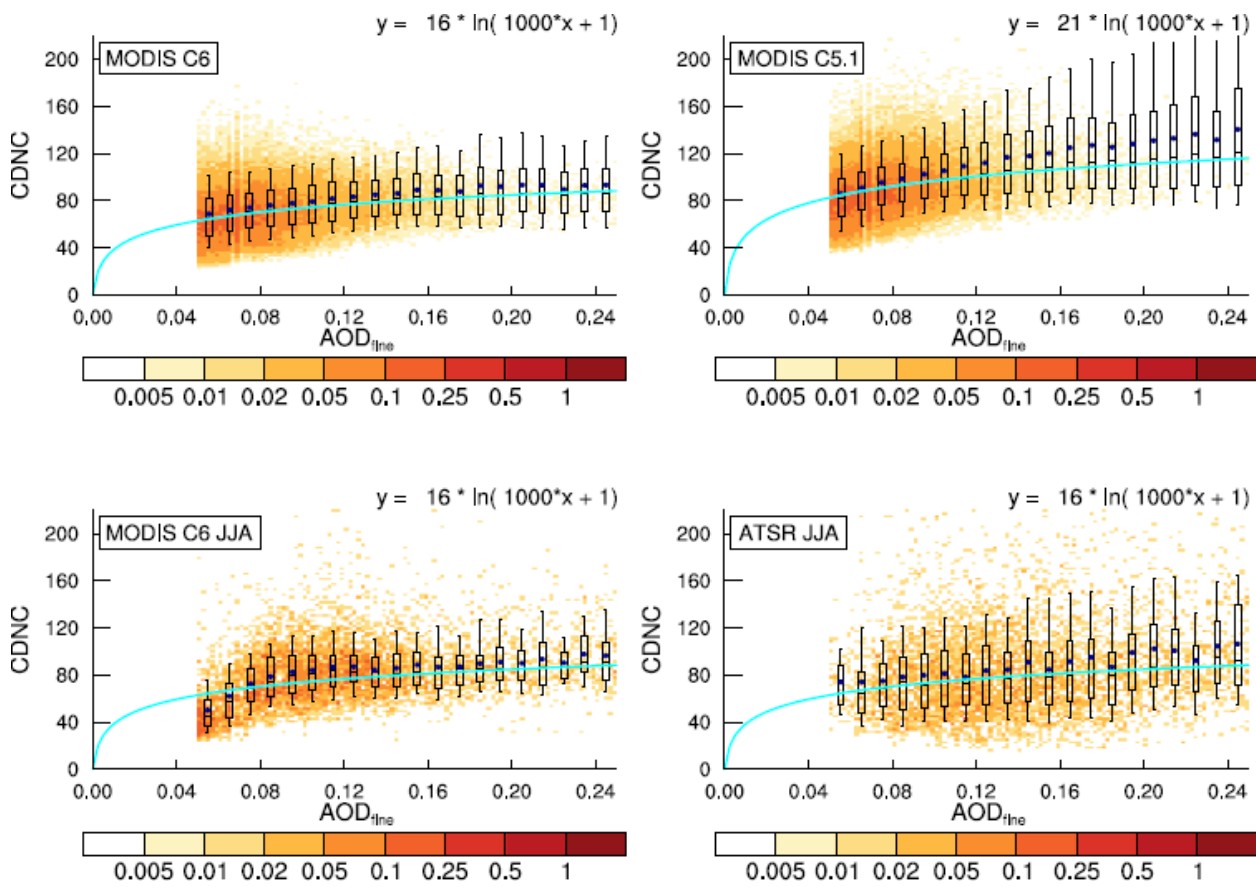

***Figure A2*** *AODf vs CDNC relationships for MODIS 6 (year 2008, upper left), for MODIS 5.1 (year 2007,*
*upper right), for MODIS 6 (JJA of year 2008, lower left) and ATSR (JJA of year 2008, lower right). Matches*
*were removed for mid-visible AODf values smaller than 0.05 (due to too poor signal to noise ratios) and*
*larger than 0.25 (due to poor statistics). For individual AOD bins box-boundaries indicate the upper and*
*lower quartiles, star symbols indicate averages and horizontal lines indicate median values. Logarithmic*
*functions (displayed on top and illustrated by light blue lines) were fitted to the median bin values.*
The fits for the four scatterplots follow an expected log-normal fit as for a given AODf increase
the associated CDNC increase will be smaller the higher the background AODf due to nuclei saturation.
While there are differences to the pre-factors of the fitting functions, the multipliers for AODf (x) are
identical (actually the same multiplier of 1000 applies well for all sensor data). The pre-factor cancels
out when determining CDNC increase factors, because these increase factor are based on the ratio of
two applications (CDNC [natur+anthrop] / CDNC [anthrop]). Thus, for the different satellite sensor data
the derived CDNC factor increases from the different sensor data or data-subsets are basically identical.
3          The agreement for the (observational) fit functions for different satellite sensor data is quite in
contrast to the diversity and to an (on average) much stronger Twomey effect in global modeling. These
results are based on an analysis of output from nine global models with complex aerosol schemes that
participated in the AeroCom ([http://aerocom.met.no](http://aerocom.met.no)) indirect experiment. For the comparisons, the
model simulated output for AODf and CDNC was sub-sampled at the same locations of the satellite
matches. Scatter plots and associated fit functions between observational data (here MODIS coll.6) and
the AeroCom model ensemble average are presented in Figure A3.

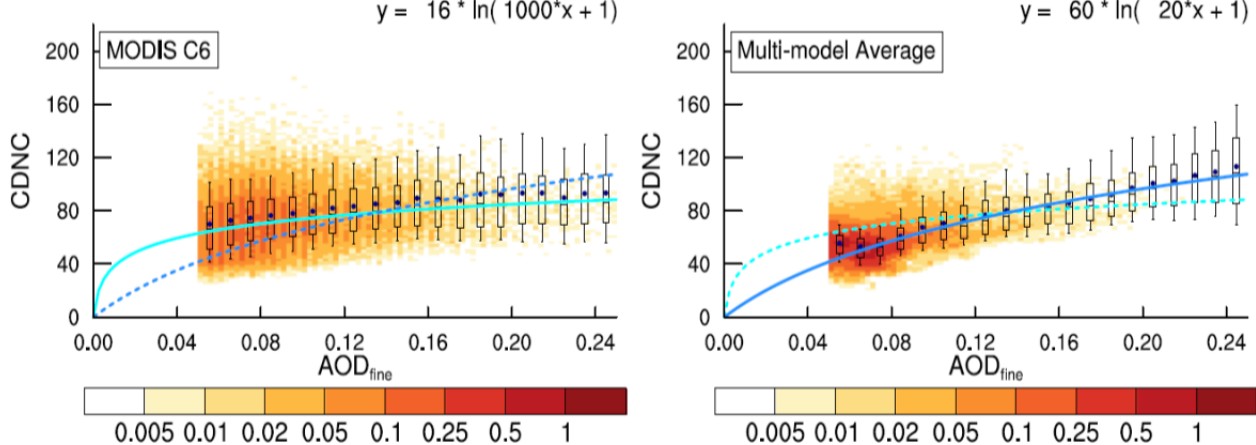

***Figure A3*** *AODf vs CDNC relationships as in figure A2 here based on a satellite retrieval (left, here MODIS*
*6) and on the AeroCom model ensemble average (right). Note the much steeper slope in global modeling*
*(dark blue lines) which indicates a much stronger Twomey effect by global modeling than by satellite*
*retrieval based 'observations' (light blue lines).*

20          The '+1' security value in the logarithmic fit (to avoid negative CDNC values) for the fine-mode
AOD relationship to CDNC is raised '+3' to account for additional nuclei from coarse-mode aerosol.
These contributions are particular important in fine-mode sparse regions over the southern oceans, to
avoid potential overestimate for CDNC-factor increases. Thus:
25          **CDNC, factor = ln (1000\*AODf [natural +anthropogenic] +3) / ln (1000\*AODf [natural] +3)**
27          Applying today's total (natural and anthropogenic) AOD and natural AOD of MACv2, the CDNC
increase factors due to anthropogenic aerosol are determined. And with the assumptions that the water
content remains unchanged these CDNC increases are easily converted into droplet radius reductions.
31          **radius, factor (%) = 100 / [ (CDNC, factor)\*\*(1/3) ]**
33          The seasonal averages for CDNC factor increases and water cloud droplet radius reductions (in
%) associated with today's anthropogenic aerosol in the context of pre-existing aerosol are presented in
Figure A4. There larger cloud droplet radius reductions are on the order of 10 %.

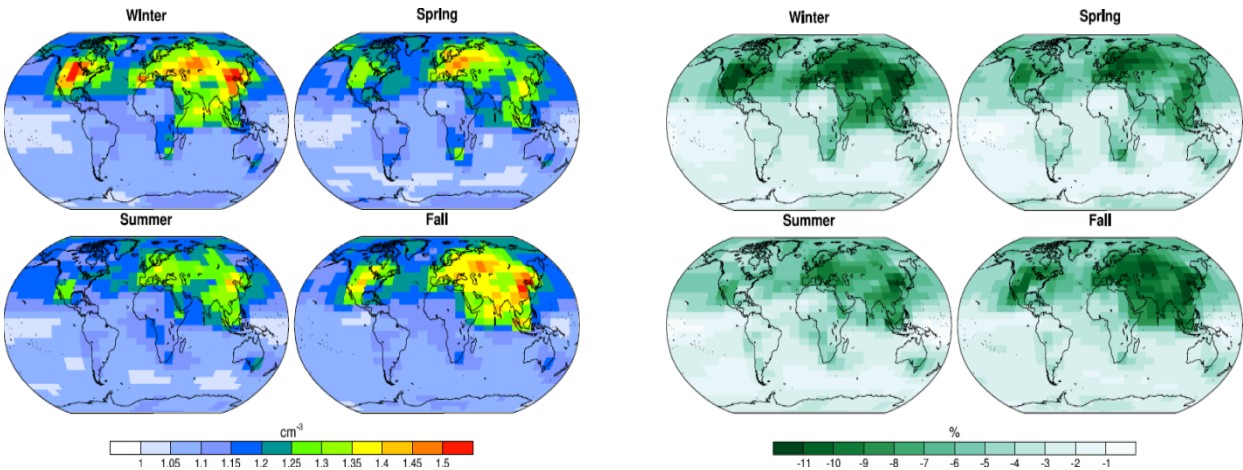

Figure A4 *seasonal CDNC factor increases (left) and associated droplet radius reductions in % (right) due to today's anthropogenic aerosol and the pre-industrial fine-mode background as defined by MACv2.*

# Appendix B    *MACv2 vs MACv1  and  MACv2 vs MACv2-SP*

There are different MAC climatology flavors for aerosol optical properties in circulation. As impact differences are of interest, radiative effects at the TOA by today's anthropogenic AOD of the MACv2 climatology are compared to those when applying in off-line simulations (1) aerosol optical properties of the older MACv1 climatology (*Kinne et al., 2013*) or (2) aerosol optical properties of the plume approximation for anthropogenic AOD (*Stevens et al., 2017, Fiedler et al., 2019*) instead. Annual maps for direct (clear-sky and all-sky) aerosol effects at TOA are compared in Figure B1.

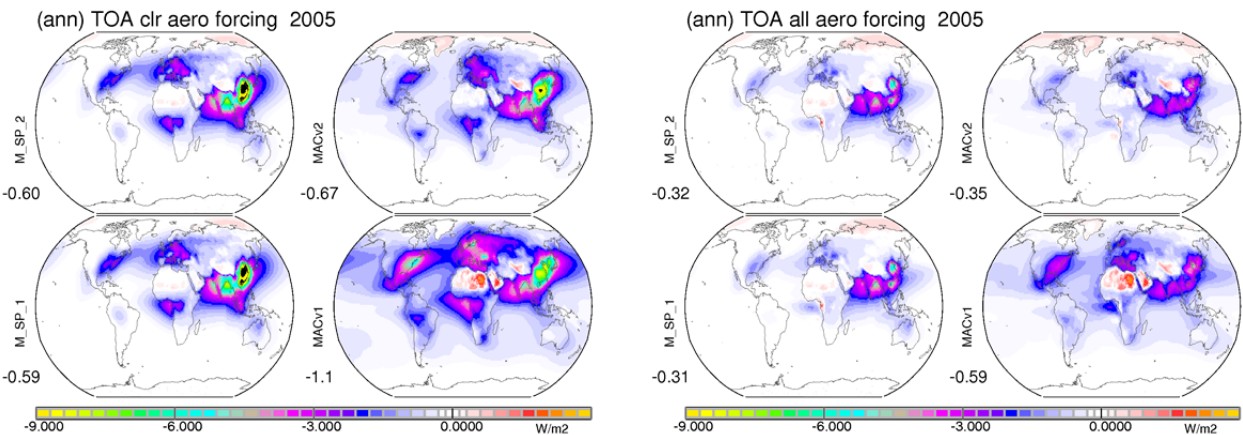

Figure B1. *annual maps for present-day aerosol direct radiative forcing in W/m2 at the TOA at clear-sky conditions (left block) and at all-sky conditions (right block) for MACv2-SP with MACv1 natural background (upper left) and with MACv2 natural background (lower left), for MACv2 (upper right) and MACv1 (lower right). Blue colors represent climate cooling and red colors indicate climate warming. Values below the labels indicate global averages.*

For the offline simulations with the MACv2-SP data with two different natural background
conditions are assumed: one with a lower MACv1 natural background (M_SP_1) and one with a higher
MACv2 natural background (M_SP_2).
The MACv2-SP plume approximation yields MACv2 similar cooling patterns for present-day
anthropogenic aerosol. However, MACv2-SP maxima are stronger (e.g. China) and weaker (e.g. South
America). In the MACv2-SP plume approximation, contributions in remote regions are lower or
completely missing, as individual plumes with limited spatial domains (even when combined) do not
cover the entire globe. The impact by using different natural backgrounds in MACv2-SP (once using
MACv1 natural aerosol and once using MACv2 natural aerosol) has only a small impact on the direct
forcing. Globally averaged, both clear-sky and all-sky TOA cooling in MACv2-SP are ca 10% smaller than
in MACv2, mainly as the mid-visible global average mid-visible anthropogenic AOD in MACv2-SP is only
0.028 compared to 0.031 in MACv2.
In the MACv1 climatology regional contributions for the direct forcing differ from MACv2 mainly
due to differences in anthropogenic AOD regional strengths. MACv1 falsely allowed significant
anthropogenic aerosols over the Sahara (associated with significant warming over the bright desert)
along with too much anthropogenic aerosol over the US and Europe (for extra cooling). In MACv1 also
anthropogenic aerosol over SE Asia is likely too low (for missing cooling). Globally averaged though, the
direct forcing between MACv2 and MACv1 are similar as effects of a larger anthropogenic AOD of 0.040
in MACv1 (compared to 0.031 in MACv2) are compensated by significant warming over the Sahara.
Annual maps for today's indirect effects at the TOA by anthropogenic aerosol are compared in Figure B2.

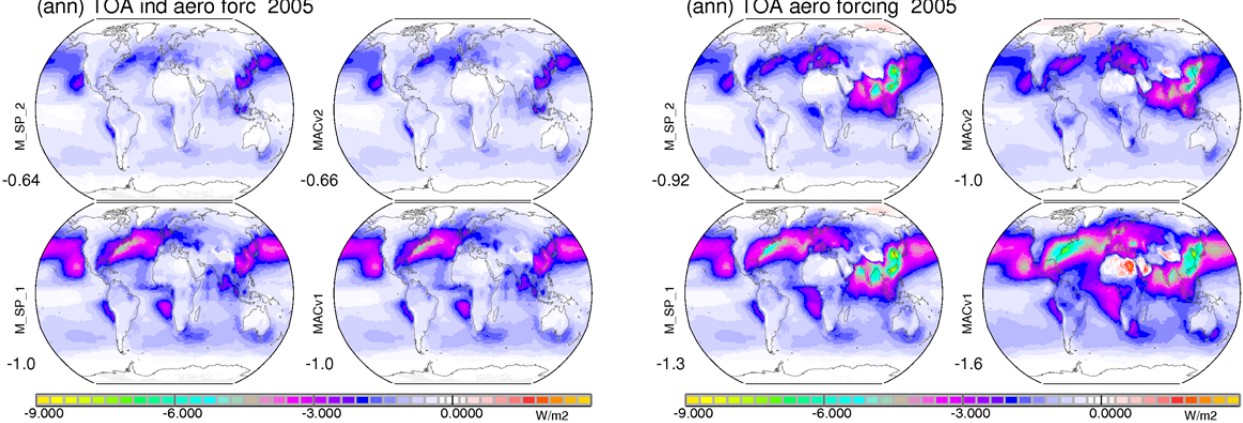

*Figure B2. annual maps for present-day aerosol indirect radiative forcing in W/m2 at the TOA at (left*
*block) and for the combined direct and indirect radiative forcing (right block) for MACv2-SP with MACv1*
*natural background (upper left) and with MACv2 natural background (lower left), for MACv2 (upper*
*right) and MACv1 (lower right). Values below the labels indicate global averages.*
In all approaches, only indirect (Twomey) cooling effects at the TOA are considered and
approximated based on the 'observed' satellite relationship between AODf and CDNC as explained in
Appendix A. The fit function is applied twice, for natural AODf and for total AODf (anthropogenic plus
natural), to extract a CDNC increase factor. In MACv2 and MACv1 required AODf values (natural AODf
and total AODf) are defined by the climatology. In MACv2-SP only anthropogenic AODf maps and
already pre-calculated associated CDNC increase factors are provided with simplified assumptions to the
natural AODf background (*Stevens et al., 2017*).
In MACv2-SP the anthropogenic AOD has no global coverage so that in regions with no
anthropogenic AOD also no indirect aerosol effects are possible. This is the main reason that the globally
averaged indirect TOA cooling in MACv2-SP is only ca 65% of the indirect TOA cooling by MACv2 (despite
a much stronger MACv2-SP indirect response over the Pacific). For MACv1, in contrast, the indirect TOA
cooling is ca 40% larger than in MACv2. The main reason here is the larger anthropogenic AODf in
combination with a reduced natural AODf. Both factors lead to larger CDNC increase factors and
stronger indirect TOA cooling when applying the (retrieval based) fit function.
In Figure B2 also annual maps for the predicted combined (direct and indirect) TOA cooling by
today's anthropogenic AOD are compared. The MACv2 climatology suggests (when globally averaged) a
combined present-day climate cooling near -1.0 W/m2, with the larger contribution from indirect
effects. The associated uncertainty involves a -0.7 to -1.6 W/m2 range a for the present-day aerosol
forcing.  To lower that range a more certain the pre-industrial reference and a better representation of
indirect effect are needed.
The main argument for more negative bound (-1.6 W/m2) is that the anthropogenic fine-mode
fraction could be larger.  An alternate anthropogenic scaling based on the AeroCom phase 1 emission
increases indirect effects (from -0.65) to -1.1 W/m2 and direct effects (from -.35) to -0.50 W/m2 for a
combined -1.6W/m2 cooling. However, the alternate larger anthropogenic fine-mode fraction applies a
year 1750 reference. Thus, such a negative aerosol forcing is very unlikely for a year 1850 reference.
The main argument for the less negative bound (-0.7W/m2) is that the lower bound of the direct
effect (which is better constrained than the indirect effect) is estimated at -0.2 W/m2 and that the lower
bound for the indirect effect is estimated at -0.5 W/m2. For the present-day direct effect a less negative
-0.2 W/m2 cooling can be explained with a higher anthropogenic fraction of absorbing BC AOD than
applied for the fine-mode AOD. For the present-day indirect effect there is a lot of uncertainty. If only
the first indirect effect is considered at least a cooling of -0.5 W/m2 should be expected (in that context
it should be noted that the less negative indirect effect MACv2-SP does not qualify as lower bound due
to its incomplete global coverage for anthropogenic AOD).
**Appendix C** *monthly TOA forcing*
Monthly maps illustrate typical variations over the year for present-day aerosol radiative effects
at the top of the atmosphere (TOA) for total aerosol in Figure C1 and anthropogenic aerosol in Figure C2.
Figure C1 presents for total aerosol the MACv2 associated radiative TOA effects. Monthly maps
are presented for clear-sky (cloud-free) and all-sky conditions. Note that the effects in Figure C1 include
infrared greenhouse effects by elevated coarse mode dust. For total aerosol, the global average effect is
a climate cooling, despite intense climate warming over the Sahara, especially from April to September.
The presented aerosol effects at all-sky conditions do not include anthropogenic indirect effects which
would increase the TOA cooling by an additional -0.65W/m2.

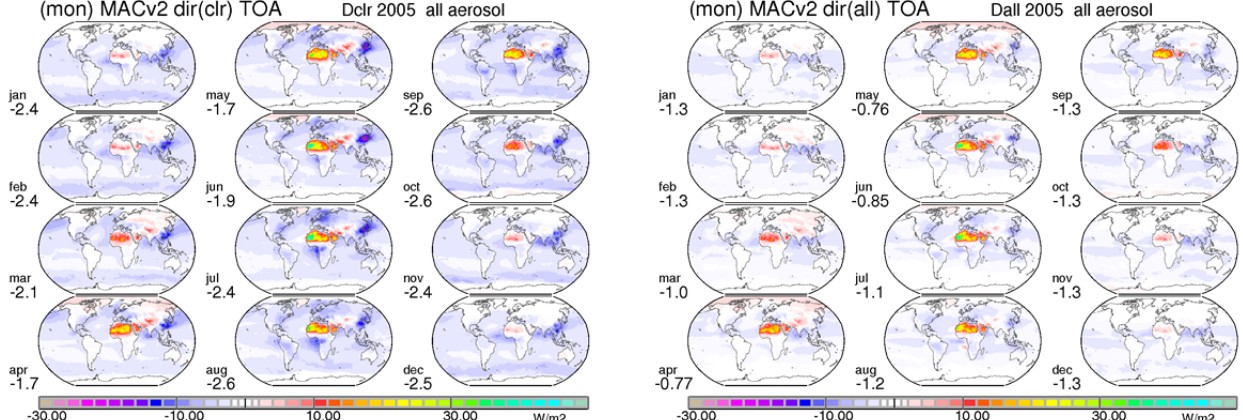

**Figure C1.** *monthly maps for the direct radiative forcing at the TOA by present-day total aerosol under*
*clear-sky conditions (left block) and all-sky conditions (right block). Blue colors indicate climate cooling*
*and red colors indicate climate warming. The infrared greenhouse effects by elevated coarse-mode*
*mineral dust aerosol are included. Values below the labels indicate global averages.*

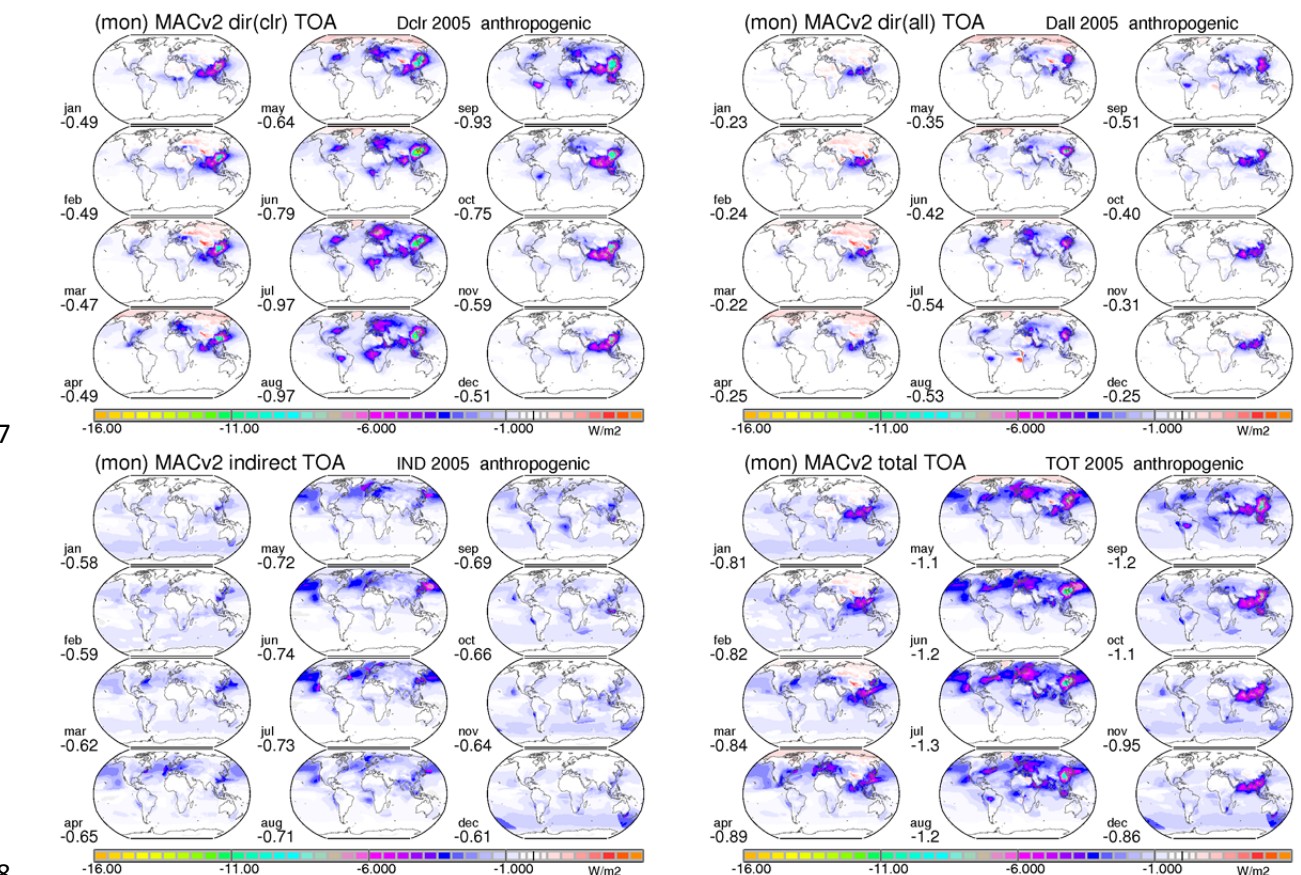

**Figure C2.** *monthly maps for radiative effects by present-day anthropogenic aerosol. Compared are*
*direct radiative forcing by today's anthropogenic aerosol under clear-sky conditions (top, left block),*
*direct radiative effects under all-sky conditions (top, right block), indirect effect (bottom, left block) and*
*the combined (direct and indirect) effect (lower, right block). Blue colors indicate climate cooling and red*
*colors indicate climate warming. Scales of all four blocks are identical to better compare contributions*
*and seasonal dependencies. Values below the labels indicate global averages.*
Figure C2 presents for anthropogenic aerosol the MACv2 associated radiative TOA effects.
Monthly maps are presented for the direct effect at cloud-free condition (Dclr) and all-sky conditions
(direct forcing, Dall), for a (Twomey based) indirect forcing (IND) and for the combined (direct and
indirect) forcing (TOT). In Figure C2 monthly maps were given the same scale for easier comparisons.
They illustrate that for present-day aerosol forcing indirect contributions (IND) dominate over direct
contributions (Dall) and that contributions are largest for the NH summer season with indirect effects
globally slightly larger from May to August and direct effects globally largest from July to September.
8          In terms of the aerosol radiative forcing some applications prefer to separate between
clear-sky and cloudy-sky contributions. A requirement for such a separation is the cloud-free fraction of
Figure C3. Figure C3 also presents the clear-sky present-day contributions to the (direct) all-sky or to the
combined (direct +indirect) forcing with MACv2. The corresponding cloudy-sky contributions to the
present-day (direct) all-sky and to the combined (direct + indirect) forcing are presented in Figure C4.

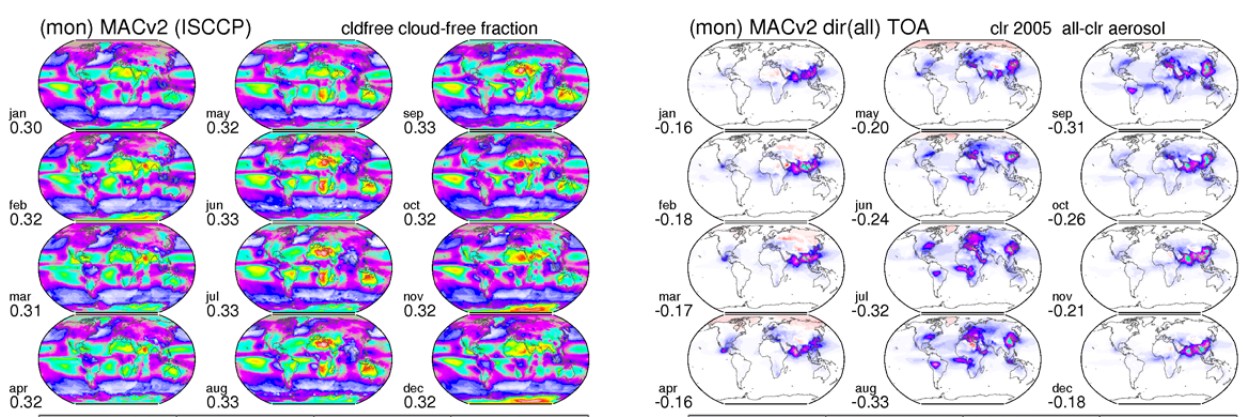

*Figure C3. monthly maps for the clear-sky fraction (left block) and monthly maps for the present day*
*clear-sky radiative TOA radiative forcing that contributes to the all-sky forcing.*

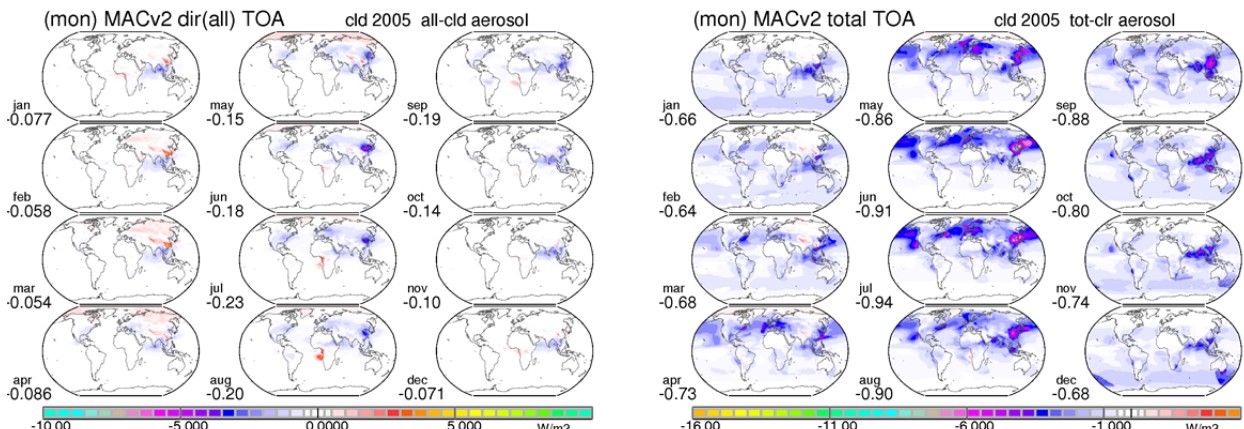

*Figure C4. monthly maps for monthly maps for the present-day cloudy-sky TOA radiative forcing that*
*contributes to the all-sky forcing (left block) and to the combined (direct and indirect) forcing.*

The radiative forcing in cloudy regions (both with and without the first indirect effect) is much
larger during the boreal summer season compared to the boreal winter. The absolute global average
seasonal range is similar for the direct and for the indirect effect, with more sunshine, less low altitude
cloud decks and more anthropogenic aerosol loads during higher latitude summer seasons.

# Appendix D  *monthly TOA direct forcing efficiencies*

Monthly maps for present-day TOA forcing efficiencies (per unit AOD) at clear-sky and all-sky
conditions are shown for total aerosol in Figure D1 and for anthropogenic aerosol in Figure D2.

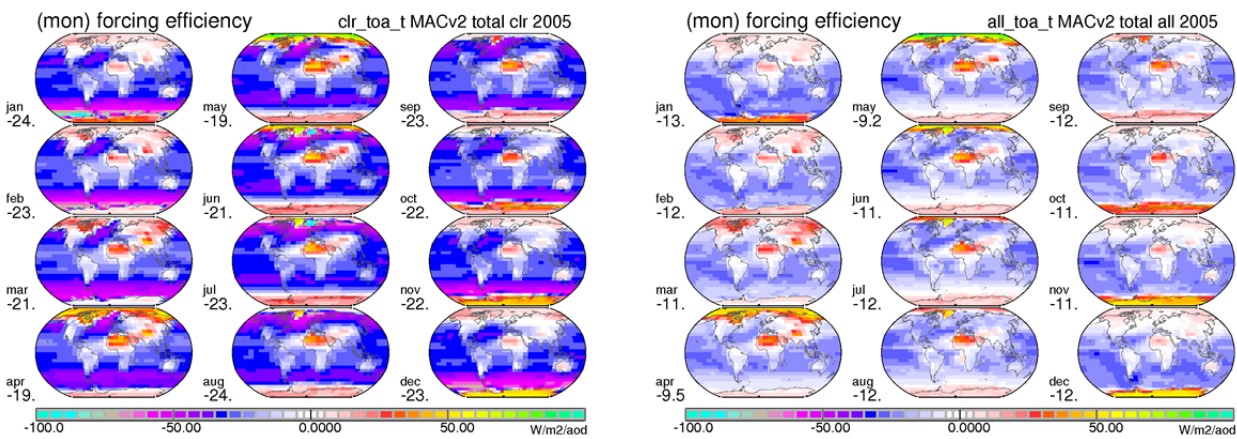

*Figure D1. monthly maps for the direct radiative forcing efficiency (per unit AOD) at the TOA by present-day total aerosol at clear-sky (left block) and all-sky conditions (right block). Blue colors indicate climate cooling potential and red colors warming potential. Values below the labels indicate global averages.*

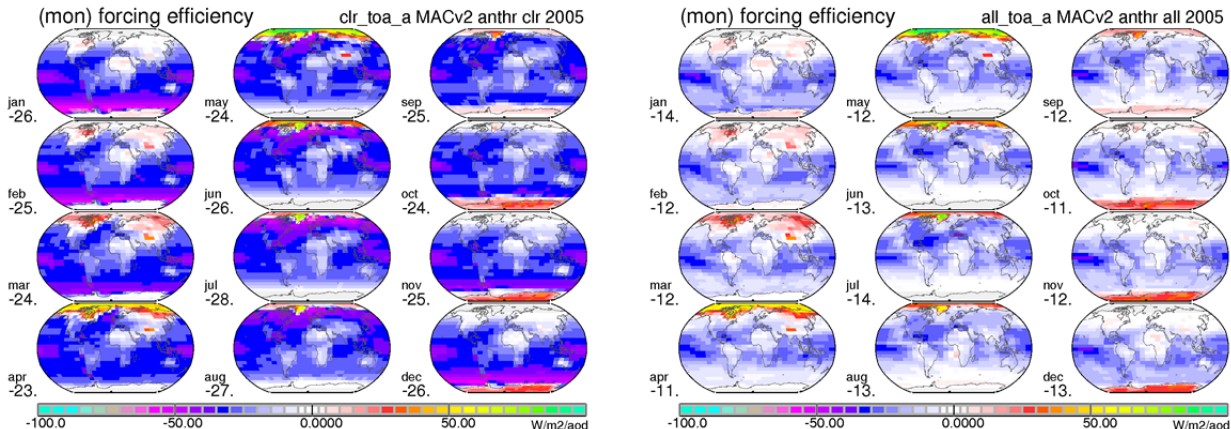

*Figure D2. monthly maps for the direct radiative forcing efficiency (per unit AOD) at the TOA by present-day anthropogenic aerosol at clear-sky (left block) and all-sky conditions (right block). Blue colors indicate climate cooling potential and red colors warming potential. Values indicate global averages.*

The global average forcing efficiencies (per unit AOD) for total and anthropogenic aerosol are
similar and not only in their global averages (near -22 W/m2 at clear-sky conditions and near -11W/m2
at all-sky conditions) but also their spatial patterns relatively stable over the year. Noteworthy is the
switch in sign over Asia to positive values during the winter and spring due to snow cover and only for
total aerosol the strong positive values over northern Africa.

## Appendix E  *component direct radiative effects*

With the attribution of optical and microphysical properties to (via size and refractive index)
pre-defined aerosol types, type (or component) contributions are assigned such that their mixture is
consistent with (local monthly data for) size-mode associated MACv2 mid-visible aerosol properties for
AOD and AAOD. The resulting MACv2 component AOD distributions are presented in Figure E1.

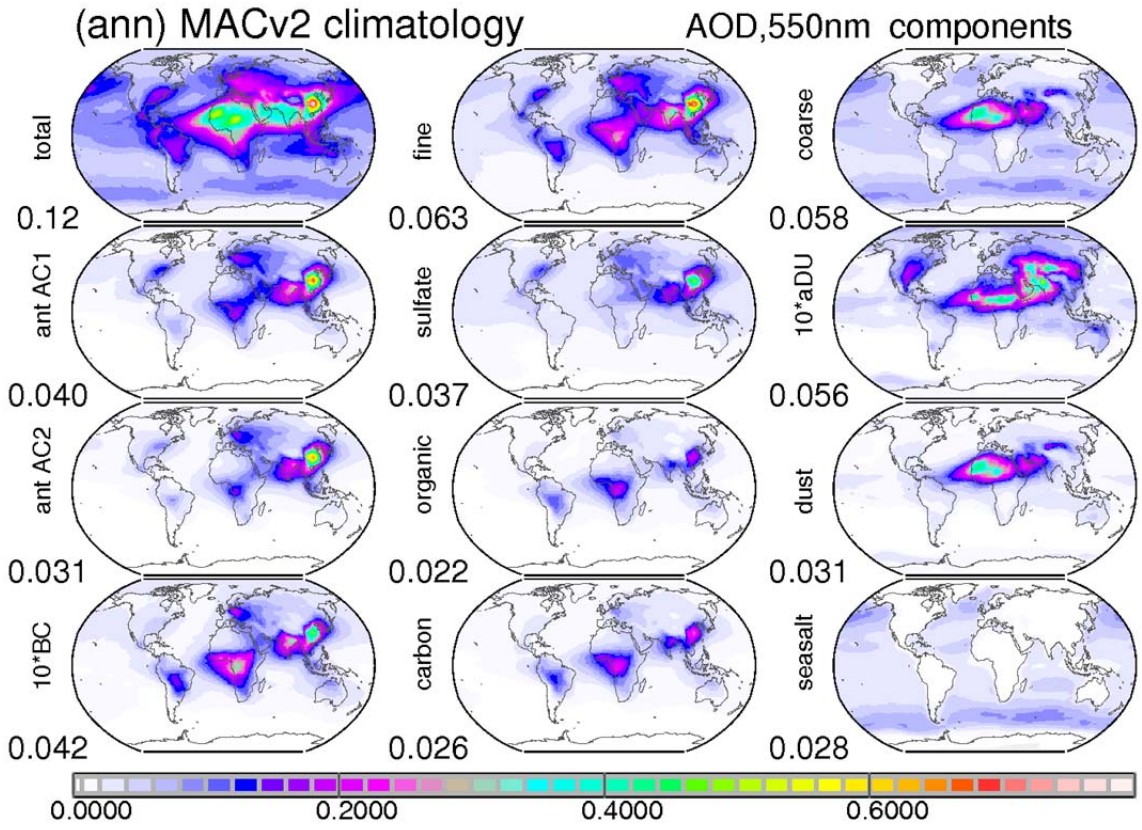

*Figure E1 annual average AOD maps for present-day tropospheric aerosol for total aerosol (top left) and*
*contributions by fine-mode sizes (top center) and coarse-mode sizes (top right). In addition, consistent*
*with mid-visible absorption data, component AOD values were assigned for each size mode. Fine-mode*
*AOD is divided into contributions by BC (soot, here multiplied by 10), OC (organic matter) and SU (where*
*SU represents non-absorbing fine-mode aerosols). The coarse mode AOD is split into contributions by*
*sea-salt and dust. In addition, annual AOD maps are presented for total carbon (OC+BC), for present day*
*anthropogenic dust ('aDU', here multiplied by 10) and estimates for present day anthropogenic (fine-*
*mode) AOD: 'ant AC2' of MACv2 and 'ant AC1' of MACv1. Lower left values indicate global averages.*
Considered components for the fine-mode are sulfate (SU - representing the fine-mode non-
absorbing type), organic matter (OC), soot (BC) and for the coarse-mode sea-salt (SS) and mineral dust
(DU). Hereby the size for SU and DU is allowed to vary, to satisfy MACv2 prescribed data for the fine-
mode effective radius and the coarse mode absorption, respectively (*Kinne, 2019*). Also presented are
two present-day estimates for (fine-mode) anthropogenic AOD based on fine-mode fraction scaling with
processed emissions in global models [=(AODf,pd -AODf,pi) /AODf,pd)]. One estimate (antAC1) applies a
fine-mode AOD fraction map based on AeroCom phase 1 emission data (*Dentener et al., 2006*), which is
tied to a year 1750 reference and applied in MACv1 (*Kinne et al, 2013*). The other estimate (antAC2)
applies a fine-mode AOD fraction map based on CMIP5 (AeroCom phase 2) emission data (*Lamarque et*
*al., 2010*), which is tied to a year 1850 reference and applied in MACv2 (*Kinne, 2019*). In addition, an
estimate for present day anthropogenic (coarse-mode) dust (aDU) is presented by applying a present
day anthropogenic dust AOD fraction map based on a satellite data analysis (*Ginoux et al, 2012*) to the
dust AOD of MACv2. This AOD for anthropogenic dust (multiplied by 10 in Figure E1) is an average
almost an order of magnitude smaller than estimates for the fine-mode anthropogenic AOD. And since,
in addition, solar and infrared contributions to the radiative forcing by dust largely cancel, coarse-mode
anthropogenic contributions were ignored for MACv2.  Still, even with anthropogenic dust, the present-
day fine-mode anthropogenic AOD (0.03 to 0.04) is highly uncertain due to differences in choices for
reference strength, including the reference year.
**SU** *(non-absorbing fine-mode)*
21          Annual maps by present-day total and anthropogenic non-absorbing fine-mode aerosol (SU - to
cover impacts mainly from sulfate and nitrate, but also from small size sea-salt) are shown in Figure E2.
Anthropogenic SU direct forcing (-.41 W/m2) is unevenly distributed and stronger near sources. Local
radiative effects at TOA and surface are basically identical, due to a lack in relevant solar absorption.

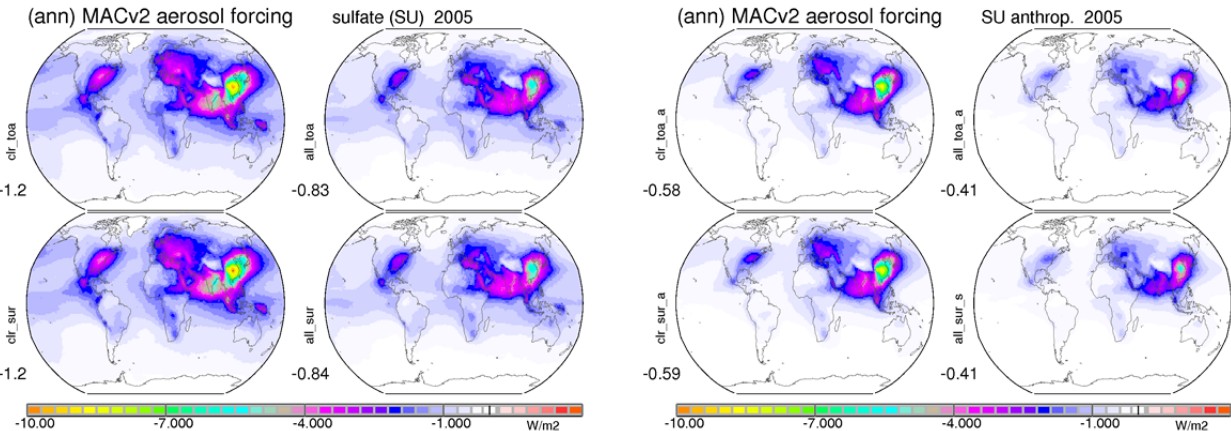

*Figure E2. Annual maps for present-day radiative effects of total (left block) and anthropogenic (right*
*block) are shown for scattering fine-mode aerosol (SU). Each block displays impacts at clear-sky (left*
*column) and all-sky conditions (right column) at both TOA (top row) and surface (bottom row). Blue to*
*purple, green, yellow and red colors indicate an increasing cooling. Values at labels are global averages.*

 **OC** *(organic matter)*

2         Annual maps by present-day total and anthropogenic organic matter are presented in Figure E3.

Anthropogenic OC direct forcing (-.22 W/m2) is unevenly distributed and relative strong near sources of
pollution and wildfire regions. Due to assumed solar absorption in the visible and especially towards and
in the UV, the cooling at the surface is larger than the cooling at TOA.

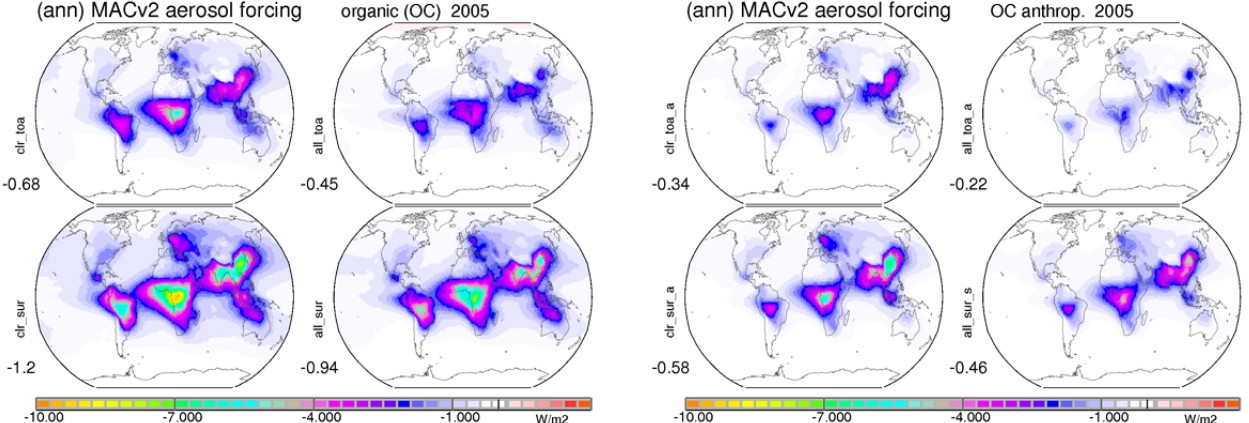

***Figure E3.*** *annual average maps for present-day radiative effects of total (left block) and anthropogenic*
*(right block) organic matter (OC). Each block displays impacts at clear-sky (left column) and all-sky*
*conditions (right column) at both TOA (top row) and surface (bottom row). Blue to purple, green and*
*yellow indicate an increasing cooling. Values below the labels are global averages.*
**BC** *(soot)*

15         Annual maps for TOA warming and surface cooling by present-day total and anthropogenic soot

(BC) are shown in Figure E4. Anthropogenic BC direct radiative effects are larger over wildfire and
pollution regions. TOA warming with clouds (all-sky) is larger, as BC dims the reflection of lower clouds.

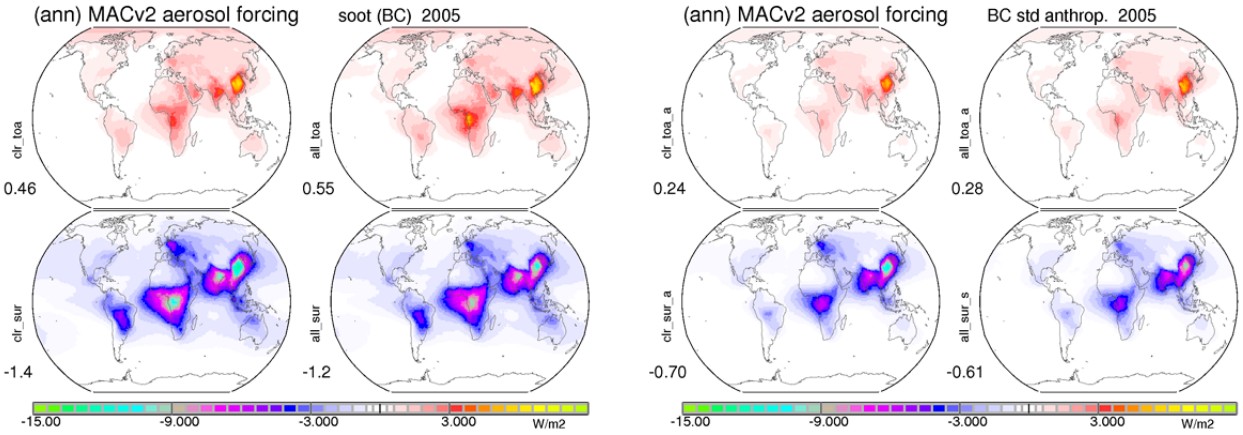

***Figure E4.*** *Annual maps for present-day radiative effects of total (left block) and anthropogenic (right*
*block) are shown for soot (or BC). Each block displays impacts at clear-sky (left column) and all-sky*
*conditions (right column) at both TOA (top row) and surface (bottom row). Blue colors indicate a cooling*
*(at the surface) and red colors a warming (at TOA). Values below the labels are global averages.*
The seasonality of the TOA all-sky BC forcing and the BC radiative effects for an alternate larger
(75% rather 50%) for the BC anthropogenic fraction are presented in Figure E5.

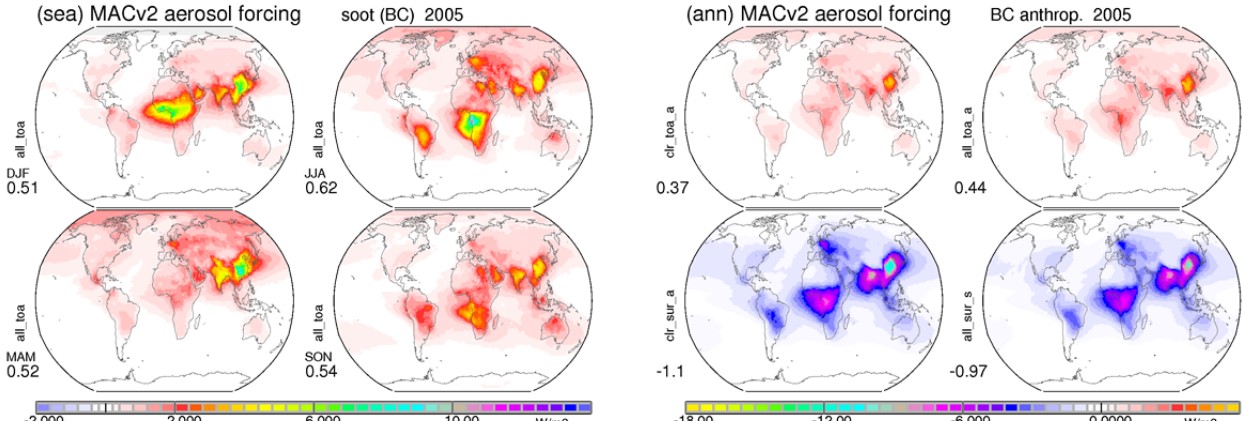

***Figure E5*** *Seasonally average maps for present-day BC radiative TOA all-sky effects at (left block) and*
*anthropogenic BC radiative effects with a higher (75%) BC anthropogenic fraction compared to the 50%*
*for BC radiative effects presented in Figure E4. Values below the labels are global averages.*
Many regional aerosol absorption maxima have a strong seasonal flavor often associated with
wildfire seasons (e.g. DJF over W. Africa, JJA and SON over central Africa and S. America) and pollution
(e.g. MAM over S. Asia and E. Asia prior to the monsoon season). The applied larger anthropogenic BC
fraction raised the TOA forcing by BC from +0.28 to +0.44W/m2.
**CA** *(carbon: OC+BC)*
Annual maps for present-day direct radiative effects by total and anthropogenic carbon are
shown in Figure E6. The anthropogenic carbon TOA response (+.05 W/m2) is almost climate neutral.

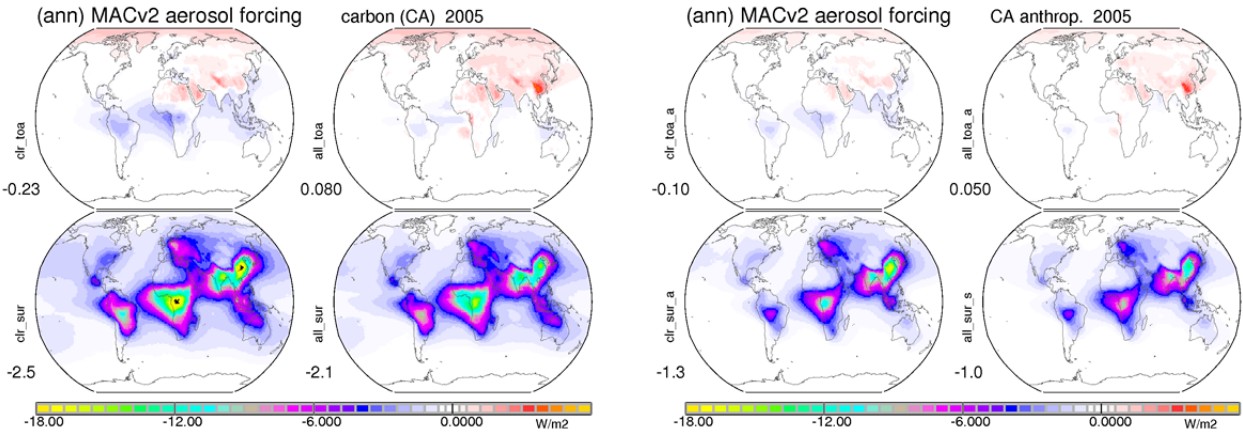

***Figure E6.*** *Annual maps for present-day radiative effects of total (left block) and anthropogenic (right*
*block) soot (or carbon (BC+OC). Each block displays impacts at clear-sky (left column) and all-sky*
*conditions (right column) at both TOA (top row) and surface (bottom row). Blue colors indicate a cooling*
*(at the surface) and red colors a warming (at TOA). Values below the labels are global averages.*
**SS** *(sea-salt)* and **DU** *(dust)*
Annual maps for direct radiative effects by coarse mode (natural) components of sea-salt and
mineral dust are compared in Figure E7. For sea-salt cooling maxima at TOA and surface are sharply
reduced with the presence of clouds. Thus the direct forcing associated with sea-salt is relatively small.
Mineral dust displays s strong TOA warming over the Sahara. Even at the surface a net-flux increase is
indicated. Both effects are associated with relatively large mineral dust AOD and dust particles sizes.
Mineral dust effects are better understood by looking at solar and infrared contributions in Figure E8.

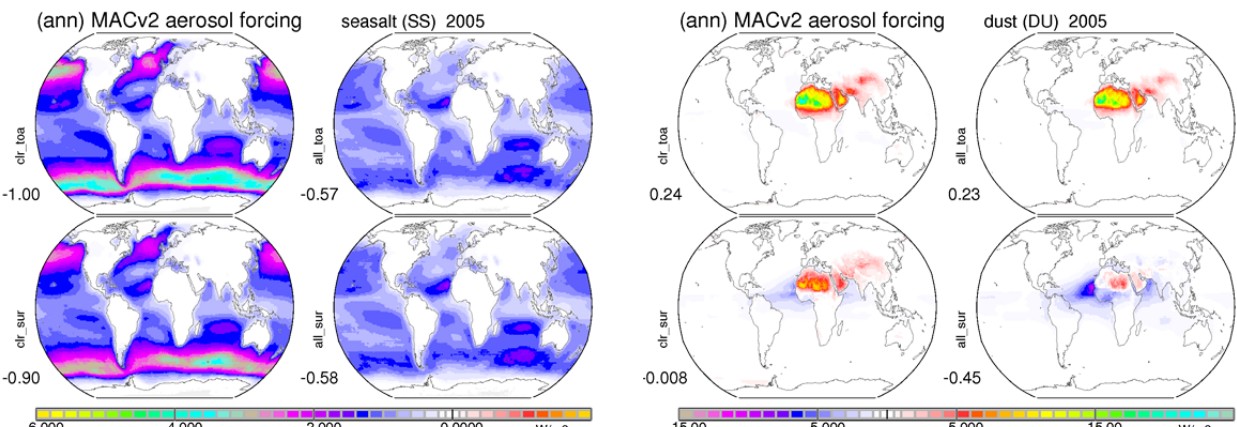

***Figure E7.*** *Annual maps for present-day TOA radiative effects of coarse mode sea-salt (SS, left block) and*
*coarse mode mineral dust (DU, right block) at clear-sky (left column) and all-sky conditions (right*
*column) both at TOA (top row) and surface (bottom row). Blue to purple colors indicate a cooling and red*
*to yellow and green colors show a warming. Values below the labels are global averages Note, that the*
*color scale for sea-salt is 10 times smaller than the color scale for dust.*
The mineral dust radiative effects are created by partially offsetting solar cooling and infrared
warming effect. Solar warming is only created by larger dust sizes over bright (Sahara) surfaces.

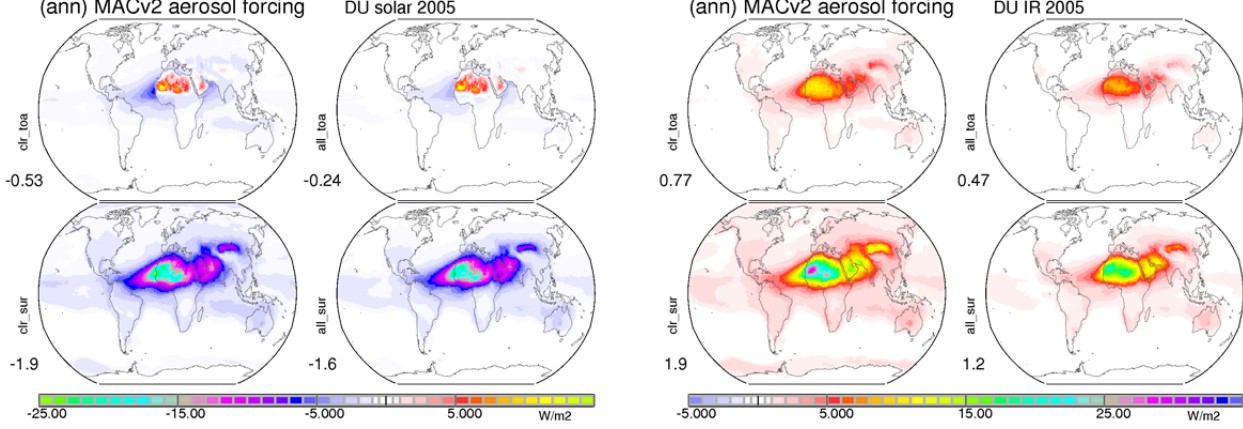

***Figure E8.*** *annual averages maps for present-day radiative effects of coarse mode mineral dust (DU) in*
*the solar spectral (left block) and the infrared spectral region (right block). Aerosol impacts are shown at*
*clear-sky (left column) and all-sky conditions (right column) for both TOA (top row) and surface (bottom*
*row). Blue colors indicate a cooling and red colors a warming. Values at the labels are global averages.*
For the climatic relevant TOA response at the top of the atmosphere the present-day global
average is a -0.24 W/m2 cooling, despite the regional solar warming over the Sahara (as stronger
absorbing larger mineral dust sizes dim the bright desert solar albedo). The infrared greenhouse effect
at +0.47 W/m2 is much larger, so that the global average response for mineral dust is a +0.23 W/m2
warming. The greenhouse effect looses (on average) its dominance for anthropogenic dust, as
anthropogenic aerosol involves more moderate sizes. And also anthropogenic aerosol loads are closer to
the surface. Thus extra anthropogenic dust on average behaves almost climate neutral.
# Appendix F  *aerosol indirect radiative forcing sensitivity*
Aerosol indirect radiative effects show spatial distribution patterns, which may surprise, as the
strongest impacts (e.g. mainly the increase in planetary albedo) are usually not at locations where
anthropogenic AOD values are largest. The aerosol indirect effects or forcing is strongly influenced by
environmental properties. A lower surface albedo, a high percentage of low altitude cloud cover without
higher altitude clouds, a moderate optical depth for highest susceptibility and the available sun-hours
during a day all favor a stronger indirect response. Based on these factors  as summarized in Table F1
monthly indirect forcing potential maps are presented in Figure F1 along with monthly maps for MACv2
indirect forcing efficiency per unit anthropogenic AOD.

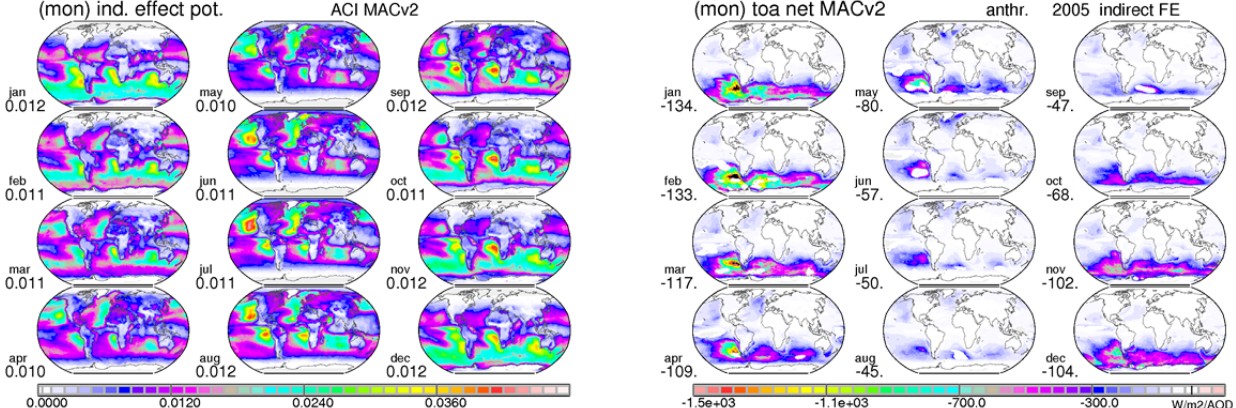

***Figure F1*** *monthly maps for environmental potential for aerosol indirect effects (left block) considering*
*low cloud cover, reflection and susceptibility, surface albedo, sunshine hours and sun elevation and the*
*indirect forcing efficiency (right block).*
The highest environmental potential for indirect effects is at oceanic stratocumulus regions off
western continental coasts in the subtropics and at mid-to-high latitude oceanic regions during spring
and summer seasons with longer sunshine periods. The aerosol indirect forcing efficiency (indirect
forcing per unit anthropogenic AOD) displays a very high sensitivity over the southern oceans, where
anthropogenic contributions, however, are very low. As this high sensitivity is caused by the part of the
logarithmic fit that is not well constrained by observations (see Appendix A) indirect effects are possibly
overestimated over aerosol sparse regions, although absolute contributions to the MACv2 indirect
forcing from these very clean regions are relatively small.
*Table F1*. *assumed properties for the indirect environmental potential*

| Property | Assumptions |
|---|---|
| cloud top temperature weight wei | >270K full impact (wei=1)     <250K no impact (wei=0) |
| solar cloud reflection based R,cld | Rcld = (1 –exp(COT/(10+.35*COT))        COT-cloud optical depth |
| Susceptibility Scep | Scep = R,cld *(1.0 - R,cld) |
| Rcld, max is 0.67 | 0.67 |
| consider the background factor B,f | B,f = 1.0-surf.albedo |
| low only cloud frequency F,low | F,low |
| scatter strength /sun-hours | sum {time(u0))*{1.0-abs(u0max-u0))**1.7} *exp(-0.025/u0**1.5)]}<br>                         *with*         u0max =0.25 +0.75*(1-exp(-0.22*cot**1.25) |
| **environmental indirect potential** | **=  wei *Scep *.67 *B,f *F,low *sun** |

# Appendix G *comparisons to other published aerosol direct radiative effects*

11          The present-day aerosol properties and associated direct radiative effects of MACv2 are here
compared to complementary results from 'bottom-up' global modeling. More detail is provided by
investigating component contributions in 'bottom-up' global modeling with assigned component data of
the 'top-down' approach in MACv2.
15          First, mid-visible AOD and AAOD differences between a MACC reanalysis of assimilated MODIS
AOD data in an ECMWF simulation (*Bellouin et al., 2013*) to MACv2 are investigated by comparing
annual averages in Table G1.
*Table G1*. *comparison of properties for present-day mid-visible (550nm) AOD and AAOD data between*
*the MACv2 aerosol climatology and the MACC reanalysis data.*

| | AOD | | | | | AAOD | | | |
|---|---|---|---|---|---|---|---|---|---|
| | *total* | *DU* | *SS* | *fine* | *anthr* | *total* | *DU* | *fine* | *anthr* |
| **MACv2** | 0.121 | 0.031 | 0.028 | 0.063 | **0.031** | 0.0072 | 0.0021 | 0.0051 | **0.0030** |
| **MACC** | 0.180 | 0.043 | 0.055 | 0.081 | **0.073** | 0.0080 | 0.0010 | 0.0070 | **0.0070** |

25          With the MODIS data-assimilation in MACC (*Bellouin et al., 2013*) the global average AOD is 50%
higher. Thus is a likely overestimate as updated MISR retrievals (*M.Garay, personal communication*) and
even the ICAP satellite assimilation ensemble (*Peng et al., 2013*) indicate and upper ceiling of 0.14 for
the present-day global mid-visible AOD. The (for anthropogenic impacts relevant) fine-mode AOD MACC
has a smaller relative contribution of the total AOD but is still larger than in MACv2. More of a concern is
the large anthropogenic fraction of the fine-mode AOD in MACC, so that the present-day anthropogenic
AOD is more than 2.5 times larger than in MACv2 - as all wildfires are incorrectly considered as
anthropogenic in MACC.  The MACC anthropogenic AOD has a slightly smaller absorption potential and
the absorption potential of mineral dust is way too small, mainly due to a simplified dust size treatment
in MACC. Differences in associated aerosol radiative effects are expected and shown in Table G2.
*Table G2*. *comparison of present-day aerosol associated aerosol radiative effects between the MACv2*
*aerosol climatology and the MACC reanalysis data (in W/m2)*

| | clear-sky TOA solar only direct radiative effect | | | | | direct total  forcing | |
|---|---|---|---|---|---|---|---|
| | *total* | *DU* | *SS* | *fine* | *anthr* | *anthr* | *corrected* |
| **MACv2** | -3.5 | -0.53 | -0.91 | -1.5 | -0.70 | -0.36 | |
| **MACC** | -7.3 | -1.6 | -2.8 | -2.8 | -2.5 | -0.70 | -0.40 |

13        The MACC total aerosol present-day effect is too large, mainly because the total AOD is too
large but also as the stronger cooling coarse-mode (missing mineral dust absorption/size) has a larger
AOD fraction (too much seasalt).
16        The MACC aerosol present-day direct forcing of -0.7 W/m2 is too large as it includes wildfire
contributions (*N. Bellouin, personal communication*). When removing those contributions the aerosol
forcing is corrected downward to an agreeable -0.4 W/m2 cooling. Still, the contributing large reduction
from clear-sky to an all-sky (from -2.5 to -0.7) in MACC raises proper cloud impact treatment questions.
20        Focusing on the fine-mode aerosol an its anthropogenic contributions the present-day aerosol
direct forcing, along with component contributions of MACv2 was compared to 'bottom-up' ensemble
averages of AeroCom (*Schulz et al., 2006*) and CMIP5 modeling (*Myhre et al., 2013*) in Table G3.
*Table G3*. *comparisons of present-day aerosol direct radiative forcing (TOA, all-sky) in W/m2*

| | total | non-absorbing | | absorbing | | | |
|---|---|---|---|---|---|---|---|
| | | SU | NI | CA | BC | OC | SOA |
| **MACv2** | -0.36 *(to -0.20)* | -0.41 | | +0.05 *(to +0.22)* | +0.28 *(to +0.44)* | -0.23 | |
| **Myhre** avg | -0.32 | -0.32 | -0.08 | +0.09 | +0.18 | -0.03 | -0.06 |
| **Myhre** med | - 0.28 | -0.32 | -0.08 | +0.13 | +0.18 | -0.03 | -0.02 |
| **Schulz** | -0.22 | -0.32 | | +0.11 | +0.25 | -0.14 | |

*MACv2 values with higher than fine-mode present-day anthropogenic BC fractions are in parenthesis*
30        There is surprising good agreement for direct forcing estimates, even on a component basis. It is
also shown, that if the prescribed fine-mode anthropogenic fraction in MACv2 is raised for soot (BC),
then extra BC warming shifts the overall aerosol direct forcing to less negative values in MACv2. Thus,
the possibility of a lower MACv2 direct effect cannot be ruled out. Here again, it is the limited
understanding of the pre-industrial reference that introduces uncertainty.