# Peer review of "Aerosol radiative effects with MACv2"

_Atmospheric Chemistry and Physics, 2018_

## Referee Comment (RC1) · Anonymous Referee #2 · 15 Mar 2019

The author describes the application of the MACv2 global aerosol climatology to calculating direct and indirect aerosol radiative effects and forcing, decomposed by aerosol composition, in both the LW and SW. Maps and time series of the different effects are presented. The author estimates a total aerosol forcing of -1Wm-2 with a possible range of -0.7 – -1.6 Wm-2. They also show a relatively constant forcing over the last decades despite a large regional shift due to shifting emissions.

Overall the paper is well structured but a number of issues need rectifying before I would consider it suitable for publication in ACP:

- The paper describes the forcing responses of this particular setup comprehensively, and would be well suited for GMD. In order for it to meet the scope of ACP however I would suggest much more discussion around how these estimates are an improvement on existing e.g. AeroCom estimates, particularly in the context of the many other

studies and efforts in this vain, both modelling and observational.

- In particular, the paper aims to provide an estimate of the aerosol forcing, but ignoring cloud adjustments seems a significant omission and requires much stronger justification, particularly when discussing potential uncertainties. For example, on P2L9-10 the author states that feedbacks (adjustments) can be considered secondary, though the provided citation makes no such assertion – they just assume it for their purposes.

- The first indirect (Twomey) effect is accounted for, but the fittings used to extract the sensitivity from satellite retrievals is non-standard and needs much more justification and discussion. Why was this functional form chosen, how were the parameters chosen, and what are their uncertainties?

- The color scales used for the plots are very difficult to interpret and at times even use the same color for different values. Worse, these non-monotonic colormaps can also distort the impression of the values being plotted (see e.g. https://matplotlib.org/users/colormaps.html and references therein) and should be replaced.

- Also, the manuscript can be difficult to read at times and would benefit from editing help from someone with full professional proficiency in English.

I also have a number of other, more minor, suggested changes:

- P1 L10: '...major aerosol indirect effect. . .' should be 'first aerosol indirect effect'

- P1 L12: 'locally' -> 'local'

- P2L13-14: The author claims that this setup avoids 'time-consuming aerosol processing' though the fields used to create the MACv2 climatology are derived from just these costly models. This should be clarified.

- P2L19: Suggest 'before impact results' be replaced with 'before forcing results'

- P3L13: The use of the AeroCom phase 1 models is introduced here, but later on

some phase 2 values are also used and should be introduced as well.

- P3L20: Shouldn't all the terms in the scaling equation related to AODf? Currently some relate to AOD.

- P3L20: IPCC5 -> CMIP5

- P6L21: The description of double radiative calls is confusing, please consider re-phrasing. The author should also probably cite Ghan 2013.

- P7L5: The assertion that feedbacks are on the order of 10% requires a citation. The author should also acknowledge that there is considerable uncertainty around this.

- P8 Fig5: Perhaps space out the maps a bit more vertically so that it's clearer which mean values apply to which maps. The 'anth all_sky' toa probably doesn't need scaling either, it's misleading when glancing at the data. Perhaps put the TOA and SURF plots on different scales? It should also be labelled as 'all_toa_a' for consistency.

- P9L3-4: The assumption of ignoring potential anthropogenic course mode aerosol should be discussed in the introduction.

- P10 Fig7: These fields appear to be presented at a degraded resolution compared to the previous ones, are they not calculated on the same grid?

- P10L23-25: The assumed anthropogenic dust contribution should be introduced in the introduction section. It's not clear when this is included and when it isn't.

- Fig 8.: It's very hard to compare these maps as they are all on different scales. There is also a lot of information to try and absorb. Could the all-sky plots be removed since the discussion focusses on the different effects of coarse mode aerosol on the solar and IR bands?

- Figure 9: The .1 scalings should be written 0.1 to make them clearer. It's also impossible to compare the magnitudes of cooling and warming using these color scales, a usual blue->red scale would be much clearer.

- P12L26: 'tome' -> 'time'

- P13L6: "also have to considered" -> "also have to be considered"

- Table 2: The coarse mode dust is used here again, it needs introducing and the introductions of anthropogenic aerosol only contributing to AODf needs clarifying

- P17L7: The assumption of constant LWC affects the size of their droplets, and hence their reflectivity, not the other way around. The sentence should be altered to make this clearer. Something like: ". . .that the resulting water cloud droplets are more numerous and smaller, assuming no change to the cloud liquid water content (LWC). With smaller drop sizes the solar reflection. . ."

- P17L11: ". . .then smaller. . ." -> ". . .the smaller. . ."

- P17L33: This sentence doesn't seem to make sense: "However, there is reliance, that regional associations will provide the needed link.". Please rephrase.

- P18L3: "to extract" -> "the extraction of"

- P18L4: "meant that" -> "means that"

- P19L9-11: There are many reasons why the satellite retrieved response could be biased low as well, and recent work has even suggested very large sensitivities (Rosenfeld et al. 2019). The most that can be said is that they're different for reasons still to be fully understood.

- P19L12-13: Why is a single sensitivity applied globally? The 1x1 degree sensitivities are available and are presumably very different locally. Or is the sensitivity calculated globally? In which case large errors are likely to be present (Grandey and Stier 2010). This section needs more discussion and justification.

- P20L14-16: Given these uncertainties it could presumably also be underestimated then?

- P25L4: continued -> continues

- P26L6: "As alternate background ICAP" -> "As an alternate background, ICAP"

- P26L12-14: This isn't clear. Does the ICAP ensemble include AODf and anthropogenic AOD? How does it calculate them? From which models? Does if assume the same Lamarque emissions as MACv2?

- P28L2-5: This is a confusing sentence, consider re-phrasing

- P28L9-10: The uncertainty due to the parameterisation of the Twomey effect is probably very large due to the large uncertainties in measuring AODf and CDNC individually, confounding (e.g. meteorological) factors, and the (weak) causal relationship between AOD and CDNC. There are also likely to be large uncertainties due to the choice of parameters used to fit these distributions since the error bars are large and small AODf values are not measured. Both of these should be discussed and estimated. Potentially large liquid cloud adjustments are also ignored and should be discussed.

- P32L35: matches -> matched

- P33L15: larger -> large

References:

Ghan, S. J.: Technical Note: Estimating aerosol effects on cloud radiative forcing, Atmos. Chem. Phys., 13, 9971-9974, https://doi.org/10.5194/acp-13-9971-2013, 2013

Daniel Rosenfeld, Yannian Zhu, Minghuai Wang, Youtong Zheng, Tom Goren, Shaocai Yu: Aerosol-driven droplet concentrations dominate coverage and water of oceanic low-level clouds, Science 2019

Grandey, B. S. and Stier, P.: A critical look at spatial scale choices in satellite-based aerosol indirect effect studies, Atmos. Chem. Phys., 10, 11459-11470, https://doi.org/10.5194/acp-10-11459-2010, 2010.

---

## Referee Comment (RC2) · Anonymous Referee #1 · 26 Mar 2019

The manuscript describes aerosol forcing calculations using the MACv2 climatology. The results presented are clear and interesting, which deserve publication after addressing my comments below.

Major comments

1. Necessary information about MACv2 are missing. The introduction is very short, with only two citations, one of which is the unpublished MACv2 climatology (Kinne, 2008). More details of both the MACv2 climatology and other aerosol climatologies are needed, especially with regard to how MACv2 was generated, how it is different from version 1 and how MACv2 is different from other aerosol climatologies, e.g. reanalysis products from MERRAaero (https://doi.org/10.5194/acp-15-5743-2015) and CAMS (https://doi.org/10.5194/acp-19-3515-2019).

2. Statements like "These global fields are the result of a data merging process for

mid-visible aerosol optical properties" (p.3, l.8) are too generic to be informative, and the whole paragraph that follows that statement is not specific on how the database was created. Important information that needs to be present includes how the climatology used the data mentioned, what the regional adjustments mean, how the monthly ensemble median of the 14 models was used, how the spectrally-defined properties are constructed from aerosol mixtures, etc. In essence, the manuscript lacks all information needed to understand how the climatology was constructed, e.g. how the fine mode was separated by the coarse mode, the anthropogenic from the total AOD, presented in Figure 1.

3. Comparisons with specific satellite instruments at the same or similar wavelength are also needed, e.g. estimates from AERONET, MODIS, etc. As the manuscript stands right now, the user is left to discover from figure to figure and from table to table that the climatology includes extinction, SSA, asymmetry parameter, 4 wavelengths, interannual variability from preindustrial to future (under which scenario?), a vertical distribution from a model (which?), studies of individual aerosol components, etc.

4. Brown carbon (BrC) is virtually absent from the manuscript. There is a mention of weakly absorbing organic matter, but it is unclear whether all organics are treated as absorbing, or a fraction of them is absorbing (and that fraction is BrC).

Minor comments

1. The abstract, especially lines 8-12, needs some editing for clarity; the wording is a little awkward. The wording is frequently cryptic or convoluted (e.g. p.5, l.11-13: "Despite strong regional shifts in regional maxima (from US and Europe to SE Asia) changes to the annual global averages over the last three decades are relatively small and are presently near a global average maximum"). As this is a monograph manuscript, I would propose that the manuscript could benefit from a colleague reading it through and providing some advice on specific wording.

2. The aerosol direct effect is not "aerosol presence", as stated in the abstract and

introduction. It is, instead, the extinction of radiation by aerosols.

3. Page 2, lines 9-12: The double-radiation calls do have memory from previous timesteps in model simulations, so this statement is only accurate in a per-timestep basis. For example, the modified clouds at a given timestep will result in changes in precipitation later, which will impact aerosol concentrations.

4. Page 5, last line: Is the GISS model the one used for the vertical distribution of aerosols as well (same page, l.14-15)?

5. Figure 4: what exactly is the total albedo? Integrated spectral albedo over the solar spectrum? Something else?

6. Page 6, line 17: is the change of solar zenith angle (SZA) as a function of time of day and season taken into account, or a mean day/night SZA representative for an annual mean as a function of latitude used for the calculations? This statement is not clear. What are the 9 SZA? Do they change with season? Are there 9 calculations, one per time of day, or just one calculation per average SZA conditions?

7. Page 6, last line: For the indirect effect, there is no change in cloud droplet number concentration assumed? Just the size? And how much is the size reduced, a fixed value or depending on conditions? How is 3d specific humidity taken into account, via model output, or?

8. Page 7, line 5: Please provide a reference that supports this statement.

9. Figure 5: Is the anthropogenic column for the total or the solar effect? Also, I believe showing the thermal effect alone would be of value, as one additional column. How do these numbers and those in Figure 6 compare with other studies, and IPCC?

Technical corrections

1. Please add leading zeros to all decimal points in the manuscript, e.g. p.1 l. 14-15.

2. Organic carbon (OC) is defined with three different ways in the text: weakly absorbing organic matter (p. 11), organic matter (p. 12), and organic carbon (p. 13). Please pick one, but if this is organic matter, consider using OM instead of OC, since the two are not the same (organic matter contains more elements than just carbon).

---

## Author Comment (AC1) · 19 Apr 2019

*Both reviews were very helpful and constructive. Thanks! The paper was revised and all plots now show the data at full spatial resolution (the color-bars have not been changed but that can be done for the final submission). Below are my answers to the individual comments and I have also attached the latest version describing the MACv2 climatology (as this paper is still in the Tellus B review process).*

**Anonymous Referee #1**

The manuscript describes aerosol forcing calculations using the MACv2 climatology. The results presented are clear and interesting, which deserve publication after addressing my comments below.

**Major comments**

1. Necessary information about MACv2 are missing. The introduction is very short, with only two citations, one of which is the unpublished MACv2 climatology (Kinne, 2008). More details of both the MACv2 climatology and other aerosol climatologies are needed, especially with regard to how MACv2 was generated, how it is different from version 1 and how MACv2 is different from other aerosol climatologies, e.g. reanalysis products from MERRAaero (https://doi.org/10.5194/acp-15-5743-2015) and CAMS ([https://doi.org/10.5194/acp-19-3515-2019](https://doi.org/10.5194/acp-19-3515-2019)).
*Understanding the concept and assumptions of MACv2 is certainly an important element to judge the results of this paper. However, I did not want to duplicate the MACv2 paper for this application here. The paper is currently in review (in its second version) for a Tellus B publication . When submitting to ACP thought I had added the MACv2 paper in a supplementary file. I will now add the latest version of the MACv2 paper at the end of the responses to the authors.*

2. Statements like "These global fields are the result of a data merging process for mid-visible aerosol optical properties" (p.3, l.8) are too generic to be informative, and the whole paragraph that follows that statement is not specific on how the database was created. Important information that needs to be present includes how the climatology used the data mentioned, what the regional adjustments mean, how the monthly ensemble median of the 14 models was used, how the spectrally-defined properties are constructed from aerosol mixtures, etc. In essence, the manuscript lacks all information needed to understand how the climatology was constructed, e.g. how the fine mode was separated by the coarse mode, the anthropogenic from the total AOD, presented in Figure 1.
*I understand the frustration of the reviewer not having this information. Hopefully the information is now provided with the supplementary information. However, I do not want to include all the detail of MACv2 in this paper, as I want to keep the focus here on aerosol radiative effects.*

3. Comparisons with specific satellite instruments at the same or similar wavelength are also needed, e.g. estimates from AERONET, MODIS, etc. As the manuscript stands right now, the user is left to discover from figure to figure and from table to table that the climatology includes extinction, SSA, asymmetry parameter, 4 wavelengths, interannual variability from preindustrial to future (under which scenario?), a vertical distribution from a model (which?), studies of individual aerosol components, etc.
*I intentionally sidestepped aerosol optical properties from satellite remote sensing. Satellite retrievals usually address only one aerosol properties (AOD). And these are only correct, if assumptions to other aerosol properties (aerosol size and especially aerosol absorption) and to environmental properties (surface reflections) are correct – not to speak about cloud issues (sub-pixel cloud contamination, dust event removal as clouds). Thus only observational data of AERONET (and MAN over oceans) are used*

*also since via inversions of sun+sky data all relevant aerosol properties (column amount, size and composition) are simultaneously addressed.*
*The MACv2 climatology is based on mid-visible aerosol optical properties, which are distributed in a consistent way on components with pre-defined spectral behavior, so that from their locally defined component mixture single scattering properties (AOD, SSA. ASY) are defined at any spectral region (not just 4 wavelengths, which are only presented in the paper for illustrations).*
*The aerosol vertical distribution is pre-scribed from a multi-annual scaling from bottom-up modeling separately for fine-mode and coarse-mode (multi-annual CALIPSO were considered but a rather used mainly for validations, as CALIPSO data – even with their types - cannot easily distinguish between fine-mode and coarse-mode extinction contributions).*

4. Brown carbon (BrC) is virtually absent from the manuscript. There is a mention of weakly absorbing organic matter, but it is unclear whether all organics are treated as absorbing, or a fraction of them is absorbing (and that fraction is BrC).
*I do not like the BrC component. This is not really an independent component to me, as it is an absorbing carbon mixture, to which black carbon (BC) and organic matter (OC) contribute. In MACv2 there is only limited microphysical information I could extract from the optical properties for AODf, AAODf and REf for fine-mode microphysics. Thus only 3 pre-defined fine-mode components are permitted in MACv2 are: strongly absorbing BC coated with OC, (towards the UV very strongly) absorbing OC and non-absorbing fine-mode (e.g. sulfate, nitrate, small seasalt) summarized by the SU component. All this (here probably distracting detail) is described in the attached MACv2 paper.*

**Minor comments**

1. The abstract, especially lines 8-12, needs some editing for clarity; the wording is a little awkward. The wording is frequently cryptic or convoluted (e.g. p.5, l.11- 13: "Despite strong regional shifts in regional maxima (from US and Europe to SE Asia) changes to the annual global averages over the last three decades are relatively small and are presently near a global average maximum"). As this is a monograph manuscript, I would propose that the manuscript could benefit from a colleague reading it through and providing some advice on specific wording.
*I tried to rewrite and simplify those sections.*

2. The aerosol direct effect is not "aerosol presence", as stated in the abstract and introduction. It is, instead, the extinction of radiation by aerosols.
*the direct effect is caused by the extra aerosol presence in the atmosphere (in contrast to indirect effects which are caused by changed cloud properties through aerosols). Certainly this added presence is expressed via extinction, size and composition of aerosols in an appropriate environment is described via a TOA response at all-sky conditions for the direct aerosol forcing. I will rephrase to "caused by the aerosol presence".*

3. Page 2, lines 9-12: The double-radiation calls do have memory from previous timesteps in model simulations, so this statement is only accurate in a per-timestep basis. For example, the modified clouds at a given timestep will result in changes in precipitation later, which will impact aerosol concentrations.
*I am working with monthly statistics. There are no time-steps with processing involved. And cloud properties assume ISCCP monthly cloud statistics for high, mid and low cover and cloud scene optical depth. In the dual calls here, there are not feedbacks involving atmospheric processing (e.g. via*

*precipitation). Fortunately, global model simulations indicate that rapid adjustments in the context the aerosol forcing are small (on the order of 10%).*

4. Page 5, last line: Is the GISS model the one used for the vertical distribution of aerosols as well (same page, l.14-15)?

*No, the GISS model was only used to define the monthly average surface temperature so that mixtures of AFGL standard atmospheric profiles could be picked to approximate the state properties and trace-gas properties for the atmosphere by month for each 1x1 lat/lon region. For the vertical resolution 20 year simulations with ECHAM-HAM were used, mainly because in comparisons to CALIPSO data, ECHAM-HAM was among on the best performing (aerosol vertical distribution) tested global models (Guibert et al., 2008).*

5. Figure 4: what exactly is the total albedo? Integrated spectral albedo over the solar spectrum? Something else?

*Yes, the total albedo is the solar constant weighted UV/VIS and near-IR surface albedo. Since there is approximately similar energy below and above a wavelength of 700nm the total albedo is approximatelt the average of UV/VIS and near-IR surface albedo. The total albedo is not used, as a spectrally resolved radiative transfer model was applied.*

6. Page 6, line 17: is the change of solar zenith angle (SZA) as a function of time of day and season taken into account, or a mean day/night SZA representative for an annual mean as a function of latitude used for the calculations? This statement is not clear. What are the 9 SZA? Do they change with season? Are there 9 calculations, one per time of day, or just one calculation per average SZA conditions?

*The calculations are done with monthly averages. To include varying impacts at different solar elevations calculations of nine solar zenith angles (CZA) - evenly spaced in cos (CZA) space – are combined according the daily fraction associated with each sun-elevation and the night-time at each month and location (considering the latitude and the latitude position of the sun).*

7. Page 6, last line: For the indirect effect, there is no change in cloud droplet number concentration assumed? Just the size? And how much is the size reduced, a fixed value or depending on conditions? How is 3d specific humidity taken into account, via model output, or?

*The indirect effect is associated with the anthropogenic AOD and the background AODf which vary locally by month via the application of the logarithmic relationship (more details are given in Appendix A). Humidity is defined by standard atmospheric profiles and is not allowed to change.*

8. Page 7, line 5: Please provide a reference that supports this statement.

*A reference has been added. It is based on radiative forcing simulations with different models a part of the CMIP6 RFMIP exercises. With 10+ year simulations with constant SST, the rapid adjustments (referring yo atmospheric feedbacks on the forcing) are on the order of only 10%.*

9. Figure 5: Is the anthropogenic column for the total or the solar effect? Also, I believe showing the thermal effect alone would be of value, as one additional column. How do these numbers and those in Figure 6 compare with other studies, and IPCC?

*The anthropogenic effect is only a solar effect, as in MACv2 only fine-mode aerosol contributions to anthropogenic aerosol are considered. The solar only responses are show for comparisons to estimates to satellite (CERES) data or ground based (BSRN) data. The differences between total and solar forcing define the IR effects predominantly by coarse-mode mineral dust.*

**Technical corrections**

1. Please add leading zeros to all decimal points in the manuscript, e.g. p.1 l. 14-15.
*done*

2. Organic carbon (OC) is defined with three different ways in the text: weakly absorbing organic matter (p. 11), organic matter (p. 12), and organic carbon (p. 13). Please pick one, but if this is organic matter, consider using OM instead of OC, since the two are not the same (organic matter contains more elements than just carbon).

*any 'organic carbon' in the text has been changed into 'organic matter'*

**Anonymous Referee #2**

**major comments**

The author describes the application of the MACv2 global aerosol climatology to calculating direct and indirect aerosol radiative effects and forcing, decomposed by aerosol composition, in both the LW and SW. Maps and time series of the different effects are presented. The author estimates a total aerosol forcing of -1Wm-2 with a possible range of -0.7 – -1.6 Wm-2. They also show a relatively constant forcing over the last decades despite a large regional shift due to shifting emissions.

Overall the paper is well structured but a number of issues need rectifying before I would consider it suitable for publication in ACP:

- The paper describes the forcing responses of this particular setup comprehensively, and would be well suited for GMD. In order for it to meet the scope of ACP however I would suggest much more discussion around how these estimates are an improvement on existing e.g. AeroCom estimates, particularly in the context of the many other studies and efforts in this vain, both modelling and observational.
*This is an alternate approach to the 'bottom-up' modeling approach with emission data, only simpler and more direct (from optics to forcing) as aerosol observations of AERONET are at the center of this approach. For global coverage AERONET optics is extended with spatial context from global modeling. The aerosol optics are applied to extract aerosol microphysics (e.g AOD partition into BC, OC, SU, SS and DU components). Thus this approach is referred to as a 'top-down' approach. This component detail is used to define solar and IR spectral properties for radiative transfer simulations. The results, even for component direct forcing and first indirect effects are in line with results from 'bottom-up' modeling as presented in a table in the summary chapter.*

- In particular, the paper aims to provide an estimate of the aerosol forcing, but ignoring cloud adjustments seems a significant omission and requires much stronger justification, particularly when discussing potential uncertainties. For example, on P2L9-10 the author states that feedbacks (adjustments) can be considered secondary, though the provided citation makes no such assertion – they just assume it for their purposes.
*Here off-line radiative transfer simulations a used. Thus, no feedbacks or (earth system) adjustments to the forcing can be considered. Fortunately, there is a recent RFMIP exercise where in long-term simulations (with fixed SST) compare results of the (direct and first indirect) forcing from dual calls with the effective radiative forcing, which allowed atmospheric adjustments to the forcing in global models. The forcing differences are found to be relatively small on the order of 10% (Fiedler et al., 2019) .*

- The first indirect (Twomey) effect is accounted for, but the fittings used to extract the sensitivity from satellite retrievals is non-standard and needs much more justification and discussion. Why was this functional form chosen, how were the parameters chosen, and what are their uncertainties?
*There is many simplification involving the first indirect radiative effects. However, rhe concept is very direct by linking aerosol number and droplet concentration. I consider it a strength that observational evidence (statistics) is used and it is very comforting that applying different retrievals and different satellite sensors (which both retrieve AODf and CDNC) yield very similar relationships. Finally, it is also*

*comforting that the resulting indirect forcing in pattern distribution and strength quite believable and very similar to those of much more complex approaches.*

- The color scales used for the plots are very difficult to interpret and at times even use the same color for different values. Worse, these non-monotonic colormaps can also distort the impression of the values being plotted (see e.g. https://matplotlib.org/users/colormaps.html and references therein) and should be replaced.
*I have got already got some criticism on my (standard) color scale from other sides. I have been working with other color scales but not always were these scales sufficient to separate all the detail. I am willing to modify for the final version in some plots the color-scales (once the scientific content is considered acceptable)*

- Also, the manuscript can be difficult to read at times and would benefit from editing help from someone with full professional proficiency in English.
*I have tried to simplify the language for better reading. The suggestion though is a good and I will try to find a colleagues.*

**minor comments**

- P1 L10: '...major aerosol indirect effect. . .' should be 'first aerosol indirect effect'
*done*

- P1 L12: 'locally' -> 'local'
*done*

- P2L13-14: The author claims that this setup avoids 'time-consuming aerosol processing' though the fields used to create the MACv2 climatology are derived from just these costly models. This should be clarified.
*the section has been rewritten "By prescribing optical properties for aerosol and clouds (with strong links to observations) the derived aerosol radiative impacts are faster, more precise and more direct than with 'bottom-up' approaches."*
- P2L19: Suggest 'before impact results' be replaced with 'before forcing results'
*I prefer to be more general with 'radiative impacts' as 'radiative forcing' refers to the climate-relevant anthropogenic aerosol impacts at the TOA*

- P3L13: The use of the AeroCom phase 1 models is introduced here, but later on some phase 2 values are also used and should be introduced as well.
*The AeroCom phase 2 data made advances with respect to fine-mode aerosol, including the use of more recent (IPCC 5) emission data. This is also the reason why an AeroCom phase 2 anthropogenic fine-mode fraction was adapted for this newer MAC version. On the other hand the AeroCom phase 2 data for natural aerosol components (e.g. dust and sea-salt) displayed strong intermodal diversity and many local biases which also affected its ensemble averages for total aerosol properties.  Since reliable fields for total aerosol properties are needed to spatially extend the AERONET data, the use of AeroCom phase 2 data for the merging process was not a good choice.*

- P3L20: Shouldn't all the terms in the scaling equation related to AODf? Currently some relate to AOD.

*Oh yes, thanks for discovering this error (has been corrected)*

 - P3L20: IPCC5 -> CMIP5
*has been changed*

- P6L21: The description of double radiative calls is confusing, please consider rephrasing. The author should also probably cite Ghan 2013.
*the forcing definitions in Ghan 2013 are much more complex also as changes to clouds are less constrained in climate simulations. The section has been rephrased.*

- P7L5: The assertion that feedbacks are on the order of 10% requires a citation. The author should also acknowledge that there is considerable uncertainty around this.
*I agree that there is uncertainty. The point is that the impact of rapid adjustments (climate simulations with fixed SST) to the aerosol forcing is small. A reference has been added.*

- P8 Fig5: Perhaps space out the maps a bit more vertically so that it's clearer which mean values apply to which maps. The 'anth all_sky' toa probably doesn't need scaling either, it's misleading when glancing at the data. Perhaps put the TOA and SURF plots on different scales? It should also be labelled as 'all_toa_a' for consistency.
*the plot has been modified as suggested, the range has been reduced, the average values have been moved upward (now more next to the text) and the 'all_toa_a' label has now been used (without multiplying that field by a factor of 10) to better illustrate how relatively small the direct climate impact in comparison is to other anthropogenic radiative impacts.*

- P9L3-4: The assumption of ignoring potential anthropogenic course mode aerosol should be discussed in the introduction.
*I modified the text and gave an explanation in the introductions why anthropogenic coarse-mode contributions were ignored. The associated AOD map is given in Appendix E – Figure E1. There are 3 reasons to do this: 1. The anthropogenic coarse-mode AOD (using on a scaling based on a satellite data analysis) is very small (0.006). 2. The associated solar and IR forcing contributions partially cancel each other and the global direct (TOA, all-sky) forcing very small (+0.02 Wm2). 3. The added (larger) dust particles also have only a minor impact on elevating the aerosol number concentrations (not to speak of the limited capability of dust to act as an effective CCN). Thus, a potential first indirect effect is at best secondary (and then even with an opposite in sign to the direct effect).*

- P10 Fig7: These fields appear to be presented at a degraded resolution compared to the previous ones, are they not calculated on the same grid?
*yes (correctly noticed!). They were calculated at a different spatial resolution (12x6 rather than 1x1). The annual averages and distributions were similar and it saved a lot of computer-time. Now all plots show data of higher (1x1) resolution simulations.*

- P10L23-25: The assumed anthropogenic dust contribution should be introduced in the introduction section. It's not clear when this is included and when it isn't.
*Yes, it is now included in the introduction and explained in more detail in Appendix E, where the global annual map for anthropogenic dust is presented in Figure E1.*

- P12L26: 'tome' -> 'time'
*thanks*

- P13L6: "also have to considered" -> "also have to be considered"
*thanks*

- Table 2: The coarse mode dust is used here again, it needs introducing and the introductions of anthropogenic aerosol only contributing to AODf needs clarifying
*good point. Done so.*

- P17L7: The assumption of constant LWC affects the size of their droplets, and hence their reflectivity, not the other way around. The sentence should be altered to make this clearer. Something like: ". . .that the resulting water cloud droplets are more numerous and smaller, assuming no change to the cloud liquid water content (LWC). With smaller drop sizes the solar reflection. . ."
*thanks. So changed*

- P17L11: ". . .then smaller. . ." -> ". . .the smaller. . ."
*'then' removed*

- P17L33: This sentence doesn't seem to make sense: "However, there is reliance, that regional associations will provide the needed link.". Please rephrase.
*I rephrased to "Here it is now stipulated, that already regional associations of monthly averages between AODf and CDNC offer meaningful statistical information on aerosol-cloud interactions."*

- P18L3: "to extract" -> "the extraction of"
*done*

- P18L4: "meant that" -> "means that"
*thanks*

- P19L9-11: There are many reasons why the satellite retrieved response could be biased low as well, and recent work has even suggested very large sensitivities (Rosenfeld et al. 2019). The most that can be said is that they're different for reasons still to be fully understood.
*I admit that there is a lot of uncertainty associated with aerosol indirect effects. I do not deny that there are varying sensitivities even based on different approaches involving satellite observations. But these are usually associated with case studies and do not directly link aerosol number and cloud droplet number. The attempt here (as explained in Appendix) was to work with solid monthly association statistics of more reliable data only (even at the expense of losing regional and seasonal variability). The fact that the logarithmic curvature (the factor of about 1000) is captured by different satellite sensors and different satellite CDNC retrievals (MODIS 5.1, MODIS 6.0 and ATSR) is encouraging (even though there is significant scatter when fitting those curves) and it builds confidence that strong observational constrains are here provided.*

- P19L12-13: Why is a single sensitivity applied globally? The 1x1 degree sensitivities are available and are presumably very different locally. Or is the sensitivity calculated globally? In which case large errors are likely to be present (Grandey and Stier 2010). This section needs more discussion and justification.
*In order to have sufficient (monthly) statistics in developing the logarithmic relationship seasonal and spatial context was sacrifized.*

– P20L14-16: Given these uncertainties it could presumably also be underestimated then?

*I do not think these effects are significant over- or underestimated. I also cannot follow the argument why an underestimate should be expected. All associations are collected from matching monthly statistics over oceans. As over oceans aerosol indirect effects are largest (due to the darker background) the most important contributions are probably captured well. The logarithmic curve is very sensitive at low background condition, so that indirect forcing efficiencies are very high over the Southern Hemisphere oceans leading to potential overestimates. However, anthropogenic AOD there is very low so absolute contributions from that region are minor. With respect to the use of large 1x1 monthly averages it was shown by Matt Christensen with ATSR data that the AODf to CDNC relationships from closest neighbors using daily data were basically identical to the monthly 1x1 AODf to CDNC relationships. (Also the relationship uncertainties seem small compared to the uncertainty about what constitutes anthropogenic in AODf.)*

- P25L4: continued -> continues
*thanks*

- P26L6: "As alternate background ICAP" -> "As an alternate background, ICAP"
*changed*

- P26L12-14: This isn't clear. Does the ICAP ensemble include AODf and anthropogenic AOD? How does it calculate them? From which models? Does if assume the same Lamarque emissions as MACv2?
*The ICAP (assimilation ensemble) data provide only global maps for fine-mode AOD (AODf) and coarse-mode AOD (AODc). Thus in the merging of only these properties these ICAP data replaced the ensemble AeroCom phase 1 background maps. Otherwise all other assumptions (anthropogenic scaling (1850, Lamarque), altitude scaling and temporal scaling) were treated identical. The section has been rewritten*

- P28L2-5: This is a confusing sentence, consider re-phrasing
*rephrased*

- P28L9-10: The uncertainty due to the parameterisation of the Twomey effect is probably very large due to the large uncertainties in measuring AODf and CDNC individually, confounding (e.g. meteorological) factors, and the (weak) causal relationship between AOD and CDNC. There are also likely to be large uncertainties due to the choice of parameters used to fit these distributions since the error bars are large and small AODf values are not measured. Both of these should be discussed and estimated. Potentially large liquid cloud adjustments are also ignored and should be discussed.
*the uncertainty discussion has been expanded*

- P32L35: matches -> matched - P33L15: larger -> large
*changed*

**suggested additional references**

Ghan, S. J.: Technical Note: Estimating aerosol effects on cloud radiative forcing, Atmos. Chem. Phys., 13, 9971-9974, https://doi.org/10.5194/acp-13-9971-2013, 2013

Daniel Rosenfeld, Yannian Zhu, Minghuai Wang, Youtong Zheng, Tom Goren, Shaocai Yu: Aerosol-driven droplet concentrations dominate coverage and water of oceanic low-level clouds, Science 2019

Grandey, B. S. and Stier, P.: A critical look at spatial scale choices in satellitebased aerosol indirect effect studies, Atmos. Chem. Phys., 10, 11459-11470, https://doi.org/10.5194/acp-10-11459-2010, 2010.

**The MACv2 Aerosol Climatology**

S.Kinne, MPI-Meteorology, Hamburg, Germany (e-mail: stefan.kinne@mpimet.mpg.de)

**Abstract**

The MAC aerosol climatology defines monthly global maps for aerosol properties. The definition of mid-visible optical and microphysical properties is strongly linked to multi-year statistics of observations by sun-photometers of the AERONET and MAN ground networks. As available statistics is spatially sparse, context from bottom-up global modeling is added. Now in its second version, oceanic MAN reference data are included, a different lower anthropogenic fraction is assumed and the merging of the data-statistics is improved. Hereby, now only absolute properties are merged and trusted photometer data are given stronger weights via regional corrections in place of local domain limited corrections. Global average mid-visible (550mn) aerosol properties are 0.12 for the aerosol optical depth (AOD), 0.94 for the single scattering albedo (SSA) and 0.7 for the asymmetry-factor (ASY). Averages for sub-micrometer (fine-mode) and super-micrometer (coarse-mode) aerosol sizes are 0.63 (AODf) / 0.58 (AODc), 0.92 (SSAf) /0.965 (SSAc) and 0.64 (ASYf) / 0.77 (ASYc), respectively. A new element is the separation of aerosol absorption (AAOD) by sky-/sun-photometers into fine-mode and coarse-mode contributions. These properties as well as the fine-mode effective radii were merged with background data from global modeling yielding global averages of 0.050 (AAODf), 0.021 (AAODc) and 0.18um (re,f). Local monthly mode detail now allows (in a 'top-down' approach) to extract global distributions for aerosol component amount and size. As the considered components for soot (BC), organics (OC), non-absorbing fine-mode (SU), sea-salt (SS) and mineral dust (DU) have pre-defined spectrally resolved properties, optical properties at other than mid-visible wavelengths are automatically defined - as required in broadband radiative transfer applications. With component information (e.g. amount, composition and size) also MAC estimates for CCN and IN concentrations are possible and also a simple MAC based aerosol retrieval model for satellite sensor data is suggested.

**1. Introduction**

Tropospheric aerosols originate from many sources of natural (e.g. windblown dust, sea-spray, wild-fires) or anthropogenic origin (e.g. fossil fuel burning, agricultural burning). Since also the aerosol lifetime is only on the order of a few days (mainly due to the removal by precipitation), tropospheric aerosols are highly variable in concentration and composition. However, only aerosol sizes larger than a tenth of a micrometer in size (>0.05um in radius) directly influence the radiative energy distribution in

the atmosphere. Sub-micrometer (fine-mode) sizes only affect the solar radiative transfer, while super-micrometer (coarse-mode) sizes modulate both solar and terrestrial radiative energy distributions.

Although the influence of aerosols is small compared to that of clouds, there is from a climate change perspective still a strong interest on the impact of aerosol (as small as it may be) because part of today's atmospheric aerosol is anthropogenic in nature. However, quantifying global aerosol radiative effects via model simulations is rather complex (*Kinne et al., 2003, Textor et al., 2007*). In these so called 'bottom-up' approaches emissions of different aerosol species and pre-cursor gases are chemically and cloud processed, mixed, transported and removed. In addition, assumptions to size, water uptake and component mixing are required to establish aerosol optical properties (which are the basis in the determination of associated aerosol radiative effects). With accuracy limitations to (emission) input, processing mechanisms in modeling and microphysical assumptions, these 'bottom-up' approaches have large uncertainties and are time-consuming as well. Thus, there is a need for faster and more direct approaches to define characteristic aerosol optical properties – even on global scales.

Quality observations on solar attenuation and solar scattering offer data all relevant aerosol optical properties. Such data are available from globally distributed ground networks. These data are already used evaluate efforts in 'bottom-up' modeling (and satellite remote sensing). However, by themselves these data also establish an elementary and quasi-global data-base on aerosol optics. Ground-based sun-/sky-photometer data inform on aerosol amount and aerosol absorption, both as function of size, to inform together in a so-called 'top-down' approach on aerosol composition and even aerosol mass.

Such an observation-data based 'top-down' approach is the topic of this MACv2 contribution. Aerosol optical properties for aerosol amount and aerosol absorption, both as function of size-modes with associated effective radii, are provided in monthly climatologies with global (1x1 deg lat/lon) coverage. Building on the concepts of MACv1 (*Kinne et al., 2013*), limitations to (spatial, temporal) coverage, and missing data for vertical distribution and pre-industrial state are addressed with scaling methods derived from ensemble averages of 'bottom-up' modeling. MACv2 updates over MACv1 include more recent AERONET data and now also oceanic MAN references, a modified spatial merging procedure, (2) a more solid (now component based) spectral extension and an update for anthropogenic AOD contributions.

In chapter 2 the MACv2 mid-visible optical column properties and their global properties are introduced. In chapter 3 choices for aerosol component types and sizes are defined and based on these definitions global distributions of component mixtures are suggested (where local monthly mixtures are consistent with the optical and microphysical properties of MACv2). Definitions for anthropogenic aerosol are covered in chapter 4. The spectral extension of the aerosol radiative properties (as needed for radiative transfer simulations) is presented in chapter 5 by combining the spectral information of the pre-defined components and their assigned local monthly mixtures. In chapter 6 the assumptions for the vertical distribution are explained. Chapter 7 offers estimates for aerosol concentrations and chapter 8 introduces a simple model for first guesses in satellite retrievals of aerosol amount, by offering likely local properties on needed assumption for absorption and size.

**2. mid-visible column properties**

**2.1 What is MAC?** The Max Planck institute Aerosol climatology (MAC) is defined by monthly global (1x1 deg) fields for aerosol optical and microphysical properties. Central to these fields are atmospheric column aerosol optical properties in the mid-visible spectral region. These column properties are based on trusted ground-based observations sampled over the last two decades. As observations are sparsely distributed, spatial context from global 'bottom-up' modeling is added to yield spatially and temporally complete fields. In a so called 'merging process' monthly maps defined by global modeling are adjusted according to monthly statistics by ground-based sun photometry.  Contributing photometry data are quality assured (level 2) direct solar attenuation and sky radiances samples of CIMEL instruments operated by the AERONET network (*Holben et al. 2001*) and solar attenuation samples by handheld MICROTOPS instruments of the Marine Aerosol Network, MAN (*Smirnov et al., 2009*). The maps from modeling are defined by local monthly median values of 14 different AeroCom (phase1) models (*Kinne et al., 2006*). For a flavor of the applied aerosol data in the development of the MAC climatology, annual averages of input maps for four important aerosol properties are compared in Figure 1.

[Figure]

*Figure 1.* Input data of MACv2. Shown are the multi-annual averages of aerosol optical properties at 550nm based on sun-photometer measurements by AERONET and MAN (left block, local data are enlarged for better viewing) and by the AeroCom 'bottom-up' modeling ensemble (right block). The four sub-panels in each block display the aerosol optical depth AOD (upper left), the absorption aerosol optical depth AAOD (lower left, multiplied by a factor 10), the fine-model aerosol optical depth AODf (upper right) and the fine-mode effective radius REf (lower right). The sun-photometer data coverage over oceans is higher for AOD and AODf due to the additional MAN data.

**2.2. What has changed since MACv1?** Compared to the initial MACv1 version (*Kinne at al., 2013*) major upgrades are incorporated. In MACv2 (1) more recent AERONET data are included so that the central reference year shifted from year 2000 to year 2005, (2) MAN data over oceans are now considered, (3) a new regional data merging procedure is applied, (4) only absolute properties are merged (e.g. AAOD

instead of SSA), (5) pre-defined aerosol types are utilized to assign local component mixtures and (6) a new (smaller) anthropogenic fraction is applied.

**2.3. What trusted quality data are applied?** Solar photometry measurements from the ground offer at cloud-free conditions reliable atmospheric column data and for aerosol on properties of amount, size (distribution) and absorption. Together these (three) aerosol properties even inform about the aerosol compositional mixtures. Particular accurate are direct solar attenuation data for aerosol column amount via the aerosol optical depth (AOD). Sun-photometers sample AOD at different solar wavelengths (e.g. AERONET's CIMEL instrument version2 and level2 samples at 380, 440, 500, 670, 870 and 1020nm are used). With AOD data at different wavelengths also estimates for the aerosol size are revealed. Aerosol particles size is either captured by the negative slope in d(ln(AOD))/(d(ln(wavelength))-space (Angstrom parameter) or in a more advanced way by a separation of AOD contributions to sub-micrometer (fine-mode) or to super-micrometer (coarse-mode) sizes (*O'Neill et al., 2003*) - via the Angstrom parameter spectral dependence. The Angstrom parameter of the AOD at 440nm and 870nm data pair is also used to determine the AOD at 550nm, which is the common aerosol reference wavelength in global modeling and satellite remote sensing. In addition to the direct solar intensity samples the added information on solar spectral sky-radiances in near forward and side scattering directions offer (via inverse radiative transfer methods) details on aerosol size distribution over the optically active size-range (radii larger than 0.05um) and absorption estimates (e.g. the *Dubovik et al, 2000* algorithm applied to AERONET's CIMEL data expresses absorption via refractive index imaginary parts at 440, 670, 870 and 1020nm). Here in version2 level 2 CIMEL samples, the refractive index data from level 1.5 data were reinstated.

**2.3.1 aerosol column amount**: Direct solar attenuation (at cloud-free conditions) data define local monthly AOD. Before averaging the photometer samples were interpolated (with the Angstrom parameter) to the value at 550nm (the reference wavelength in modeling and remote sensing). The established local monthly averages are based on multi-year (between 1995 and 2015) quality checked CIMEL version 2 and level 2 samples at more than 700 continental or island AERONET sites worldwide (https://aeronet.gsfc.nasa.gov/). Also MICROTOPS samples of almost 100 ship-cruises between 2006 and 2015 were included (https://aeronet.gsfc.nasa.gov/new_web/maritime_aerosol_network.html).

**2.3.2 aerosol column amount by size mode**: The separating radius between the smaller fine-mode and larger coarse mode in MACv2 is set at 0.5um. From (to CIMEL sky-/sun samples) applied inversions, concentrations of the 9 lower size bins (0.05 to 0.5um) contributed to the fine-mode AOD and concentrations of the 13 larger size bins (0.5um to 15um) contributed to the coarse-mode AOD. Alternately, from the sun-attenuation only samples of CIMEL and MICROTOPS the Angstrom parameter spectral dependence (based on simultaneous AOD data at least five different solar wavelengths) defined via the SDA method (*O'Neill et al., 2003*) approximate estimates for AOD attributions to the fine-mode (AODf) and to the coarse-mode (AODc). The ratio between fine-mode AOD and total AOD is referred to as fine-mode fraction (FMF).

**2.3.3 aerosol column absorption**: The imaginary parts of the refractive index define the aerosol absorption. The aerosol absorption is quantified by the Absorption Aerosol Optical Depth (AAOD). The

AAOD (= AOD* [1-SSA]) is the product of AOD and the absorption potential 1-SSA (where SSA is the alternate scattering potential). In the mid-visible spectral region SSA is on average more than an order of magnitude larger than (1-SSA). Thus, at lower aerosol loads the reduced sky scattering due to absorption is difficult to detect. Thus only absorption data at larger aerosol loads (AOD at 550nm >0.2) are more reliable. CIMEL sky-/sun-photometer samples offer refractive indices but in their version 2 level 2 data at absorption defining data are very conservatively removed, probably too conservative (*Dubovik, private communication*). For MACv2 all removed refractive indices in AERONET level 2 data were recovered from level 1.5 data. Then for aerosol loads below the above threshold (based on local monthly statistics) the absorption potential (1-SSA) at that threshold (or at the statistically largest aerosol loads if below that threshold) were prescribed for all lower aerosol load cases. This method assured that the high uncertainty for absorption at low aerosol loads was reduced without biasing the sample statistics.

**2.3.4 aerosol column absorption by size-mode**: A new element in MACv2 is the separation of the AAOD into fine-mode and coarse-mode contribution. The estimate for the fine-mode AAOD is based on an analysis of AERONET data over western African sites where both fine-mode (wildfire) and coarse mode (dust) coexist with stronger fine-mode AOD contributions from Nov to Feb and stronger coarse mode contributions from Apr to Sep. The spectral dependence of the AAOD data (offered by CIMEL data inversion data) alone was not sufficient to separate the AAOD. However with the added information of the AOD fine mode fraction (FMF = AODf / AOD), estimates for the fine-mode absorption (AAODf) were possible - also automatically defining the coarse-mode absorption (AAODc) contributions.

**AAODf$_{,550}$ = AAOD$_{,550}$ *FMF**(A*B/C)**   with      **FMF** = AODf$_{,550}$ / AOD$_{,550}$

AAODc$_{,550}$ = AAOD$_{,550}$ - AAODf$_{,550}$          **A** = 0.2 +0.25*(1-FMF)**2,

$\qquad\qquad\qquad\qquad\qquad\qquad\qquad\qquad$ **B** = ln(AAOD$_{,440}$/AAOD$_{,670}$) / ln(.67/.44)

$\qquad\qquad\qquad\qquad\qquad\qquad\qquad\qquad$ **C** = min (ln(AAOD$_{,670}$/AAOD$_{,1020}$) / ln(1.02/.67), B)

**2.3.5 aerosol column effective radius**: The relevance of aerosol size is already addressed, as amount and absorption of aerosol is (see above) are already split into contributions by sub-micrometer (fine-mode) aerosol and super-micrometer (coarse-mode) sizes. In addition, from (to CIMEL sky-/sun samples) applied inversions, size distribution data define effective radii (re=sum{n*r**3}/sum{n*r**2}) for both the fine-mode (REf) and for the coarse mode (REc). REf is an important parameter to estimate the aerosol number concentrations of optically detectable sizes (r>0.05um) and REf is essential for CCN estimates with MACv2 data (at low supersaturations).

**2.4. How are data merged?** The merging combines the accuracy of local photometer statistics with gap-filling spatial context by global modeling using monthly statistics at 1x1 deg lat/lon spatially resolution.. Then, global sub-regions with sufficient photometer sites are picked as illustrated in Figure 2. In those regions for each month interquartile averages of only matching grid-points are compared. Their ratio (photometer to modeling data) defines multipliers that are applied to all modeling background values in that region (and month). Finally, spatial inconsistencies of multipliers at regional boundaries are smoothed, by replacing local multipliers with +/- 6 deg in longitude and +/- 3 deg in latitude averages Annual average multipliers for properties of Figure 1 are presented in Figure 2.

[revised manuscript text omitted]

*Figure 5 assumed real (left block) and imaginary part (right block) parts of the refractive indices for the different aerosol components (and for water and ice) of Figure 4 for central wavelengths representing the 30 spectral bands of the RRTM radiative transfer model. References are Hale and Query (1973) for water, Warren (1984) for ice, Palmer and Williams (1974) for sulfate solutions, Nakayama et al. (2010) for organic aerosol, Bond (private communication) for BC, Nilsson (1979) for seasalt and Sokolik (private communication) for dust in the infrared and Wiedensohler (private communication) for dust in the solar region. Note, that the imaginary part of (only) 0.001 for dust was derived from by the mid-visible coarse-mode size and absorption of MACv2.*

Hereby SU, OC and CA types only contribute to the fine-mode AOD, whereas DU and SS types only contribute to the coarse-mode AOD. Hereby the CA type (a BC soot core with an OC shell) also accounts for the fine-mode solar absorption enhancement and the SU type represents the non-absorbing fine-mode type, including nitrate and fine-mode SS. SU and DU types are allowed to vary in size with effective radii ranging from 0.05 to 0.64um for SU and from 1.5 to 10um for DU.

**3.3 how are components assigned?** To assign the pre-defined aerosol components to the MACv2 data (AODf, AODc, AAODf, AAODc) first the mid-visible AAOD for each component is determined via (MIE-) scattering simulations (*Dave, 1968*). With the size (-distribution) assumptions (as listed in of Table 1 and shown in Figure 4) the three SU, OC and CA components only contribute to the mid-visible AODf, while DU and SS component only contribute to the mid-visible AODc. Mid-visible absorption I indicated when the SSA falls significantly below 1.0. Thus, for the fine-mode only the CA and OC components contribute to AAODf and only the DU component contributes to AAODc. To start the component assignment, some ancillary global monthly maps data are utilized:
- Local monthly (BC+OC)/(BC+OC+SU) fractions for AOD components from 'bottom up' global modeling provide initial guesses for AODf splits between absorbing (CA, OC) and non-absorbing (SU) components.
- An ocean influence weight factors to avoid unrealistic sea-salt contributions over continents with a 1.0 weight over oceans but increasingly lower fractions with continental distances from the coast.
- Near-surface wind-speeds (in m/s) from re-analysis over oceans.

**3.3.1 coarse mode assignments:** For the coarse mode component properties, ocean influence weight (ocean), near wind-speed (wind ), local latitude (lat) and the sun's seasonal varying latitude position (sun) define initial estimates for the coarse mode AOD of sea-salt (SS_AODc,ini) based on a fit to global monthly sea-salt mid-visible AOD maps from global 'bottom-up' modeling:

$$SS\_AODc,ini = ocean * 0.03*wind *(2.0 - cos(2.0*lat - sun/2.0))$$

[revised manuscript text omitted]

*\* anthropogenic SSA and ASY are that of the fine-mode*

**6. vertical distribution**

**6.1 how vertically distributed?** All properties discussed so far refer to column averages. The aerosol vertical distribution is not only important for the IR radiative transfer (e.g. greenhouse effect of elevated mineral dust) but also for solar radiative transfer - mainly via the relative altitude of aerosol to clouds. Clouds above aerosol cut off potential aerosol impacts, aerosol at cloud altitude may modify clouds and with clouds below any aerosol absorption is enhanced.  The aerosol vertical distribution is prescribed separately for AODf and for AODc, as in MACv1. With the AOD scaling by mode the total values for SSA and ASY at each altitude will depend on the relative AOD contributions.

**6.2 ECHAM modeling**: The prescribed AOD vertical scaling in MACv2 is based on 'bottom-up modeling. Multi-annual (1986-2006) monthly AOD scaling factors were created from ECHAM-HAM aerosol component simulation not just for total AOD but also more importantly separately for fine-mode and coarse mode AOD. The results of the ECHAM model were selected, because in comparisons to active remote sensing from space, ECHAM-HAM simulations demonstrated one of the better skill scores for AOD vertical distributions among the tested AeroCom models (*B.Koffi et al., 2016*). The global annual averages for AOD, AODf and AODc in a relative and an absolute sense are presented for four pre-defined tropospheric sub layers in Table 3.  And the associated annual maps are presented in Appendix B.

**6.3 CALIPSO data**: Alternate data that could have been used to define aerosol vertical distributions are multi-annual (2006-2016) mid-visible extinction profiles offered CALIPSO lidar data from space. The processing of CALIPSO level 3 data (***Tackett et al, 2018*) in both, version 2 (already tested for MACv1) and (more recently) version 3, however, place too much aerosol in a relative sense too in the lowest atmospheric layers (0-1km), as if some low clouds are mistaken for aerosol. This apparent bias is demonstrated in comparisons of relative altitude distributions (sum of weights for all layers is 1.0) for total AOD in Table 3 and Appendix B.  Another handicap of CALIPSO data is that the needed separation into fine and coarse mode AOD contributions, which is needed for MAC, is almost impossible, despite CALIPSO efforts for aerosol typing, which however is radiatively not precise enough for a distinction.**

*Table 3* *annual averages of vertical distributions for the mid-visible AOD for four altitude (above sea-level) atmospheric layers (associated global maps are presented in Appendix B).  For total AOD, relative vertical distributions are compared between CALIPSO version 2 and 3 (columns 2 and 3) to ECHAM-HAM (column4). The relative vertical distributions of ECHAM_HAM for coarse and fine-mode (columns 5 and 6) are applied the AODc and AODf of MACv2 to the layer AODs of column8 and 9.*

| | fraction of column AOD | | | | | AOD per layer | | |
|---|---|---|---|---|---|---|---|---|
| | *CALv2 (z)* | *CALv3 (z)* | ECHAM (z) | | | ECHAM(z) *MACv2 | | |
| alt (km asl) | | *total* | | *coarse* | *fine* | *total* | *coarse* | *fine* |
| **6-12** | *0.003* | *0.030* | 0.059 | **0.039** | **0.076** | 0.004 | **0.001** | **0.003** |
| **3-6** | *0.080* | *0.091* | 0.097 | **0.093** | **0.104** | 0.015 | **0.006** | **0.009** |

| 1-3 | *0.347* | *0.280* | 0.457 | **0.432** | **0.468** | 0.059 | **0.026** | **0.033** |
|-----|---------|---------|-------|-----------|-----------|-------|-----------|-----------|
| 0-1 | *0.568* | *0.573* | 0.364 | **0.419** | **0.323** | 0.041 | **0.024** | **0.018** |

**7. cloud nuclei**

**7.1 aerosol as cloud particle nuclei?** The cloud particle formation is usually helped, when aerosols are present and available. However, the affinity to serve as nuclei depends on aerosol composition, aerosol size and environment. Usually there is a distinction between nuclei for water clouds, referred to as Cloud Condensation Nuclei (CCN) and nuclei for ice clouds usually referred to as Ice Nuclei (IN).  At lower altitude, where aerosol concentration are usually much higher, typical CCN concentrations range from about 30/cm3 in remote regions to more than 1000/cm3 in polluted regions. IN concentrations at a few/liter, in contrast, are much less frequent, with preferences for mineral dust and cold temperatures.

**7.2 CCN concentrations:** Usually all aerosols of the coarse-mode are large enough to serve as CCN, but only a fraction of the accumulation-mode aerosol qualifies. This fraction depends on the supersaturation (or upward wind at cloud-base altitude) and on the aerosol ability to attract water. The stronger the water attraction and/or the larger the supersaturations is, the smaller the aerosol sizes that can be activated.  Four different supersaturations (0.05, 0.07, 0.10 and 0.20%) are assumed and the ability to attract water is expressed by the kappa parameter (*Petters and Kreidenweiss, 2007*). Hereby component kappa values of BC (0.0) of OC (0.1) and non-absorbing fine-mode (SU, 0.8) are combined according to their MACv2 fine-mode mixture by mass (by applying the listed mass extinction efficiencies of Table 1, based on pre-defined density and size-distribution of a component). Then adding to the local kappa (k) value information on local temperature (T) and an applied  supersaturation (SS) the critical dry radius cR_dry can be determined (*Rose et.al, 2008, formula A32*):

$$cR\_dry \text{ (um)} = ((4.0 *(.66e\text{-}6/T)**3)/(27*k*\ln(SS/100+1))**2.0)**.33$$

Next, the dry radius cR_dry needs to be converted into the wet radius cR_wet in order to be applicable to ambient fine-mode size distributions. This is done by a simple parametrization that depends on kappa k and the ambient relative humidity rh (*Petters and Kreidenweiss, 2007*). As it is of interest to define the CCN concentrations near a low altitude cloud base is assumed that the relative humidity (rh) is at 90%.

$$cR\_wet \text{ (um)} = cR\_dry \ *(1 +k*rh/(1-rh))**.33, \quad rh=0.9$$

cR_wet are applied to the log-normal size-distribution of the accumulation mode. This distribution is defined of the fine-mode AOD and the fine-mode effective radius of the MACv2 climatology and an assumed standard deviation of 1.7. All accumulation-mode particles larger than cR_wet and all coarse mode particles are counted as CCN. CCN concentrations at cloud base (ca. 1km above ground) are determined by scaling column CCN data with the ratio of the fine-mode AOD fraction in the layer near the cloud-base and the layer thickness. CCN concentrations for different SS are presented in Figure 9.

A supersaturation of 0.1% yields the most realistic MACv2 based CCN concentrations considering that at relative low CCN concentration almost all available CCN turn into Cloud Droplet Number Concentrations (CDNC). Also the comparison between natural and anthropogenic CCN concentrations is interesting as in

regions of urban pollution anthropogenic concentrations outnumber natural concentrations occasionally by even more than an order of magnitude.

[Figure]

***Figure 9*** *annual averages for critical radii (left block) at 4 different supersaturations: 0.05% (upper left), 0.07% (lower left), 0.1% (upper right) and 0.2% (lower right). Corresponding CCN concentrations at lower cloud base (at 1km above the ground) at these supersaturations presented as well (right block). Hereby natural (left column), anthropogenic (mid column) and total contributions (right column) are compared.*

**7.3 IN concentrations:** It is assumed that only (coarse-mode) dust particles serve as IN and that this capability is sharply decreased when ambient temperatures get warmer than 238K and even more so if warmer than 258K. The T-reduced IN efficiency for dust concentrations is expressed by the T-factor.

$$\text{T-factor} = \exp(-A) * \exp(-B)$$

| | |
|---|---|
| A=0 | [T < 238K] |
| A=0.5*T-238, [238K >T > 258K] | B =0     [T < 258K] |
| A=10     [T > 258K] | B =T-258 [T > 258K] |

At selected three selected altitudes (7, 9 and 12km) the dust component fractional AOD in that altitude layer (as defined by the coarse-mode vertical distribution) is divided by the layer thickness and then multiplied with the T-factor. The resulting 'IN' concentrations are presented in Figure 10, where dust IN are missing over the tropics at 7km altitude due to too warm temperatures and is also missing over high latitudes at 12km altitude, at that altitude there is already well in the stratosphere.

[Figure]

*Figure 10* *annual average estimates for IN (dust) concentrations at 7km (left), at 9km (center) and at 12km (right). Note the log10 axis, so that 2 refers to 100 (/m3) and 6 to 1000000 (/m-3).*

**8. retrieval assistance**

**8.1 how to help satellite retrievals for aerosol ?** The local and monthly variability captured by the aerosol properties of the MACv2 climatology can help under-determined satellite AOD retrievals with smart assumptions to aerosol size and absorption, although other uncertainties (e.g. the solar surface reflectance) remain open issues.

**8.2 how to simplify size?** A bi-modal size-distribution is assumed, as in most aerosol retrievals models. Such a size-distribution shape is almost always retrieved by ground based sun-/sky photometry (*Dubovik et al., 2002*) showing a concentration minimum at radii near 0.5um. Thus, only two sizes are required. Based on frequency occurrences for effective radii of AERONET retrieved size-distribution samples, the most frequent effective radii are at 0.145um for the fine-mode and 1.9um for the coarse mode, as illustrated in Figure 11. Corresponding selections for log-normal distribution parameters are 0.096um (fine) and 0.97um (coarse) for the mode radius (defining the size with the largest number concentration) and 1.5 (fine) and 1.7 (coarse) for standard deviation (defining the distribution width).

[Figure]

*Figure 11* *frequency of coarse-mode effective radius (x-axis) and fine-mode effective radius (y-axis) based on size-distribution detail of AERONET at all available sites (11245 monthly averages in total)*

**8.3 how to simplify absorption?**  For both log-normal size distributions (to represent the fine-mode and the coarse mode) a non-absorbing (SSA=1.0) composition and a maximum absorbing composition are defined. Hereby the maximum absorbing type for each size mode had to be at (or exceed the) largest absorption potential of the MACv2 climatology at each month and location: SSA =0.77 for the fine-mode and SSA = 0.74 for the coarse mode. This way the local MACv2 absorption for each mode is simply determined by a mixture of two equal size aerosol types with different absorption potential.  Annual averages of MACv2 suggested absorbing (and non-absorbing) type fractions are presented for fine-mode and coarse mode in Figure 12.

**8.4 steps of application:**  Once, look-up tables for the four aerosol types (2 sizes, 2 absorption strengths) have been created (to convert solar reflectance at any viewing direction into an AOD), contributing  AOD values are combined into a total AOD, based on MACv2 local fractions for fine-mode AOD (AODf/AOD) and absorption type for each of the two size-modes. Then, the first guess for the total AOD is compared to the suggested total AOD of MACv2 at that location. In case of a difference an alternate fine-mode fraction is applied for an improved final AOD estimate.  The alternate AOD fine-mode fraction choice is based on multi-annual statistical relationships from ICAP satellite data assimilation ensemble between AOD and fine-mode AOD (*Peng et al., 2018*). Derived fine-mode fractions at twice and half the average AOD are compared to the fine mode fraction of the average AOD in Figure 12.

[Figure]

***Figure 12*** *MACv2 associated mid-visible global maps for assistance in aerosol model choices in satellite retrievals.  Annual averages (left block, left column) and seasonal averages (right block) are presented for total AOD at 550nm and absorption type fractions for fine-mode AOD and coarse mode AOD. The AOD split is assigned by the fine mode fraction whose annual averages (left block, right column) are presented for twice the MACv2 AOD, for the standard MACv2 AOD and half the MACv2 AOD. In the context of the absorption fraction, absorption types have a single scattering albedo of .7693 (SSAf) and .7457 (SSAc) for fine-mode and coarse-mode, respectively, while the alternate scattering types have a single scattering albedo of 1.0. Note, the coarse mode absorption includes enhanced absorption by larger mineral dust sizes. Also shown in the left block are annual averages for (today's) anthropogenic AOD and the complementary scattering fractions.*

**9. summary**

**why MAC?** The Max-Planck Aerosol Climatology MAC informs on typical aerosol properties on a monthly global basis. The climatology offers likely aerosol properties which can be applied for more general evaluations of 'bottom-up' modeling and in satellite remote sensing.  In its second version, MACv2 now extracts in a 'top-down' approach also compositional information, which in turn assists in a more consistent and confident spectral extension for the aerosol radiative properties needed in broadband radiative transfer simulations. The radiative properties, which can be accommodated to any radiative transfer scheme in global modeling, offer simple input for fast and more direct (since linked to observations of aerosol optics) alternative when estimating aerosol direct radiative impacts in global modeling. And with a simple satellite retrieval based relationship between aerosol and particle number concentrations (*Kinne 2019, Appendix A*) also first order aerosol indirect radiative impacts can then be included.

**how different is MACv2?**  The MACv2 properties are listed in Table 4 and its annual global averages of selected aerosol properties  compared to those of the older MACv1 version (Kinne et al., 2013), to the AeroCom 1 ensemble average (Kinne et al., 2006, Schulz et al., 2006) and to ICAP ensemble modeling (Peng et al., 2019).

*Table 4 MACv2 properties in comparison to those of MACv1 (Kinne et al., 2013), of the AeroCom phase 1 median (Kinne et al., 2006) and of the ICAP ensemble modeling (Peng et al., 2019). The MACv2 radiative forcing estimates are from a companion paper (Kinne, 2019).*

| | MACv1 | MACv2 | AeroCom 1 | ICAP |
|---|---|---|---|---|
| spatial resolution (deg) | 1x1 | 1x1 | 1x1 | 1x1 |
| temporal resolution | monthly | monthly | monthly | 3hr |
| reference year | 2000 | 2005 | | |
| quality references | AERONET (->2010) | ANET +MAN (->2015) | | |
| scaling background | AeroCom 1 | Aerocom 1 | | |
| AOD, 550nm | 0.13 | 0.12 | 0.12 | 0.14 |
| ant AOD, 550nm | 0.037 (Dentener) | 0.032 (Lamarque) | 0.029 | |
| AAOD, 550nm | 0.0054 | 0.0072 | 0.0043 | |
| FMF  (AODf/AOD) | 0.47 | 0.53 | 0.46 | 0.47 |
| absFMF  (AAODf/AAOD) | not available | 0.71 | | |
| vertical distribution | ECHAM | ECHAM  (CALIPSO) | | |
| Components | NO | YES | | |
| CCN (IN) estimates | NO | YES | | |
| spectral rad. properties | AOD, SSA, ASY | AOD, SSA, ASY | | |
| TOA (all aerosol - sol+IR) | -1.6 W/m2 | -1.1 W/m2 | | |
| TOA (all aerosol - solar) | -2.1 W/m2 | -1.8 W/m2 | | |

| TOA (ant aerosol) | -0.5 W/m2 | -0.35 W/m2 | -0.22 W/m2 | |

Compared to MACv1 (in terms of global annual averages) the total AOD and its anthropogenic fraction are slightly decreased. In contrast the fine-mode fraction is increased and in particular the (mid-visible) aerosol absorption is stronger.  A significant 29% of this absorption is attributed to dust (size) also now yielding a more positive direct IR radiative effect. The direct radiative forcing by today's anthropogenic aerosol with MACv2 data is estimated at -0.35 W/m2. This is less negative than with MACv1 input but still more negative than the AeroCom 1 average.

**What is next ?** There are several elements for improvement. One limitation is that the climatology refers to an average year at current emissions. To improve the relevance to a particular year since 2000, fine-mode AOD and coarse-mode AOD regional anomalies (as offered via MODIS or MISR sensor data) could be applied. Another limitation is the current restriction to a single local aerosol property per-month. For future versions also local pdfs will be offered and in particular inter-relationships how variability to aerosol amount impacts other aerosol properties (e.g. the fine-mode-fraction dependence on AOD offered by ICAP data). Also an update is needed for the temporal scaling for anthropogenic aerosol which currently goes back to MACv1.  There are current efforts underway for an update based on 'bottom-up' transient AeroCom simulation, which also consider the nitrate component. These simulations will also inform on and compositional changes of anthropogenic aerosol over time, which is considered as another element for improvement.

**Resources**

[revised manuscript text omitted]

**Tackett, J. D. M. Winker, B. J. Getzewich, M. A. Vaughan, S. A. Young and J. Kar (2018): *CALIPSO lidar level 3 aerosol profile product*** *Atmos. Meas. Tech., 11, 4129-4152, 2018*

**T**extor, C., M.Schulz, S.Guibert, S.Kinne, Y.Balkanski, T.Berntsen, T.Berglen, O.Boucher, M.Chin, F.Dentener, T.Diehl, H.Feichter, D.Fillmore, S. Ghan, P.Ginoux, S.Gong, A.Grini, J.Hendricks, L.Horrowitz, I.S.A.Isaksen, T.Iversen, , A.Kirkevag, D.Koch, J.E.Kristjansson, M.Krol, A.Lauer, J.F.Lamarque, X.Liu, V.Montanaro, G.Myhre,  J.Penner, G.Pitari, S.Reddy, O.Seland, P.Stier, T.Takemura, X.Tie (2007): ***The effect of harmonized emissions on aerosol properties in global models – an Aerocom experiment***, *Atmos. Chem. Phys., 7, 4489-4501.*

**S**mirnov, A., B. Holben, I. Slutsker, D. Giles, C. R. McClain, T. Eck, S. Sakerin, A. Macke, P. Croot, G. Zibordi, P. Quinn, J. Sciare, S. Kinne, M. Harvey, T. Smyth, S. Piketh, T. Zielinski, A. Proshutinsky, J. Goes, N. Nelson, P. Larouche, V. Radionov, P. Goloub, K. Krishna Moorthy, R. Matarrese, E. Robertson and F. Jourdin (2009): ***Maritime Aerosol Network as a component of Aerosol Robotic Network***, *J. Geophys. Res., 114,* D06204, *doi:10.1029/2008JD011257.*

**Appendix A**

Global maps for annual average radiative properties (AOD, SSA, ASY) are listed at selected wavelengths for MACv2 fine-mode and coarse-mode aerosol as well as for total and anthropogenic aerosol for today.

[Figure]

***Figure A1*** *MACv2 fine-mode radiative properties of AOD, SSA and ASY  (at .45, .55, 1.0 and 1.6um)*

[Figure]

***Figure A2*** *MACv2 coarse-mode radiative properties of AOD, SSA and ASY  (at .45, .55, 1.0 and 10um)*

[Figure]

***Figure A3*** *MACv2 total radiative properties of AOD, SSA and ASY (at .45, .55, 1.0 and 10 um) for today*

[Figure]

***Figure A4*** *MACv2 anthropogenic radiative properties of AOD, SSA and ASY (at .45, .55, 1.0 and 10 um) for today's conditions. In MACv2 anthropogenic AOD is a fraction of the fine-mode AOD and properties for anthropogenic SSA (or composition) and the anthropogenic ASY (or aerosol size) are assumed to be that of fine-mode aerosol and do not change over time.*

In all above plots the values below the labels indicate global annual averages.

**Appendix B**

Global maps are presented for aerosol altitude distributions of AOD with respect 4 atmospheric layers: 0-1km (row 4), 1-3km (row 3), 3-6km (row 2) and 6-12km (row 1) above sea-level.

[Figure]

*Figure B1* *ECHAM-HAM based multi-annual relative altitude distribution fractions (sum over all layer is 1) for total aerosol (left column), for the fine-mode (center column) and for the coarse mode (right column).*

For MACv2 the relative vertical distribution for fine-mode and coarse mode of ECHAM-HAM are applied. The multiplication of these fractions with the AOD, AODf and AODc column values of MACv2 yields the vertical AOD distributions of Figure B2.

The relative vertical distribution for total aerosol by ECHAM-HAM is mainly presented to illustrate in Figure B3 the relative altitude AOD distribution differences of multi-annual CALIPSO data of version 2 and version 3 data. The CALIPSO data show in a relative sense much more aerosol in the lowest 1km but it cannot be ruled out that this is an artifact (as low level clouds may be counted as aerosol).

[Figure]

**Figure B2** *MACv2 AOD vertical distribution with the ECHAM-HAM vertical scaling for total aerosol (left column), for fine-mode aerosol (center column) and the coarse mode aerosol (right column).*

[Figure]

**Figure B2** *Multi-annual relative altitude distribution fractions (sum over all layer is 1) for total aerosol between ECHAM_HAM (left column) and CALIPSO version2 (center column) and version 3 (right column).*

In all above plots the values below the labels indicate global annual averages

---

## Referee Report (RR1)

**Review of Aerosol radiative effects with MACv2 by S. Kinne for ACP**

Stephen E. Schwartz Brookhaven National Laboratory ses@bnl.gov June 5, 2019

I have examined the submitted manuscript Aerosol radiative effects with MACv2 (file name acp-2018-949-manuscript-version4.pdf) and the companion manuscript in review at *Tellus* The MACv2 Aerosol Climatology (file name MACv2\_clima\_2019\_06.pdf)

To great extent the manuscript under review, and also the companion manuscript, read like a reports, stating what was done, to support model calculations using these results. Specifically the present manuscript reports the influence (direct, indirect, i.e., Twomey effect) of aerosols (natural and anthropogenic) on atmospheric irradiance (at the top of the atmosphere and at the surface) as a function of location (1° x 1°), in monthly average and annual average and in global average, based on a similarly resolved aerosol climatology described in the companion manuscript; and also historically, at 40-year intervals from 1865 to 2065. There is no question that such a report is essential to understanding the forcing that would be employed in climate model calculations using this climatology and is thus essential to understanding the output that would result from such climate model calculations, and in this sense is an admirable activity and one to be encouraged. Although such documentation is essential to bolster confidence in subsequent calculations, the question remains whether publication of this documentation in the refereed scientific literature is the appropriate means of providing this documentation to the interested community, as opposed to simply being made available in online files. On reflection my decision is to favor publication. There is much that was done that goes well beyond "turn the crank" in terms of decisions that were made how to treat this or that issue and justification thereof. That said, there is also a lot of "turn the crank," but I must say that the crank seems very well turned.

As anyone who peruses this manuscript will immediately discern, that, including the appendices, it contains several dozen multi panel figures with each panel consisting of multiple global maps of various aerosol radiative quantities. In general I am a great fan of what Tufte (1983) refers to as "small multiples", multiple images in the same format that in the aggregate tell a story that cannot be told by a single image. Decipher the image once, and the deciphering pays off in the ready comprehension of the multiple images.

Where do all these numbers come from? And there are a lot of numbers. Each map presents results at  $1^{\circ}$  lat x  $1^{\circ}$  long or 64800 numbers. Of course these numbers cannot be accurately read off the figures, but the figures do serve to give a very valuable two dimensional picture of the pattern of the quantity being plotted.

The numbers are available at as netcdf files

**ftp://ftp-projects.zmaw.de/aerocom/climatology/MACv2\_2018/**

so the user can download and manipulate (and use) them to his or her heart's content to bring out the results in much finer detail, or to use in subsequent calculations, compare with observations or other model calculations, etc. The website is alive at the time of writing of this review although some of the files seem to have been entered or updated subsequent to submission of the manuscript. I did not attempt

to download or examine any of the files. Presumably the files will be frozen as of some date, and any changes will be noted, with prior versions remaining available.

In this context the question might be raised whether it is worth the (electronic) journal pages to present all these figures. I would say yes. I found it very interesting, for example, to compare TOA forcing and surface forcing efficiencies, (W m-2 AOD-1), for different aerosol components and try to understand the reasons behind the spatial variation. The author does present some discussion to some of these spatial variations, though perhaps not enough. Maps are presented for both direct and indirect aerosol effects, and total aerosol effects and anthropogenic enhancement, by species (e.g., sulfate) or groups of species (e.g., organics), annual, and monthly. So quickly this adds up to a lot of maps and a lot of data files and a lot of bytes per file (I saw some files as large as 146 Mbytes).

From the perspective of a reviewer I must address the question of how the numbers are generated and the extent to which they can be believed, that is their accuracy. The approach taken by the author is to combine information about the distribution and properties of aerosols from both modeling and observations. It seems that the author places more confidence in aerosol amount (extensive variable: optical depth, column burden) as constrained by observations, but uses models to apportion to aerosol substances and to infer intensive properties (phase function, asymmetry parameter, effective radius, composition, single scattering albedo). This seems reasonable. Observations (from the surface or satellite) cannot give much information about the intensive variables, which variables greatly influence the forcing efficiency. On the other hand, available aerosol models yield a diversity in extensive quantities that would render them useless without constraint.

Here I might register a concern. The aerosol models used for the present study date from the mid 2000's as presented in the first AeroCom study (of which the present author was lead investigator) as documented in Kinne et al. (2006) and Textor et al. (2006), two enormously important papers. To my thinking one of the insights of the that study (perhaps not intended by the investigators) is shown in Figure 3 of Kinne et al. paper, which shows the total annual global AOD from some 16 models agreeing (for all but one) with surface based and satellite based products within 10% or so, but the components contributing to the total AOD differing substantially from model to model, e.g., a factor of 3 for sulfate, a factor of 5 for organic and black carbon. Similarly also for the intensive variable mass scattering efficiency, with Max/Min 6.7 for sulfate, 3.5 for black carbon, 2.8 for organics. Pretty clearly the agreement with observation of global annual AOD for each of these models within about 10% or so is a consequence of various compensations. Another enormously valuable figure in the Kinne *et al.* paper is Figure 4, which presents maps of spatial distribution (annual average) what the authors denoted "central diversity", max/min ratios of the central 2/3 of the modeled quantities, for yearly averages, which are several fold for several of the quantities, and more than an order of magnitude for aerosol water over continental regions. Variations in other model output quantities such as residence times of modeled aerosols are presented in the Textor et al. paper. One can only surmise that the divergence in maps of monthly distributions would be even greater, and more importantly that there is a large amount of compensation in the models for them to be able to obtain such accurate global annual mean AOD.

I go into this detail regarding the results of the AeroCom study because the chemical transport model results that serve as the basis for the present analysis, as described in the companion *Tellus* manuscript, are in fact taken from the 2006 AeroCom intercomparison, specifically as the monthly median values of 14 different models that participated in that study. So the model diversity reflected in the 2006 papers should raise if not a red flag, at least a yellow flag, as to the confidence that can be placed in the apportionment of optical properties, radiative effects, and CCN properties that are used in the study under review and that are the basis for attributing these effects to anthropogenic and natural sources.

Given this situation, should the present analysis be rejected out of hand? I do not think so. I think the study presented in the manuscript under review (and the companion paper submitted to *Tellus*) probably represents the state of the art in such analyses, although I do wonder whether there are not more recent (not necessarily more accurate) aerosol chemical transport models the results of which might have been used in the present study

I called attention above to the large divergence in aerosol water across the models, an order of magnitude max/min ratio in the central 2/3 of the models over continental regions in annual average. This should be viewed as a further red flag. The effect of relative humidity on the amount of condensed phase water in an aerosol and the resultant radiative effect are well known from a theory perspective (given knowledge of particle composition; e.g., Nemesure *et al.*, 1995) but enormously difficult to represent in large-scale models. A brief period during which there is a relative humidity over 90 or 95% at the top of a boundary layer containing hygroscopic aerosol can dominate the instantaneous optical depth and contribute substantially to the 24-hour forcing but would never be captured models absent high 3D spatial resolution and temporal resolution. As a result, forcing based on average RH will certainly underestimate actual forcing.

A further question with respect to the publishability of the present manuscript is the extent to which the manuscript presents scientific findings. Here "findings" can be taken broadly. For example does the climatology developed change or confirm present quantitative understanding of aerosol forcing, at various levels of spatial and temporal resolution? Turning to the abstract, the second paragraph indeed reports such findings:

Likely present-day global annual radiative effects for anthropogenic aerosol are (1) a climate cooling of -1.0 W/m2 at the top of the atmosphere (TOA), (2) a surface net flux-reduction of -2.1 W/m2 and (3) by difference an atmospheric effect of a +1.1 W/m2. This associated atmospheric solar heating is almost entirely a direct effect. For the climate relevant TOA response the indirect effect (-0.65 W/m2) on average dominates the direct effect (-0.35 W/m2). In contrast, at the surface the direct effect (-1.45 W/m2) dominates on average the indirect effect (-0.65 W/m2).

Of course, such estimates have been provided previously, importantly in the IPCC AR5 Assessment, which summarizes a large body of previous work. However a key distinction between that assessment and the findings quoted above is the absence in the abstract of any statement of uncertainty associated with these numbers, and as well comparison with prior estimates. To be sure, these uncertainty estimates are provided in the body of the manuscript, (p. 24, line 29),

Considering these different uncertainties, a [-]0.7 to -1.6 W/m2 range is estimated for present-day aerosol forcing (assuming a year 1850 reference), with the best guess value aerosol forcing at about -1.0 W/m2. Hereby the less negative lower bound is more certain than the more negative upper bound.

(a minus sign before the 0.7 appears to be required but is missing), with some justification provided in the subsequent paragraphs. To my thinking such an uncertainty range needs much stronger justification than is provided, especially given the strong sensitivity of forcing to uncertainties in AOD, single scattering albedo, asymmetry parameter, and surface reflectance (e.g., McComiskey *et al.*, 2008) as well as uncertainty due to aerosol water noted above and the uncertainty in attribution to anthropogenic components arising out of the model diversity. Similar considerations apply to the estimate of uncertainty in the indirect effect that is folded into the total forcing and uncertainty reported. I would

hope that the author would re-think his estimate of uncertainty in light of these considerations, and also that his (revised) estimate of uncertainty be included in the abstract.

More broadly, I would hope that the author would reflect on the comments I have provided and perhaps elaborate in the manuscript on the issues raised or appropriately revise.

Finally, I commend the author on the enormous body of work represented in these two manuscripts. I would hope that others preparing such a forcing climatology would do such a careful and well documented job.

**Additional comments, by page and line**

1,22. At TOA?1,38. ... is poorly defined; maybe better ... is quite uncertain.3,28. radiative properties --> optical properties16,36-7. Do not understand what is meant.

**References:**

- Kinne, S., Schulz, M., Textor, C., et al., An AeroCom initial assessment optical properties in aerosol component modules of global models, Atmos. Chem. Phys., 6, 1815-1834, https://doi.org/10.5194/acp-6-1815-2006, 2006.
- McComiskey, A., Schwartz, S. E., Schmid, B., Guan, H., Lewis, E. R., Ricchiazzi, P., & Ogren, J. A. (2008). Direct aerosol forcing: Calculation from observables and sensitivities to inputs. Journal of Geophysical Research: Atmospheres (1984–2012), 113(D9).
- Nemesure S., Wagener R., and Schwartz S. E. (1995) Direct shortwave forcing of climate by anthropogenic sulfate aerosol: Sensitivity to particle size, composition, and relative humidity. *J. Geophys. Res.* **100**, 26105-26116.
- Textor, C., Schulz, M., Guibert, S., Kinne, S., Balkanski, Y., Bauer, S., Berntsen, T., Berglen, T., Boucher, O., Chin, M. and Dentener, F., 2006. Analysis and quantification of the diversities of aerosol life cycles within AeroCom. Atmospheric Chemistry and Physics, 6(7), pp.1777-1813.
- Tufte E. R., The Visual Display of Quantitative Information, Graphics Press, Cheshire, Ct, 1983

---

## Author Response (AR2)

Steve

Thanks for your very generous and editorial review. You raised some points and concerns. So let me get to those.

**frozen version**:  the version that is now placed (along with programming code) on the ftp-site is frozen in as MACv2 – also in the context of the Tellus and ACP paper (potential future changes will be referred to as potential versions 3 or 4 …). Recent updates were done to made corrections to the format (correct times in netcdf files), to add programming code or to cater to some special MACv2 output requests

**concerns on using output from global modeling:** It is nice that you refer to the 2006 AeroCom modeling paper, which displayed a lot of diversity. In those 'bottom-up' approaches there were different paths (e.g. component strengths, water uptake) to eventually come up with approximately the same global AOD. Thus you state concerns that this will affect MACv2 optical properties. These modeling uncertainties are avoided here by relying on ambient (measured) AOD (amount) and (observation tied via inversions) AAOD (absorption) data. But it is true that AODf (SU+OC+BC) and AODc (DU+SS) monthly global distributions from 'bottom-up' simulations are applied for spatial context. But firstly, the observational data are just scaled (no absolute modeling values come into play) and secondly, a 14 model ensemble median distribution is used so that spatial distribution outliers of particular models are avoided.  There is certainly a valid question, why model data of AEROCOM phase1 and not from the more recent phases 2 and 3 are used. This is simply, because many more submissions into the AeroCom data-base are not well tested. And with a focus on anthropogenic fine-mode AOD or a related process, often natural components (dust and seasalt) are not given the needed attention. Many model display for natural aerosol extreme features so that the diversity of ensemble data of more recent model submissions for total AOD (and also AODf and AODc) is much larger. That stated, for the fine-mode AOD (AODf) there is improvement. This is also the reason, why for the analysis of the 1.indirect effect model output of AeroCom phase 2 is discussed on compared to similar associations from satellite observations. The unique approach to component radiative impacts in this paper is that AOD component contributions (and even dust coarse sizes) are in a 'top-down' approach deduced from (2D column) aerosol optical properties (AODf, AODc, AAODf, AAODc ) and ambient REf (for fine-mode size)… so that component assignments  are possible without the need to deal with water uptake uncertainties.
For the anthropogenic definition there, there is admittedly a strong reliance on global modeling although only in a relative sense (as ratios - and NOT absolute values - from ensemble modeling are applied).

**concerns about the missing uncertainty discussions**: There are many uncertainties and I tried to focus on those that matter most:  anthropogenic definition and representation of indirect effects.  For instance: What is the pre-industrial state? Can it be reduced to fine-mode AOD additions? How much fine mode AOD was there? How much BC was there?  Is it sufficient to limit indirect effect to Twomey? How important are spatial variation of the Twomey effect?
 I do not think that other issues matter that much as this paper works with observations, which use data on ambient aerosol (water uptake is non-issue) and on absorption (so that SSA and even dust size).  The impact of surface reflection (and temperature for large dust) are an issue, but only if they change (and

he in the forcing simulations they do not). So in summary I consider my uncertainty estimate (under all asumptions made) as fair.  And the uncertainty range, as suggested, was already stated via the range in the abstract.

minor points

- thanks for noticing the missing negative sign for the likely aerosol TOA forcing range

-'at TOA' … has been added (and climate warming by BC is stronger at all-sky than at clear-sky) still +0.55W/m2 would be the maximum BC warming and only if all BC todays would be anthropogenic … what is not. So Aeronet based data are help to set an upper limit rejecting the large BC warming suggestions on global scales (e.g. Ramanathan, Carmichael). And considering all the scattering co-emitters with BC emission the climate warming mitigation even on short time-scale is rather limited

- I would stick with radiative properties (in describing the single scattering properties at 550nm)

Dear reviewer.

Thanks for having a look at the revised paper !

The author has adopted most of the suggested changes, and satisfactorily justified not adopting the others, apart from one of the original major comments regarding omitting cloud adjustments in their work

I am aware that there are likely cloud adjustments to the imposed radiative forcing. Although my method has some advantages (e.g. as not being affected you variability of freely developing and possibly also even so slightly changing clouds), my method is static and cannot account for these aerosol induced changes other than the (the via satellite associations) prescribed 1.indirect effect on low altitude clouds. I am banking here on multi-decadal climate simulations with different models, which suggest that these adjustments (when the same direct and indirect forcing is imposed) are relatively minor.

"In particular, the paper aims to provide an estimate of the aerosol forcing, but ignoring cloud adjustments seems a significant omission and requires much stronger justification, particularly when discussing potential uncertainties. For example, on P2L9-10 the author states that feedbacks (adjustments) can be considered secondary, though the provided citation makes no such assertion – they just assume it for their purposes."

That paper (Fiedler et al) makes as distinction between static and instantaneous radiative forcing (based on direct and satellite based 1 direct effects as in my approach) and one, where feedbacks on clouds are permitted (there referred to as 'effective radiative forcing) … and the differences are on the order of 10%, which I consider secondary. I will try to strengthen this point in the uncertainty discussions

Author response: "Here off-line radiative transfer simulations a used. Thus, no feedbacks or (earth system) adjustments to the forcing can be considered. Fortunately, there is a recent RFMIP exercise where in long-term simulations (with fixed SST) compare results of the (direct and first indirect) forcing from dual calls with the effective radiative forcing, which allowed atmospheric adjustments to the forcing in global models. The forcing differences are found to be relatively small on the order of 10% (Fiedler et al., 2019)."

Yes that was (and still would be) my response.

While I appreciate these cannot easily be accounted for in the off-line framework used by the author, this omission requires justification in the introduction and an acknowledgment in the discussion on uncertainties. In particular, neither Fiedler et al. 2017 or 2019 show that cloud adjustments to the aerosol indirect effect are small. They choose to ignore cloud adjustments to simplify their framework, see in particular Section 2.2.2 of Fiedler et al. 2017. Whilst a similar argument could be made here, it should be done explicitly.

It is true that the rapid adjustments directly via cloud microphysics are not explicitly simulated when MACv2-SP is implemented in the radiation code, although one can incorporate a net effect on the radiation balance as effective parameter. Some models, however, choose to implement the MACv2-SP perturbation of the cloud droplet number in their cloud microphysics, not the radiative transfer calculation only. This is the case for EC-Earth so far (see Fiedler et al., 2019). The model therefore simulates cloud adjustments like more complex aerosol-climate models do. The contributions from cloud adjustments will be further assessed in RFMIP, but based on the current evidence the contribution of rapid adjustments is small compared to the instantaneous radiative forcing.

That said, the point should be made that cloud feedbacks are ignored, at least in the uncertainty discussions.  We know relatively little about cloud responses and even simulations with global models have to be questioned as smaller scale processes and responses are in global models are heavily parameterized and possibly unable to represent actual effects.

the abstract has been rewritten and the second sentence now states: "This model-setup cannot address rapid adjustments by clouds, but these effects are believed to be small."

In the uncertainty estimates this sentence was added
"Finally all presented results refer to an instantaneous impact. Short-term cloud adjustments are not included, but they are much smaller (on the order of 10%) in comparison (Fiedler et al., 2019) with a tendency to reduce the aerosol radiative effects".

---

## Author Response (AR3)

the sentence in the abstract was changed as suggested by the co-editor

"This model-setup cannot address rapid adjustments by clouds, but current evidence suggests their contribution to be small when compared to the instantaneous radiative forcing."